# ORSO: ACCELERATING REWARD DESIGN VIA ONLINE REWARD SELECTION AND POLICY OPTIMIZATION

**Chen Bo Calvin Zhang**[*†‡], **Zhang-Wei Hong**[†], **Aldo Pacchiano**[§¶], **Pulkit Agrawal**[†]

Improbable AI Lab, Massachusetts Institute of Technology[†]      ETH Zurich[‡]
Boston University[§]      Broad Institute of MIT and Harvard[¶]

## ABSTRACT

Reward shaping is critical in reinforcement learning (RL), particularly for complex tasks where sparse rewards can hinder learning. However, choosing effective shaping rewards from a set of reward functions in a computationally efficient manner remains an open challenge. We propose Online Reward Selection and Policy Optimization (ORSO), a novel approach that frames the selection of shaping reward function as an online model selection problem. ORSO automatically identifies performant shaping reward functions without human intervention with provable regret guarantees. We demonstrate ORSO's effectiveness across various continuous control tasks. Compared to prior approaches, ORSO significantly reduces the amount of data required to evaluate a shaping reward function, resulting in superior data efficiency and a significant reduction in computational time (up to $8\times$). ORSO consistently identifies high-quality reward functions outperforming prior methods by more than 50% and on average identifies policies as performant as the ones learned using manually engineered reward functions by domain experts. Code is available at https://github.com/Improbable-AI/orso.

## 1 INTRODUCTION

Reward functions are crucial in reinforcement learning (RL; Sutton & Barto (2018)) as they guide the learning of successful policies. In many real-world scenarios, the ultimate objective involves maximizing long-term rewards that are not immediately available (Vecerik et al., 2017; Rengarajan et al.; Vasan et al.), making optimization challenging. To address this, practitioners often introduce shaping rewards (Margolis & Agrawal, 2022; Liu et al., 2024; Mahmood et al., 2018; Ng et al., 1999; Park et al., 2024) to provide additional guidance during training. Instead of directly maximizing the task reward ($R$), it is therefore common for the RL algorithm to maximize an easier-to-optimize shaped reward function $F$ in the hope of obtaining high performance as measured by the task reward, $R$. While shaping rewards are designed to guide the agent to complete a task, maximizing them does not necessarily solve the task. For instance, an agent tasked with finding an exit (i.e., longer-term reward in the future) may be provided with shaping rewards to avoid obstacles. However, the task's success ultimately depends on reaching the exit, not just avoiding obstacles. If poorly designed, the shaped rewards $F$ can mislead the agent because prioritizing the maximization of $F$ may not maximize $R$ (Chen et al., 2022; Agrawal, 2021; Hadfield-Menell et al., 2017; Ng et al., 1999), leading to training failure or suboptimal performance.

Designing effective shaping reward functions $F$ is therefore challenging and time-consuming. It requires multiple iterations of training agents with different shaping rewards, evaluating their performance on the task reward $R$, and refining $F$ accordingly. This process is inefficient due to the lengthy training runs and because the performance measured early in training may be misleading, making it challenging to quickly iterate over different shaping rewards.

To address this challenge, we propose treating the selection of the shaping reward function as an exploration-exploitation problem and solving it using provably efficient online decision-making algorithms similar to those in multi-armed bandits (Auer et al., 2002; Auer, 2002) and model selection (Agarwal et al., 2017; Pacchiano et al., 2020; Dann et al., 2024; Foster et al., 2019; Lee et al., 2021).

---

[*]Correspondence to cbczhang@mit.edu, pulkitag@mit.edu.

In this setup, each shaping reward function $F$ corresponds to an arm or model. The utility of a shaping reward function is $\mathcal{J}(\pi^f)$, the expected cumulative reward of the policy $\pi^f$ under the task reward $R$. We aim to efficiently select the shaping reward function that leads to the best policy under the task reward $R$. To achieve this, we allocate computational resources strategically, exploring each reward function sufficiently to evaluate its potential while avoiding excessive exploration so that enough computational budget is left to optimize the policy using the best reward function.

This approach presents unique challenges. Unlike standard multi-armed bandit settings with stationary reward distributions, the utility of a shaping reward function in our case is nonstationary. As the agent explores new parts of the state space during training, the task reward distribution changes. Additionally, we must balance *exploration* and *exploitation* of a set of shaping reward functions to efficiently allocate training time among these shaping rewards without committing too early to high-performing options or wasting time on low-performing ones.

We introduce *Online Reward Selection and Policy Optimization* (ORSO), an algorithm that efficiently selects the best shaping reward function from a set of candidate shaping reward functions that maximize the task performance. ORSO provides regret guarantees and adaptively allocates training time to each shaping reward based on a model selection algorithm at each step. Our empirical results across various continuous control tasks using the Isaac Gym simulator (Makoviychuk et al., 2021) demonstrate that ORSO identifies performant reward functions with 56% better performance on average compared to methods like EUREKA (Ma et al., 2024) within a fixed interaction budget. Moreover, ORSO consistently selects reward functions that are comparable to, and sometimes surpass, those designed by domain experts, all while using up to $8\times$ less compute.

## 2  PRELIMINARIES

**Reinforcement Learning (RL)**    In RL, the objective is to learn a policy for an agent (e.g., a robot) that maximizes the expected cumulative reward during the interaction with the environment. The interaction between the agent and the environment is formulated as a Markov decision process (MDP) (Puterman, 2014), $\mathcal{M} = (\mathcal{S}, \mathcal{A}, P, r, \gamma, \rho_0)$, where the $\mathcal{S}$ and $\mathcal{A}$ denote state and action spaces, respectively, $P : \mathcal{S} \times \mathcal{A} \rightarrow \Delta_{\mathcal{S}}$[1] is the state transition dynamics, $r : \mathcal{S} \times \mathcal{A} \rightarrow \Delta_{\mathbb{R}}$ denotes the reward function, $\gamma \in [0, 1)$ is the discount factor, and $\rho_0 \in \Delta_{\mathcal{S}}$ is the initial state distribution. At each timestep $t \in \mathbb{N}$ of interaction, the agent selects an action $a_t \sim \pi(\,\cdot\mid s_t)$ based on its policy $\pi$, receives a (possibly) stochastic reward $r_t \sim r(s_t, a_t)$, and transitions to the next state $s_{t+1} \sim P(\,\cdot\mid s_t, a_t)$ according to the transition dynamics. Here, $r$ is the task reward, also referred to as extrinsic reward (Chen et al., 2022). RL algorithms aim to find a policy $\pi^\star$ that maximizes the discounted cumulative reward, i.e.,

$$\pi^\star \in \arg\max_{\pi} \mathcal{J}(\pi) := \mathbb{E}\left[\sum_{t=0}^{\infty} \gamma^t r_t \,\middle|\, \begin{array}{l} s_0 \sim \rho_0, \ a_t \sim \pi(\,\cdot\mid s_t), \\ r_t \sim r(s_t, a_t), \ s_{t+1} \sim P(\,\cdot\mid s_t, a_t) \end{array}\right]. \tag{1}$$

**Notation**    We denote $\mathfrak{A}$ a reinforcement learning algorithm that takes an MDP $\mathcal{M} = (\mathcal{S}, \mathcal{A}, P, r, \gamma, \rho_0)$, a reward function $f$, a number of interaction steps with the environment $N$, and an initial policy $\pi_0$ as input and returns a policy $\pi^f = \mathfrak{A}_f(\mathcal{M}, N, \pi_0)$.

## 3  METHOD: REWARD DESIGN AS SEQUENTIAL DECISION MAKING

As previously stated, the reward function $r$ encodes the task objective but can be difficult to optimize using RL methods directly. We formalize the reward design problem as follows.

**Definition 3.1** (Reward Design). *Given $\mathcal{M}$ and $\mathfrak{A}$, the reward design problem aims to find a reward function $f : \mathcal{S} \times \mathcal{A} \rightarrow \Delta_{\mathbb{R}}$, with $f \in \mathcal{R}$, the space of reward functions, such that the policy $\pi^f = \mathfrak{A}_f(\mathcal{M})$ achieves an expected return under the task reward $r$, such that $\mathcal{J}\left(\pi^f\right) \approx \max_{r' \in \mathcal{R}} \mathcal{J}(\pi^{r'}) = \mathcal{J}\left(\pi^\star\right)$.*

While this could be achieved by running the algorithm $\mathfrak{A}$ on every possible reward function $r' \in \mathcal{R}$, this is computationally prohibitive. The reward space $\mathcal{R}$ can be extremely large, and attempting to

---

[1] $\Delta_{\mathcal{S}}$ denotes the set of probability distributions over $\mathcal{S}$.

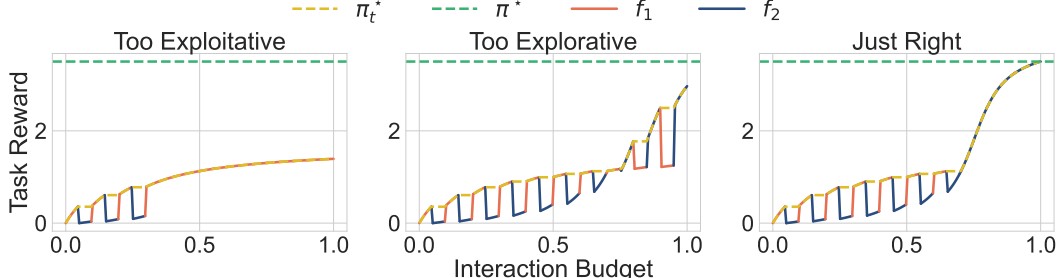

Figure 1: Comparison of three reward selection strategies given a fixed interaction budget. The green dashed line represents the task reward of the optimal policy, $\pi^\star$. The red and blue curves show the task rewards for policies trained with reward functions $f^1$ and $f^2$, respectively. The yellow curve, $\pi_t^\star$, tracks the maximum of the red and blue curves. **Left:** This selection strategy is overly exploitative, greedily selecting the reward function that seems to perform best early on but plateaus later in training. **Center:** On the other hand, this strategy continuously switched between $f^1$ and $f^2$, exploring the suboptimal reward function too much. **Right:** The ideal strategy initially explore $f^1$ and $f^2$, but quickly latches onto the better reward function.

optimize over all possible rewards is impractical, especially when the available interaction budget is constrained.

To make the problem tractable, we assume access to a finite set of candidate shaping reward functions $\mathcal{R}^K = \{f^1, \ldots, f^K\} \sim G(\mathcal{R})$, where $G$ is a distribution over the set of reward functions, that contains at least one near-optimal reward function and a budget of interactions $B$. If the budget does not allow training on each $f^i \in \mathcal{R}^K$, we need to allocate resources to gather useful information about the quality of each candidate, while simultaneously optimizing the most promising ones. This introduces a fundamental exploration-exploitation tradeoff. On the one hand, we must explore various rewards to identify high performers; on the other, we need to exploit promising candidates to train performant policies.

Figure 1 shows a simple example with two reward functions, $f^1$ and $f^2$, and three selection strategies. The selection strategies on the left and center are too exploitative and too explorative respectively. The exploitative selection strategy greedily spends the interaction budget on the reward function that performs best early in training, neglecting the potential of $f^2$. On the other hand, the selection strategy in the center equally explores both reward functions, even when one is clearly preferred. The exploration strategy on the right initially explores both reward functions but quickly latches onto the best one and spends the remaining interaction budget on the better reward function. The suboptimality gap at iteration $t$ of training can be measured via the *regret* of the best policy so far defined as $\text{reg}(t) = \mathcal{J}(\pi^\star) - \mathcal{J}(\pi_t^\star)$, where $\pi_t^\star := \arg\max_{\pi_\ell, \ell \in [t]} \mathcal{J}(\pi_\ell)$. The choice of $\pi_t^\star$ reflects a practical preference, as we are more interested in the best-performing solution available at a given point, rather than the most recent update. For instance, in deploying a robotic running policy, one would select the fastest policy observed thus far – assuming the objective is to run as fast as possible.

A further discussion of the online model selection problem can be found in Appendix B.

## 3.1 ORSO: ONLINE REWARD SELECTION AND POLICY OPTIMIZATION

In this section, we introduce ORSO (*Online Reward Selection and Optimization*), a novel approach to *efficiently* and *effectively* select shaping reward functions for reinforcement learning. Our method operates in two phases: (1) reward generation and (2) online reward selection and policy optimization.

**Reward Generation** In the first phase of ORSO, we generate a set of candidate reward functions $\mathcal{R}^K$ for the online selection phase. Given an MDP $\mathcal{M} = (\mathcal{S}, \mathcal{A}, P, r, \rho_0)$ and a stochastic generator $G$, we sample a set of $K$ reward function candidates, $\mathcal{R}^K = \{f^1, \ldots, f^K \mid \forall i \in [K], \ f^i : \mathcal{S} \times \mathcal{A} \to \Delta_{\mathbb{R}}, \ f^i \sim G\}$, from $G$ during the reward design phase. The generator $G$ can be any distribution

over the reward function space $\mathcal{R}$. For instance, if the set of possible reward functions is given by a linear combination of two reward components $c_1, c_2$, which are functions of the current state and action, such that $r(s, a) = w_1 c_1(s, a) + w_2 c_2(s, a)$, then the generator $G$ can be represented by the means and variances of two normal distributions, one for each weight $w_1, w_2$.

**Online Reward Selection and Policy Optimization**   Our algorithm for online reward selection and policy optimization is described in Algorithm 1. On a high level, the algorithm proceeds as follows. Given an MDP $\mathcal{M} = (\mathcal{S}, \mathcal{A}, P, r, \gamma, \rho_0)$, an RL algorithm $\mathfrak{A}$ and a reward generator $G$, we sample set of $K$ reward functions $\mathcal{R}^K \sim G$ and initialize $K$ distinct policies $\pi^1, \ldots, \pi^K$. At step $t$ of the reward selection process, the algorithm selects a learner $i_t \in [K]$ according to a selection strategy. We then run algorithm $\mathfrak{A}$, updating the policy corresponding to reward function $i_t$ to obtain $\pi^{i_t}$. Policy $\pi^{i_t}$ is simultaneously evaluated under the task reward function $r$ and the necessary variables for the model selection algorithm are then updated (e.g., reward estimates, reward function visitation counts, and confidence intervals). The algorithm returns the reward function $f_T^\star$ and the corresponding policy $\pi_T^\star$ that performs the best under the task reward function $r$. We discuss the implementation details in Section 5.

---

**Algorithm 1** ORSO: Online Reward Selection and Policy Optimization

---

**Require:** MDP $\mathcal{M} = (\mathcal{S}, \mathcal{A}, P, r, \gamma, \rho_0)$, algorithm $\mathfrak{A}$, generator $G$
 1: Sample $K$ reward functions $\mathcal{R}^K = \left\{ f^1, \ldots, f^K \right\} \sim G$
 2: Initialize $K$ policies $\left\{ \pi^1, \ldots, \pi^K \right\}$
 3: **for** $t = 1, 2, \ldots, T$ **do**
 4:     Select an model $i_t \in [K]$ according to a selection strategy
 5:     Update $\pi^{i_t} \leftarrow \mathfrak{A}_{f^{i_t}}(\mathcal{M}, \pi^{i_t})$
 6:     Evaluate $\mathcal{J}(\pi^{i_t}) \leftarrow \texttt{Eval}(\pi^{i_t})$
 7:     Update variables (e.g., reward estimates and confidence intervals)
 8: **end for**
 9: **return** $\pi_T^\star, f_T^\star = \arg\max_{i \in [K]} \mathcal{J}(\pi^i)$

---

**Choice of Selection Algorithm**   While ORSO is a general algorithm that can employ any selection method to pick the reward function to train on, the performance depends on the choice of algorithm.

For instance, using a simple selection method like $\varepsilon$-greedy introduces an element of exploration by occasionally selecting a random reward function (with probability $\varepsilon$), but it risks overcommitting to a seemingly promising reward function early on. This can lead to suboptimal performance if the chosen reward function causes the task performance to plateau in the long run. However, greedier methods, such as $\varepsilon$-greedy, can achieve lower regret if they commit to the optimal reward function early in the process. These methods are particularly effective when early performance signals are strong indicators of long-term success.

However, if initial performance is not a reliable predictor of future outcomes, these greedy approaches may struggle, as they risk prematurely locking onto suboptimal rewards. In contrast, more exploratory algorithms like the exponential-weight algorithm for exploration and exploitation (Exp3) (Auer et al., 2002) maintain a broader search, potentially discovering better rewards in the long run, especially in environments where early signals are less informative. We empirically validate different choices of selection algorithms in Section 5.

## 4   THEORETICAL GUARANTEES

In this section, we provide regret guarantees for ORSO with the Doubling Data-Driven Regret Balancing (D$^3$RB) algorithm by Dann et al. (2024). A discussion of the intuition behind the D$^3$RB algorithm and the full pseudo-code for ORSO with D$^3$RB is provided in Appendix C.

We first introduce some useful definitions for our analysis.

**Definition 4.1** (Definition 2.1 from Dann et al. (2024))**.** *The regret scale of learner $i$ after being played $t$ times is $\frac{\sum_{\ell=1}^{t} \mathrm{reg}(\pi_{(\ell)}^i)}{\sqrt{t}}$ where $\mathrm{reg}(\pi_{(\ell)}^i) = \mathcal{J}(\pi^\star) - \mathcal{J}(\pi_{(\ell)}^i)$ and $\pi_{(\ell)}^i = \mathfrak{A}_{f^i}(\mathcal{M}, \ell)$ in the reward design problem.*

*For a positive constant $d_{\min} > 0$, the regret coefficient of learner $i$ after being played for $t$ rounds is $d^i_{(t)} = \max\{d_{\min}, \sum_{\ell=1}^{t} \text{reg}(\pi^i_{(\ell)})/\sqrt{t}\}$. That is, $d^i_{(t)} \geq d_{\min}$ is the smallest number such that the incurred regret is bounded as $\sum_{\ell=1}^{t} \text{reg}(\pi^i_{(\ell)}) \leq d^i_{(t)}\sqrt{t}$.*

Dann et al. (2024) use $\sqrt{t}$ as this is the most commonly targeted regret rate in stochastic settings. The main idea underlying our regret guarantees is that the internal state of all suboptimal reward functions is only updated up to a point where the regret equals that of the best policy so far.

We assume there exists a learner that monotonically dominates every other learner.

**Assumption 4.2.** *There is a learner $i_\star$ such that at all time steps, its expected sum of rewards dominates any other learner, i.e., $u^{i_\star}_{(t)} \geq u^i_{(t)}$, for all $i \in [K], t \in \mathbb{N}$ and such that its average expected rewards are increasing, i.e., $\frac{u^{i_\star}_{(t)}}{t} \leq \frac{u^{i_\star}_{(t+1)}}{t+1}$, $\forall t \in \mathbb{N}$. This is equivalent to saying that $d^{i_\star}_{(t)} \geq d^{i_\star}_{(t+1)}$, for all $t \in \mathbb{N}$.*

Assumption 4.2 guarantees that the cumulative expected reward of the optimal learner $i_\star$ is always at least as large as the cumulative expected reward of any other learner and that its average performance increases monotonically.

Following the notation of Dann et al. (2024), we refer to the event that the confidence intervals for the reward estimator are valid as $\mathcal{E}$.

**Definition 4.3** (Definition 8.1 from Dann et al. (2024))**.** *We define the event $\mathcal{E}$ as the event in which for all rounds $t \in \mathbb{N}$ and learners $i \in [K]$ the following inequalities hold*

$$-c\sqrt{n^i_t \ln \frac{K \ln n^i_t}{\delta}} \leq \widehat{u}^i_t - u^i_t \leq c\sqrt{n^i_t \ln \frac{K \ln n^i_t}{\delta}} \tag{2}$$

*for the algorithm parameter $\delta \in (0, 1)$ and a universal constant $c > 0$, where $n^i_t = \sum_{\ell=1}^{t} \mathbb{1}(i_\ell = i)$.*

Then we can refine Lemma 9.3 from Dann et al. (2024) in the case where Assumption 4.2 holds.

**Lemma 4.4.** *Under event $\mathcal{E}$ and Assumption 4.2, with probability $1 - \delta$, the regret of all learners $i$ is bounded in all rounds $T$ as*

$$\sum_{t=1}^{n^i_T} \text{reg}(\pi^i_{(t)}) \leq 6d^{i_\star}_T \sqrt{n^{i_\star}_T + 1} + 5c\sqrt{(n^{i_\star}_T + 1) \ln \frac{K \ln T}{\delta}}, \tag{3}$$

*where $d^{i_\star}_T = d^{i_\star}_{(n^{i_\star}_T)}$.*

We provide the proof for Lemma 4.4 in Appendix D. Lemma 4.4 implies that when Assumption 4.2 holds, the regrets are perfectly balanced. This is in stark contrast with the regret guarantees of Dann et al. (2024) that prove the D$^3$RB algorithm's overall regret to scale as $\left(\bar{d}^{i_\star}_T\right)^2 \sqrt{T}$ where $\bar{d}^{i_\star}_t = \max_{\ell \leq t} d^{i_\star}_\ell$. Instead, our results above depend not on the monotonic regret coefficients $\bar{d}^{i_\star}_t$ but on the true regret coefficients $d^{i_\star}_t$. Even if learner $i_\star$ has a slow start (and therefore a large $\bar{d}^{i_\star}_T$), as long as monotonicity holds and the $i_\star$-th learner recovers in the later stages of learning, our results show that D$^3$RB will achieve a regret guarantee comparable with running learner $i_\star$ in isolation.

## 5 Practical Implementation and Experimental Results

In this section, we present a practical implementation[2] of ORSO and its experimental results on several continuous control tasks. We study the ability of ORSO to design effective reward functions with varying budget constraints. We also study how different sample sizes, $K$, of the set of reward functions $\mathcal{R}^K$ influence the performance of ORSO and compare different selection algorithms.

This section is structured as follows. First, we present the experimental setup, including the environments and baselines, and the practical consideration of the reward generator $G$ and the algorithms used in the online reward selection phase. Then, we present the main results and ablate our design choices. Further experimental results can be found in Appendix H

---

[2]The code for ORSO is available at `https://github.com/Improbable-AI/orso`

## 5.1 EXPERIMENTAL SETUP

**Environments and RL Algorithm**  We evaluate ORSO on a set of continuous control tasks using the Isaac Gym simulator (Makoviychuk et al., 2021). Specifically, we consider the following tasks: CARTPOLE and BALLBALANCE, which are relatively simple; two locomotion tasks, ANT and HUMNAOID, which have dense but unshaped task rewards – for instance, the agent is rewarded for running fast, but the reward function lacks terms to encourage upright posture or smooth movement; and two complex manipulation tasks, ALLEGROHAND and SHADOWHAND, which feature sparse task reward functions.

Our policies are trained using the proximal policy optimization (PPO) algorithm (Schulman et al., 2017), with our implementation built on CleanRL (Huang et al., 2022). We chose PPO because Makoviychuk et al. (2021) provide hyperparameters, which we use, that enable it to perform well on these tasks when using the human-engineered reward functions.

### 5.1.1 BASELINES

In our experiments, we consider three baselines. We analyze the performance of policies trained using each reward function detailed below. We evaluate the reward function selection *efficiency* of ORSO compared to more naive selection strategies.

**No Design**  *(Task Reward with No Shaping)* We train the agent with the task reward function $r$ for each MDP. These reward functions can be sparse (for manipulation) or unshaped (for locomotion). We use the same reward definitions as prior work (Ma et al., 2024), which we report in Appendix E.

**Human**  We consider the human-engineered reward functions for each task provided by (Makoviychuk et al., 2021). We note that these are constructed such that training PPO with the given hyperparameters yields a performant policy with respect to the task reward function. The function definitions are reported in Appendix E.

**Naive Selection**  A naive selection strategy involves sampling a set of reward functions and training policies on each reward to convergence. While this approach is simple and widely used (Ma et al., 2024), it can be inefficient because it uniformly explores each reward function for a fixed number of iterations, regardless of the task performance.

### 5.1.2 IMPLEMENTATION

**Reward Generation**  Similarly to recent works on reward design, which demonstrate that LLMs can generate effective reward functions for training agents (Park et al., 2024; Ma et al., 2024; Xie et al., 2024; Yu et al., 2024), we follow this paradigm by using GPT-4 (Achiam et al., 2023) to avoid manually designing reward function components. The language model is prompted to generate reward function code in Python based on some minimal environment code describing the observation space and useful class variables. We employ prompts similar to those used by Ma et al. (2024). Since the exact prompts are not the primary focus of our work, we do not detail them here; instead, we refer readers to our codebase for further details on the prompt construction.

While the LLM produces seemingly good code, this does not guarantee that the sampled code is bug-free and runnable. In ORSO, we employ a simple rejection sampling technique to construct sets of only valid reward functions with high probability. We also note that the initial set of generated reward functions in ORSO might not contain an effective reward function.[3] To address this limitation, we introduce a mechanism for improving the reward function set through iterative resampling and in-context evolution of new sets $\mathcal{R}^K$. We provide more details on the rejection sampling mechanism and the iterative refinement process Appendix F.

**Online Reward Selection Algorithms**  We evaluate multiple reward selection algorithms from the multi-armed bandit and online model selection literature: explore-then-commit (ETC), $\varepsilon$-greedy (EG), upper confidence bound (UCB) (Auer, 2002), exponential-weight algorithm for exploration

---

[3]An effective reward function is one that leads to high performance with respect to the task reward $r$ when used for training.

and exploitation (Exp3) (Auer et al., 2002), and doubling data-driven regret balancing ($D^3RB$) (Dann et al., 2024). We provide the pseudocode and the hyperparameters used for each selection algorithm in Appendix G. For every environment, we set the number of iterations $N$ in Algorithm 1 used to train the policy before we select a different reward function to $N = \texttt{n\_iters}/100$, where $\texttt{n\_iters}$ is the number of iterations used to train the baselines, i.e., we perform at least 100 iterations of online reward selection before the iterative resampling.

## 5.2 RESULTS

In this section, we present the experimental results of ORSO. We evaluate ORSO's ability to efficiently select reward functions with varying budget constraints and reward function set size $K$. We consider interaction budgets $B \in \{5, 10, 15\} \times \texttt{n\_iters}$ and sample sizes $K \in \{4, 8, 16\}$.

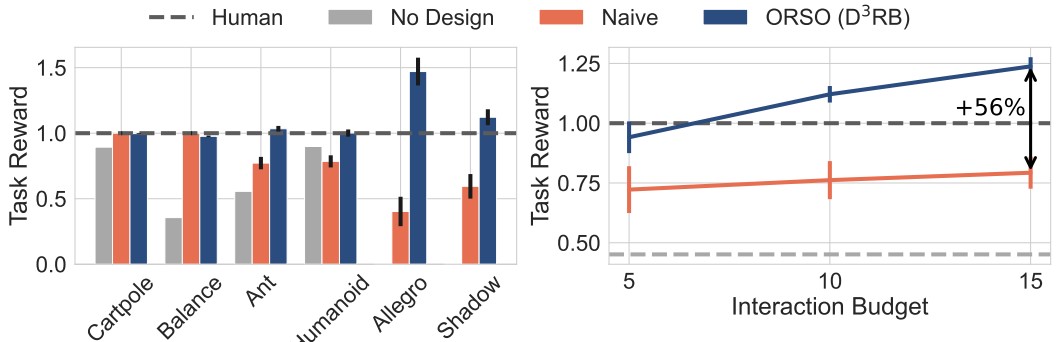

Figure 2: **Left**: Normalized task rewards averaged over interaction budgets and seeds. ORSO consistently matches or surpasses human-designed reward functions. **Right**: Normalized task reward as a function of interaction budget, averaged across tasks. ORSO scales effectively with increased budgets, achieving a 56% higher task reward than the naive strategy at the highest budget. Vertical bars in the plots indicate standard errors.

**ORSO Surpasses Human-Designed Reward Functions**   Figure 2 (left) illustrates the average performance of ORSO compared to human-designed reward functions, the task reward function, and the naive selection strategy across different tasks. We observe that ORSO consistently matches or exceeds human-designed rewards, particularly in more complex environments. For each task, the results are averaged over 3 interaction budgets, 3 reward function set sizes and with 3 seed per configuration, totaling 27 runs. The full breakdown is reported in Appendix H.

**ORSO Achieves 56% Higher Task Reward**   Figure 2 (right) shows how task performance scales with the interaction budget. While both ORSO and naive selection benefit from larger budgets, ORSO is consistently superior and surpasses human-designed rewards when $B \geq 10$. Moreover, ORSO achieves 56% higher reward than the naive strategy at the highest budget level. This demonstrates that ORSO can more effectively make use of the additional interactions with the environment, allocating more compute to better reward functions. Detailed per-task and per-budget results are reported in Appendix H.

**ORSO Can Reach Human Performance with Fewer GPUs**   One advantage of the naive selection strategy is that it can be easily parallelized on many GPUs. Figure 3 reports the estimated time required to achieve the same performance as policies trained with human-designed reward functions as a function of the number of GPUs used. Notably, ORSO performs at a comparable speed to the naive selection strategy even when the latter leverages up to 8 GPUs in parallel, achieving similar performance within the same timeframe. It should be noted that the plotted time is an approximation based on the time needed to complete one iteration of PPO for each task, where one iteration consists of collecting a batch of trajectories, updating the policy, and the value function. We report the results for all interaction budgets in the Appendix H.

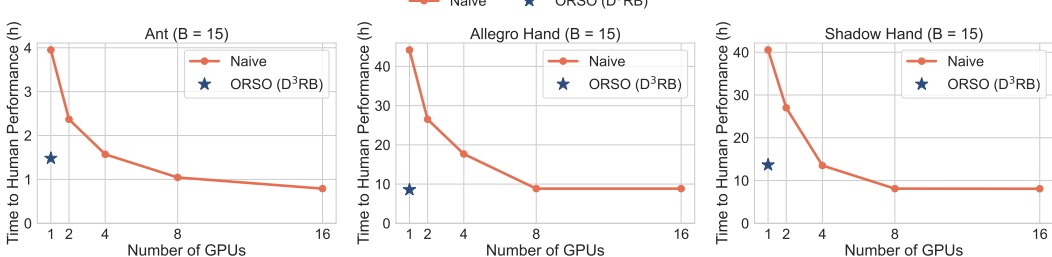

Figure 3: Median time to human-level performance as a function of number of parallel GPUs. Policies trained with ORSO can achieve the same performance as policies trained with the human-engineered reward functions with up to $8\times$ fewer GPUs.

## 5.3 ABLATION STUDY

**Choice of Selection Algorithm**  In Figure 4, we compare different selection algorithms for ORSO. We find that $D^3RB$ performs best on average, consistently outperforming other algorithms, followed closely by Exp3. These algorithms allow ORSO to balance exploration and exploitation effectively, leading to superior performance compared to more greedy approaches like UCB, ETC, and EG. Interestingly, even simpler strategies like EG and ETC substantially outperform the naive strategy, which highlights the importance of properly balancing exploration and exploitation for efficient reward selection. By framing reward design as an exploration-exploitation problem, we demonstrate that even basic strategies offer considerable gains over static, inefficient methods.

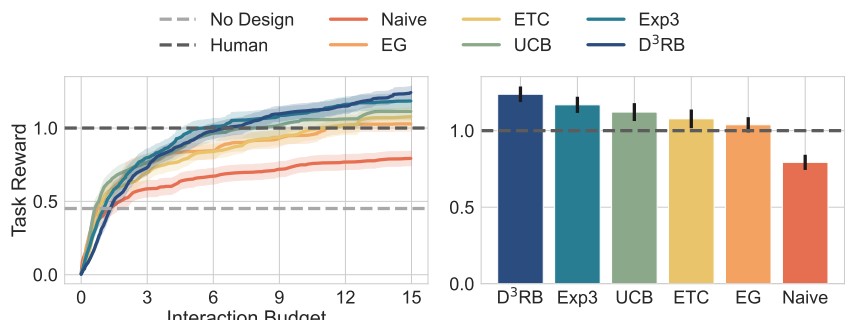

Figure 4: Comparison of different rewards selection algorithms for ORSO. **Left**: Number of iterations necessary for human-level performance. **Right**: Average normalized task reward for different selection algorithms. We provide a more granular breakdown in Appendix H.

**Regret of Different Selection Algorithms**  To further quantify ORSO's performance, we analyze its regret with respect to human-engineered reward functions.[4] This formulation is motivated by two key considerations. First, we lack access to the true optimal policy with respect to the task reward function $\pi^\star$. Second, the PPO hyperparameters used in our experiments were specifically tuned for the human-engineered reward function, making the policy trained with it a reasonable proxy for the optimal policy. Regret provides a useful metric for understanding how much performance is lost due to suboptimal reward selection over time. Lower regret indicates that a method quickly identifies high-quality reward functions, reducing the number of iterations wasted on poorly performing ones. Figure 5 shows the normalized regret for different selection algorithms. Notably, ORSO's regret can become negative, indicating that it finds reward functions that outperform the human-designed ones.

---

[4]The normalized regret with respect to the human-engineered reward functions is defined as $\frac{\texttt{Human}-\mathcal{J}(\pi_t^\star)}{\texttt{Human}}$.

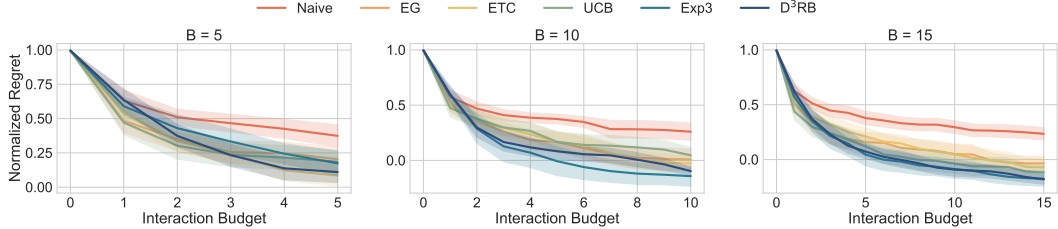

Figure 5: Regret of different selection algorithms with varying budgets. We recall that a budget $B$ indicates that the ORSO has been run for $B \times$ n_iters iterations.

**ORSO is Effective with Large Reward Sets**  Previous experiments considered at most 16 reward functions at once, raising the question of whether ORSO remains effective when the candidate set is significantly larger. A larger set could pose challenges: excessive exploration might leave insufficient time for learning, while premature commitment could lead to suboptimal performance. To investigate this, we evaluate ORSO with different selection algorithms on the ANT task, using a interaction budget of $B = 15$ and reward sets of sizes $K \in \{48, 96\}$.

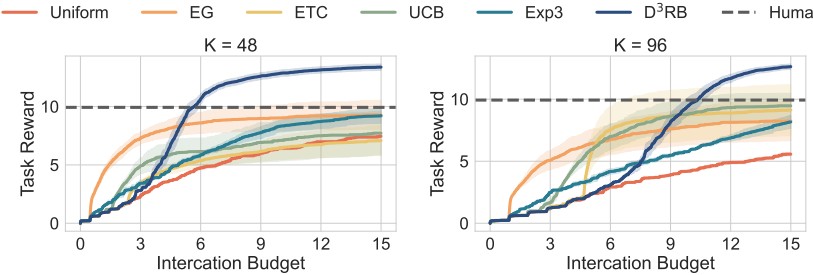

Figure 6: Comparison of multiple selection algorithms for the ANT task with a high number of reward function candidates. The shaded areas represent standard errors over 5 different seeds. The order of the reward functions is randomized for each seed.

In this setting – with a fixed budget and reward function set[5] – algorithms that commit to a selection earlier can allocate more iterations to training on the chosen reward functions. On the other hand, exploring for longer may allow us to find the optimal reward function but potentially leave insufficient time for training.

As illustrated in Figure 6, D[3]RB consistently identifies and selects an effective reward function from the set. In contrast, "greedier" methods such as $\varepsilon$-greedy, explore-then-commit, and UCB can depend more on the stochasticity of training and on average do not surpass human-designed reward functions. Exp3 and uniform exploration, while more exploratory, may overemphasize exploration at the expense of exploiting promising reward functions, leading to suboptimal performance. We report the task reward of each reward function in Table 3 to validate that ORSO with D[3]RB truly selects the best reward functions.

## 6  RELATED WORK

Traditionally, researchers manually specified reward components and tuned their coefficients (Ng et al., 1999; Margolis & Agrawal, 2022; Liu et al., 2024), a method that often requires significant domain expertise and involves numerous iterations of trial and error.

Recent work has increasingly explored the potential of foundation models in reward design. Approaches like L2R (Yu et al., 2023) leverage large language models to generate reward functions by

---

[5]In this experiment, we do not perform iterative improvement. That is, the set of reward functions is fixed over the entire training.

converting natural language descriptions into code using predefined reward API primitives, though this requires notable effort in manual template design. Other works such as EUREKA (Ma et al., 2024) and Text2Reward (Xie et al., 2024) use language models to generate dense reward functions based on task descriptions and environment codes.

Foundation models have also been directly employed as reward models. Researchers have used cosine similarity of CLIP embeddings (Rocamonde et al., 2024), vision language models for trajectory preference labeling (Wang et al., 2024), and large language models for constructing preference datasets and intrinsic reward modeling (Klissarov et al., 2024; Kwon et al., 2023).

Non-stationary scenarios have been extensively explored in the bandit literature, with the restless bandit model (Whittle, 1988; Weber & Weiss, 1990) being a prominent example. In this model, each arm evolves according to a potentially unknown Markov decision process (MDP). Various solution approaches have been developed for this setting, including those leveraging the Whittle Index (Gittins et al., 2011). However, the restless bandit framework does not fully capture the problem considered in this work. Unlike a setting where each base learner corresponds to a fixed MDP, the problem of reward selection involves base learners that when chosen advance their internal state never to revisit it. This forms the basis of the online model selection literature that has addressed the challenge of dynamically choosing suitable models in sequential decision-making environments (Agarwal et al., 2017; Foster et al., 2019; Pacchiano et al., 2020; Lee et al., 2021).

A more comprehensive review of related work is provided in Appendix A.

# 7 CONCLUSION

In this paper, we introduce ORSO, a novel approach for reward design in reinforcement learning that significantly accelerates the design of shaped reward functions. We find that even simple strategies like $\varepsilon$-greedy and explore-then-commit yield substantial improvements over naive selection, suggesting that reward design can be effectively framed as a sequential decision problem. ORSO reduces both time and computational costs by more than half compared to earlier methods, making reward design accessible to a wider range of researchers. What once required a larger amount of computational resources can not be done on a single desktop in a reasonable time. By formalizing the reward design problem and providing a theoretical analysis of ORSO's regret when using the D³RB algorithm, we also contribute to the theoretical understanding of reward design in RL.

Looking ahead, our work opens several promising directions for future research, including the development of more sophisticated exploration strategies tailored for reward design, and the application of our approach to more complex, real-world RL problems.

## 7.1 LIMITATIONS AND FUTURE WORK

Our experiments explored up to 16 reward functions with resampling of the reward function set and up to 96 without resampling. We find that using 8 reward functions with as much interaction budget as possible yields the best results. We leave the study of even larger reward function sets for future work.

A key limitation of ORSO is its reliance on a predefined task reward, which is typically straightforward to construct for simpler tasks but can be challenging for more complex ones or for tasks that include a qualitative element to them, e.g., making a quadruped walk with a "nice" gait. Future work could explore eliminating the need for such hand-crafted task rewards by leveraging techniques that translate natural language instructions directly into evaluators, potentially using vision-language models, similarly to Wang et al. (2024); Rocamonde et al. (2024). Another alternative is to use preference data to learn a task reward model (Christiano et al., 2017; Zhang & Ramponi, 2023) and use the latter as a signal for the model selection algorithm.

Finally, in principle, ORSO could be run on multiple GPUs in parallel, with results from different runs aggregated for improved efficiency. Investigating parallelization strategies and their impact on reward selection remains an interesting direction for future study.

## ACKNOWLEDGEMENTS

We thank members of the Improbable AI Lab for helpful discussions and feedback. We are grateful to MIT Supercloud and the Lincoln Laboratory Supercomputing Center for providing HPC resources. This research was supported in part by Hyundai Motor Company, Quanta Computer Inc., an AWS MLRA research grant, ARO MURI under Grant Number W911NF-23-1-0277, DARPA Machine Common Sense Program, ARO MURI under Grant Number W911NF-21-1-0328, and ONR MURI under Grant Number N00014-22-1-2740. The views and conclusions contained in this document are those of the authors and should not be interpreted as representing the official policies, either expressed or implied, of the Army Research Office or the United States Air Force or the U.S. Government. The U.S. Government is authorized to reproduce and distribute reprints for Government purposes, notwithstanding any copyright notation herein.

## AUTHOR CONTRIBUTIONS

- **Chen Bo Calvin Zhang:** Led the project, wrote the manuscript, ideated the method, implemented the algorithm, and conducted the experiments.
- **Zhang-Wei Hong:** Provided guidance on the types of experiments to run and assisted with the implementation of saving and loading the simulator state in Isaac Gym. Drafted the introduction and advised on writing.
- **Aldo Pacchiano:** Contributed to the theoretical aspects of the work, specifically the model selection part, including the development and proof of key concepts.
- **Pulkit Agrawal:** Oversaw the project, assisted with positioning the paper, and contributed to the writing and presentation of the results.

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

# A RELATED WORK

**Reward Design for RL**   Designing effective reward functions for reinforcement learning has been a long-standing challenge. Several approaches have been proposed to tackle it.

Traditionally, researchers manually specify reward components and tune their coefficients (Ng et al., 1999; Margolis & Agrawal, 2022; Liu et al., 2024). This method often demands significant domain expertise and can be highly resource-intensive, involving numerous iterations of trial and error in designing reward functions, training policies, and adjusting reward parameters.

Another approach is to learn reward functions from expert demonstrations via methods like apprenticeship learning (Abbeel & Ng, 2004) and maximum entropy inverse RL (Ziebart et al., 2008). While these methods can capture complex behaviors, they often rely on high-quality demonstrations and may struggle in environments where such data is scarce or noisy.

Preferences can also be used to learn reward functions (Zhang & Ramponi, 2023; Christiano et al., 2017). This approach involves collecting feedback in the form of preferences between different trajectories, which are then used to infer a reward function that aligns with the desired behavior. This method is particularly useful in scenarios where it is difficult to explicitly define a reward function or obtain expert demonstrations, as it allows for more intuitive and accessible feedback from users.

**Foundation Models and Reward Functions**   Recent work has explored the use of large language/vision models (LL/VMs) to aid in the reward design process. L2R (Yu et al., 2023) leverages large language models to generate reward functions for RL tasks by first creating a natural language "motion description" and then converting it into code using predefined reward API primitives. While innovative, L2R has notable limitations: it requires significant manual effort in designing templates and primitives and is constrained by the latter. EUREKA (Ma et al., 2024) and Text2Reward (Xie et al., 2024) use LLMs to generate dense reward functions for RL given the task description in natural language and the environment code.

Foundation models have also been directly used as reward models. Rocamonde et al. (2024) uses the cosine similarity of CLIP embeddings of language instructions and renderings of the state as a state-only reward model. Similarly, Wang et al. (2024) automatically generates reward functions for RL using a vision language model to label pairs of trajectories with preference, given a task description. Motif (Klissarov et al., 2024) first constructs a pair-wise preferences dataset using a large language model (LLM), learns a preference-based intrinsic reward model with the Bradley-Terry (Bradley & Terry, 1952) model, and then uses this reward model to train a reinforcement learning agent. Kwon et al. (2023) uses a similar approach, where an LLM is used during training to evaluate an RL policy, given a few examples of successful behavior or a description of the desired behavior.

**Online Model Selection**   The problem of model selection in sequential decision-making environments has gained significant attention in recent years (Agarwal et al., 2017; Foster et al., 2019; Pacchiano et al., 2020; Lee et al., 2021). This area of research addresses the challenge of dynamically choosing the most suitable model or algorithm from a set of candidates while learning.

Agarwal et al. (2017) introduced CORRAL, a method to combine multiple bandit algorithms in a master algorithm. Foster et al. (2019) proposed model selection guarantees for linear contextual bandits. Pacchiano et al. (2020) extend the CORRAL algorithm and propose Stochastic CORRAL. Lastly, Lee et al. (2021) propose Explore-Commit-Eliminate (ECE), an algorithm for model selection in RL with function approximation. A common requirement across all these approaches is the need to know the regret guarantees of the base algorithms.

Our work is closely related to Dann et al. (2024), which removes the need for known regret guarantees and instead uses *realized* regret bounds for the base learners. In our setting, the set of models comprises the reward functions set and their corresponding policies.

# B   ONLINE MODEL SELECTION

In this section, we introduce the model selection problem and some necessary notation modified from Dann et al. (2024) for our analysis.

We consider a general sequential decision-making process consisting of a *meta learner* interacting with an environment over $T \in \mathbb{N}$ rounds via a set of *base learners*. At each round of interaction $t = 1, 2, \ldots, T$, the meta learner selects a base learner $b_t$ and after executing $b_t$, the environment returns a model selection reward $R_t \in \mathbb{R}$. The objective of the meta learner is to sequentially choose base learners $b_1, \ldots, b_T$ to maximize the expected cumulative sum of model selection rewards, i.e., $\max \mathbb{E}\left[\sum_{t=1}^{T} R_t\right]$. We denote by $v^b = \mathbb{E}[R \mid b]$ the expected model selection reward, given that the learner chooses base learner $b$, i.e., the value of base learner $b$. The total model selection reward accumulated by the algorithm over $T$ rounds is denoted by $u_T = \sum_{t=1}^{T} v^{b_t}$. The objective is to minimize the cumulative regret after $T$ rounds of interaction,

$$\mathrm{Reg}(T) := \sum_{t=1}^{T} \mathrm{reg}(b_t) = \sum_{t=1}^{T} v^\star - v^{b_t}, \tag{4}$$

where $v^\star$ is the value of the optimal base learner.

In our setting, each base learner corresponds to a reward function $f$ and its associated policy $\pi$, i.e., $b = (f, \pi)$. In this case, choosing to execute base learner $b$ means training with algorithm $\mathfrak{A}$ starting from checkpoint $\pi$ and using RL reward function $f$. The model selection reward $R$ is then the evaluation of the trained policy under the task reward $r$, i.e., $\mathcal{J}(\pi)$. The regret of base learner $b$ can therefore be written as $\mathrm{reg}(b) = v^\star - v^b = \mathcal{J}(\pi^\star) - \mathcal{J}(\pi)$, where $\pi^\star$ is the optimal policy. Therefore the objective becomes minimizing

$$\mathrm{Reg}(T) = \sum_{t=1}^{T} \mathcal{J}(\pi^\star) - \mathcal{J}(\pi_t). \tag{5}$$

**Notation**   The policy associated with base learner $i$ at round $t$ is denoted by $\pi_t^i$, so that $\pi_t = \pi_t^{i_t}$. We denote the number base learner $i$ has been played up to round $t$ as $n_t^i = \sum_{\ell=1}^{t} \mathbb{1}\{i_\ell = i\}$ and the total cumulative reward for learner $i$ as $u_t^i = \sum_{\ell=1}^{t} \mathbb{1}\{i_\ell = i\} v^{\pi_\ell^i}$, where we use $v^{\pi_\ell^i} = v^{b_\ell^i}$ to highlight that the policy associated with base learner $i$ changes over time, but the reward function used for RL does not. We denote the internal clock for each base learner with a subscript $(k)$ such that $\pi_{(k)}^i$ is the policy of learner $i$ when chosen for the $k$-th time, i.e., $\pi_t^i = \pi_{(n_t^i)}^i$.

We note an important difference between the online model selection problem and the multi-armed bandit (MAB) problem. In model selection, the meta learner interacts with an environment over $T$ rounds, selecting from $K$ base learners. In each round $t$, the meta learner picks a base learner $i_t \in [K]$ (index of base learner chosen at step $t$) and follows its policy, updating the base learner's state with new data. Unlike MAB problems, where mean rewards are stationary, the mean rewards here are non-stationary due to the stateful nature of base learners (the base learners are learning as they see more data), making the design of effective model selection algorithms challenging.

**Remark 1.** *The cumulative regret in Equation* (5) *is an upper bound for the cumulative regret with respect to the best so far.*

*Proof.* This is straightforward to see. Let us first note that, by definition, for all $t \in [T]$, we have

$$\mathcal{J}(\pi_t^\star) \geq \mathcal{J}(\pi_t). \tag{6}$$

Therefore,

$$\sum_{t=1}^{T} \mathcal{J}(\pi^\star) - \mathcal{J}(\pi_t^\star) \leq \sum_{t=1}^{T} \mathcal{J}(\pi^\star) - \mathcal{J}(\pi_t), \tag{7}$$

concluding the proof.   $\square$

## C  ORSO WITH DOUBLING DATA-DRIVEN REGRET BALANCING

Here, we present the complete ORSO algorithm with Doubling Data-Driven Regret Balancing (D$^3$RB) as the model selection algorithm.

D$^3$RB is built upon the idea of *regret balancing*, which aims to optimize the performance of multiple models by balancing their respective regrets. Imagine weighing two models on a balance scale where the "weight" corresponds to their regret; the goal is to keep the regret of both models balanced. This approach ensures that models with higher regret rates are selected less frequently, while those with lower regret rates are favored.

Concretely, regret balancing involves associating each learner with a candidate regret bound. The model selection algorithm then competes against the regret bound of the best-performing learner among those that are well-specified – meaning their realized regret stays within their candidate bounds. Traditional approaches often rely on known *expected* regret bounds. In contrast, D$^3$RB focuses on *realized* regret, allowing the model selection algorithm to compete based on the actual regret outcomes of each base learner. The algorithm dynamically adjusts the regret bounds in a data-driven manner, adapting to the realized regret of the best-performing learner over time. This approach overcomes the limitation of needing known regret bounds, which are often unavailable for complex problems.

D$^3$RB maintains three estimators for each base learner: regret coefficients $\widehat{d}_t^i$, average rewards $\widehat{u}_t^i/n_t^i$ and balancing potentials $\phi_t^i$. At each step $t$, D$^3$RB selects the base learner with the lower balancing potential and executes it. Then it performs the misspecification test in Equation (8) to check if the estimated regret coefficient for base learner $i_t$ is consistent with the observed data. If the test triggers, i.e., the $\widehat{d}^{i_t}$ is too small, then the algorithm doubles it. Lastly, D$^3$RB sets the balancing potential $\phi_t^i$ to $\widehat{d}_t^{i_t}\sqrt{n_t^{i_t}}$.

$$\frac{\widehat{u}_t^{i_t}}{n_t^{i_t}} + \frac{\widehat{d}_{t-1}^{i_t}\sqrt{n_t^{i_t}}}{n_t^{i_t}} + c\sqrt{\frac{\ln\frac{K\ln n_t^{i_t}}{\delta}}{n_t^{i_t}}} < \max_{j\in[K]}\frac{\widehat{u}_t^j}{n_t^j} - c\sqrt{\frac{\ln\frac{K\ln n_t^j}{\delta}}{n_t^j}} \tag{8}$$

---

**Algorithm 2** ORSO with D$^3$RB

---

**Require:** MDP $\mathcal{M} = (\mathcal{S}, \mathcal{A}, P, r, \gamma, \rho_0)$, algorithm $\mathfrak{A}$, generator $G$, minimum regret coefficients $d_{\min}$, failure probability $\delta$

1: Sample $K$ reward functions $\mathcal{R}^K = \left\{ f^1, \ldots, f^K \right\} \sim G$
2: Initialize $K$ policies $\left\{ \pi^1, \ldots, \pi^K \right\}$
3: Initialize balancing potentials $\phi_1^i = d_{\min}$ for all $i \in [K]$
4: Initialize regret coefficients $\widehat{d}_0^i = d_{\min}$ for add $i \in [K]$
5: Initialize counts $n_0^i = 0$ and total values $\widehat{u}_0^i = 0$ for all $i \in [K]$
6: **for** $t = 1, 2, \ldots, T$ **do**
7:     Select a base learner $i_t \in [K] \in \arg\min_{i \in [K]} \phi_t^i$
8:     Update $\pi^{i_t} \leftarrow \mathfrak{A}_{f^{i_t}}(\mathcal{M}_{i_t}, \pi^{i_t})$
9:     Evaluate $R_t = \mathcal{J}\left(\pi^{i_t}\right) \leftarrow \texttt{Eval}(\pi^{i_t})$
10:     // Update necessary variables
11:     Set $n_t^i = n_{t-1}^i, \widehat{u}_t^i = \widehat{u}_{t-1}^i, \widehat{d}_t^i = \widehat{d}_{t-1}^i$, and $\phi_{t+1}^i = \phi_t^i$ for all $i \in [K] \setminus \{i_t\}$
12:     Update statistics for current learner $n_t^{i_t} = n_{t-1}^{i_t} + 1$ and $\widehat{u}_t^{i_t} = \widehat{u}_{t-1}^{i_t} + R_t$
13:     Perform misspecification test

$$\frac{\widehat{u}_t^{i_t}}{n_t^{i_t}} + \frac{\widehat{d}_{t-1}^{i_t} \sqrt{n_t^{i_t}}}{n_t^{i_t}} + c\sqrt{\frac{\ln \frac{K \ln n_t^{i_t}}{\delta}}{n_t^{i_t}}} < \max_{j \in [K]} \frac{\widehat{u}_t^j}{n_t^j} - c\sqrt{\frac{\ln \frac{K \ln n_t^j}{\delta}}{n_t^j}} \qquad (9)$$

14:     **if** test is triggered **then**
15:         Double the regret coefficient $\widehat{d}_t^{i_1} \leftarrow 2\widehat{d}_{t-1}^{i_t}$
16:     **else**
17:         Keep the regret coefficient unchanged $\widehat{d}_t^{i_1} \leftarrow \widehat{d}_{t-1}^{i_t}$
18:     **end if**
19:     Update the balancing potential $\phi_{t+1}^{i_t} \leftarrow \widehat{d}_t^{i_t} \sqrt{n_t^{i_t}}$
20: **end for**
21: // Best policy and reward function under the task reward
22: **return** $\pi_T^\star, f_T^\star = \arg\max_{i \in [K]} \mathcal{J}(\pi^i)$

---

## D    PROOF OF LEMMA 4.4

In this section, we present the complete proof of Lemma 4.4. We will start by showing that when Assumption 4.2 holds, then with probability at least $1 - \delta$, the estimated regret coefficient of learner $i_\star$ will never double provided that $d_{\min} \geq c$, where $c$ is the confidence multiplier in D$^3$RB.

**Lemma D.1** (Non-doubling regret coefficient). *When $\mathcal{E}$ holds, and algorithm D$^3$RB is in use*

$$\widehat{\widetilde{d}}_t^{i_\star} = d_{\min} \quad and \quad n_T^i \leq n_T^{i_\star} + 1 \quad for \ all \ i \in [K] \tag{10}$$

*for all $t \in \mathbb{N}$.*

*Proof.* In order to show this result it is sufficient to show that when $\mathcal{E}$ holds, algorithm $i_\star$ does not undergo any doubling event. Doubling of the regret coefficients only happens when the misspecification test triggers for algorithm $i_\star$.

We will show this by induction.

**Base Case ($t = 1$)**   At $t = 1$, for all algorithms $i \in [K]$:

- $\widehat{d}_1^i = d_{\min}$ (by initialization)

- $n_1^i = 1$ if $i$ is the first algorithm chosen, 0 otherwise

Therefore $n_1^i \leq n_1^{i_\star} + 1$ holds

**Inductive Step**   Inductive hypothesis: assume that for some $t \geq 1$:

- $\widehat{\widetilde{d}}_{t-1}^{i_\star} = d_{\min}$

- $n_{t-1}^i \leq n_{t-1}^{i_\star} + 1$ for all $i \in [K]$

We need to show these properties hold for $t$. Let $i_t = i_\star$. When $\mathcal{E}$ holds, the left-hand side (LHS) of D$^3$RB's misspecification test satisfies

$$\frac{\widehat{u}_t^{i_t}}{n_t^{i_t}} + \frac{\widehat{\widetilde{d}}_{t-1}^{i_t}\sqrt{n_t^{i_t}}}{n_t^{i_t}} + c\sqrt{\frac{\ln \frac{K \ln n_t^{i_t}}{\delta}}{n_t^{i_t}}} = \frac{\widehat{u}_t^{i_\star}}{n_t^{i_\star}} + \frac{\widehat{\widetilde{d}}_{t-1}^{i_\star}\sqrt{n_t^{i_\star}}}{n_t^{i_\star}} + c\sqrt{\frac{\ln \frac{K \ln n_t^{i_\star}}{\delta}}{n_t^{i_\star}}} \quad (i_t = i_\star)$$

$$\geq \frac{u_t^{i_\star}}{n_t^{i_\star}} + \frac{\widehat{\widetilde{d}}_{t-1}^{i_\star}\sqrt{n_t^{i_\star}}}{n_t^{i_\star}} \quad (\text{event } \mathcal{E})$$

$$\overset{(i)}{=} \frac{u_t^{i_\star}}{n_t^{i_\star}} + \frac{d_{\min}\sqrt{n_t^{i_\star}}}{n_t^{i_\star}} \tag{11}$$

where $(i)$ holds because by the induction hypothesis $\widehat{\widetilde{d}}_{t-1}^{i_\star} = d_{\min}$. We will now show that $n_t^{i_\star} \geq n_t^j$ for all $j \in [K]$. Since by the inductive hypothesis $\widehat{\widetilde{d}}_\ell^{i_\star} = d_{\min}$ for all $\ell \leq t - 1$, the potential $\phi_\ell^{i_\star} = d_{\min}\sqrt{n_{\ell-1}^{i_\star}}$ for all $\ell \leq t$.

For $i \in [K]$ let $t(i)$, be the last time – before time $t$ – algorithm $i$ was played. For $i \neq i_\star$ we have $t(i) < t$. Since $i$ was selected at time $t(i)$, by definition of the potentials,

$$\widehat{\widetilde{d}}_{t(i)-1}^{i_\star}\sqrt{n_{t(i)-1}^{i_\star}} = d_{\min}\sqrt{n_{t(i)-1}^{i_\star}} \geq \widehat{\widetilde{d}}_{t(i)-1}^{i}\sqrt{n_{t(i)-1}^{i}} \geq d_{\min}\sqrt{n_{t(i)-1}^{i}}$$

so that $n_{t(i)-1}^{i_\star} \geq n_{t(i)-1}^{i}$. Since both $n_t^{i_\star} = n_{t(i)-1}^{i_\star} + 1$ and $n_t^i = n_{t(i)-1}^i + 1$ we conclude that $n_t^{i_\star} \geq n_t^i$.

We now turn our attention to the right-hand side (RHS) of D³RB's misspecification test. When $\mathcal{E}$ holds, the RHS of D³RB's misspecification test satisfies,

$$
\max_{j \in [K]} \frac{\widehat{u}_t^j}{n_t^j} - c \sqrt{\frac{\ln \frac{K \ln n_t^j}{\delta}}{n_t^j}} \leq \max_{j \in [K]} \frac{u_t^j}{n_t^j}
$$

$$
\overset{(i)}{\leq} \max_{j \in [K]} \frac{u_{(n_t^j)}^{i_\star}}{n_t^j}
$$

$$
\overset{(ii)}{\leq} \frac{u_t^{i_\star}}{n_t^{i_\star}} \tag{12}
$$

where inequalities $(i)$ and $(ii)$ hold because of Assumption 4.2. Combining inequalities 11 and 12 we conclude the misspecification test of algorithm D³RB will not trigger. Thus, $\widehat{d}_t^{i_\star}$ remains at $d_{\min}$ and for all $i \in [K]$, $n_t^i \leq n_t^{i_\star} + 1$ continues to hold. This finalizes the proof. $\qquad\square$

We are now ready to prove the regret bound on the base learners given in Lemma 4.4.

**Lemma 4.4.** *Under event $\mathcal{E}$ and Assumption 4.2, with probability $1 - \delta$, the regret of all learners $i$ is bounded in all rounds $T$ as*

$$
\sum_{t=1}^{n_T^i} \operatorname{reg}(\pi_{(t)}^i) \leq 6 d_T^{i_\star} \sqrt{n_T^{i_\star} + 1} + 5c \sqrt{(n_T^{i_\star} + 1) \ln \frac{K \ln T}{\delta}}, \tag{3}
$$

*where $d_T^{i_\star} = d_{(n_T^{i_\star})}^{i_\star}$.*

*Proof.* Consider a fixed base learner $i$ and time horizon $T$, and let $t \leq T$ be the last round where $i$ was played but the misspecification test did not trigger. If no such round exists, then set $t = 0$. By Corollary 9.1 in Dann et al. (2024), $i$ can be played at most $1 + \log_2 \frac{\bar{d}_T^i}{d_{\min}}$ times between $t$ and $T$, where $\bar{d}_T^i = \max_{\ell \leq T} d_\ell^i$. Thus,

$$
\sum_{k=1}^{n_T^i} \operatorname{reg}\left(\pi_{(k)}^i\right) \leq \sum_{k=1}^{n_t^i} \operatorname{reg}\left(\pi_{(k)}^i\right) + 1 + \log_2 \frac{\bar{d}_T^i}{d_{\min}}.
$$

If $t = 0$, then the desired statement holds. Thus, it remains to bound the first term in the RHS above when $t > 0$. Since $i = i_t$ and the test did not trigger we have, for any base learner $j$ with $n_t^j > 0$,

$$
\sum_{k=1}^{n_t^i} \operatorname{reg}\left(\pi_{(k)}^i\right) = n_t^i v^\star - u_t^i \qquad\qquad \text{(definition of regret)}
$$

$$
= n_t^i v^\star - \frac{n_t^i}{n_t^j} u_t^j + \frac{n_t^i}{n_t^j} u_t^j - u_t^i
$$

$$
= \frac{n_t^i}{n_t^j}\left(n_t^j v^\star - u_t^j\right) + \frac{n_t^i}{n_t^j} u_t^j - u_t^i
$$

$$
= \frac{n_t^i}{n_t^j}\left(\sum_{k=1}^{n_t^j} \operatorname{reg}\left(\pi_{(k)}^j\right)\right) + \frac{n_t^i}{n_t^j} u_t^j - u_t^i \qquad\qquad \text{(definition of regret)}
$$

$$
\leq \frac{n_t^i}{n_t^j}\left(d_t^j \sqrt{n_t^j}\right) + \frac{n_t^i}{n_t^j} u_t^j - u_t^i \qquad\qquad \text{(definition of regret rate)}
$$

$$
= \sqrt{\frac{n_t^i}{n_t^j}} d_t^j \sqrt{n_t^i} + \frac{n_t^i}{n_t^j} u_t^j - u_t^i.
$$

We now focus on $j = i_\star$ and use the balancing condition in Lemma 9.2 in Dann et al. (2024) to bound the first factor $\sqrt{n_t^i / n_t^{i_\star}}$. This condition gives that $\phi_{t+1}^i \leq 3\phi_{t+1}^{i_\star}$. Since both $n_t^{i_\star} > 0$ and $n_t^i > 0$, we have $\phi_{t+1}^i = \widehat{d}_t^i \sqrt{n_t^i}$ and $\phi_{t+1}^{i_\star} = \widehat{d}_t^{i_\star} \sqrt{n_t^{i_\star}}$. Thus, we get

$$\sqrt{\frac{n_t^i}{n_t^{i_\star}}} = \sqrt{\frac{n_t^i}{n_t^{i_\star}} \cdot \frac{\widehat{d}_t^i}{\widehat{d}_t^{i_\star}} \cdot \frac{\widehat{d}_t^{i_\star}}{\widehat{d}_t^i}} = \frac{\phi_{t+1}^i}{\phi_{t+1}^{i_\star}} \cdot \frac{\widehat{d}_t^{i_\star}}{\widehat{d}_t^i} \leq 3\frac{\widehat{d}_t^{i_\star}}{\widehat{d}_t^i} \leq 3, \tag{13}$$

where the last inequality holds because of Lemma D.1 and because $\widehat{d}_t^i \geq d_{\min}$.

Plugging this back into the expression above and setting $j = i_\star$, we have

$$\sum_{k=1}^{n_t^i} \mathrm{reg}\left(\pi_{(k)}^i\right) \leq 3d_t^{i_\star} \sqrt{n_t^i} + \frac{n_t^i}{n_t^{i_\star}} u_t^{i_\star} - u_t^i.$$

To bound the last two terms, we use the fact that the misspecification test did not trigger in round $t$. Therefore,

$$u_t^i \geq \widehat{u}_t^i - c\sqrt{n_t^i \ln \frac{K \ln n_t^i}{\delta}} \qquad (\text{event } \mathcal{E})$$

$$= n_t^i \left( \frac{\widehat{u}_t^i}{n_t^i} + c\sqrt{\frac{\ln \frac{K \ln n_t^i}{\delta}}{n_t^i}} + \frac{\widehat{d}_t^i \sqrt{n_t^i}}{n_t^i} \right) - 2c\sqrt{n_t^i \ln \frac{K \ln n_t^i}{\delta}} - \widehat{d}_t^i \sqrt{n_t^i}$$

$$\geq \frac{n_t^i}{n_t^{i_\star}} \widehat{u}_t^{i_\star} - \sqrt{\frac{n_t^i}{n_t^{i_\star}}} c\sqrt{n_t^i \ln \frac{K \ln n_t^{i_\star}}{\delta}} - 2c\sqrt{n_t^i \ln \frac{K \ln n_t^i}{\delta}} - \widehat{d}_t^i \sqrt{n_t^i}. \qquad (\text{test not triggered})$$

Rearranging terms and plugging this expression in the bound above gives

$$\sum_{k=1}^{n_t^i} \mathrm{reg}(\pi_{(k)}^i) \leq 3d_t^{i_\star} \sqrt{n_t^i} + \sqrt{\frac{n_t^i}{n_t^{i_\star}}} c\sqrt{n_t^i \ln \frac{K \ln n_t^{i_\star}}{\delta}} + 2c\sqrt{n_t^i \ln \frac{K \ln n_t^i}{\delta}} + \widehat{d}_t^i \sqrt{n_t^i}$$

$$\leq 3d_t^{i_\star} \sqrt{n_t^i} + 3c\sqrt{n_t^i \ln \frac{K \ln n_t^{i_\star}}{\delta}} + 2c\sqrt{n_t^i \ln \frac{K \ln n_t^i}{\delta}} + \widehat{d}_t^i \sqrt{n_t^i} \qquad (\text{Equation (13)})$$

$$\leq 3d_t^{i_\star} \sqrt{n_t^i} + 3c\sqrt{n_t^i \ln \frac{K \ln n_t^{i_\star}}{\delta}} + 2c\sqrt{n_t^i \ln \frac{K \ln n_t^i}{\delta}} + 3\widehat{d}_t^{i_\star} \sqrt{n_t^{i_\star}}$$

$$\qquad\qquad (\text{Equation (13)})$$

$$\leq 3d_t^{i_\star} \sqrt{n_t^i} + 3\widehat{d}_t^{i_\star} \sqrt{n_t^{i_\star}} + 5c\sqrt{n_t^i \ln \frac{K \ln t}{\delta}} \qquad (\max(n_t^i, n_t^{i_\star}) \leq t)$$

$$\leq 3d_t^{i_\star} \sqrt{n_t^i} + 3d_{\min} \sqrt{n_t^{i_\star}} + 5c\sqrt{n_t^i \ln \frac{K \ln t}{\delta}} \qquad (\text{Lemma D.1})$$

Finally, Lemma D.1 also implies $n_t^i \leq n_t^{i_\star} + 1$ and since $d_{\min} \leq d_t^{i_\star}$,

$$\sum_{k=1}^{n_t^i} \mathrm{reg}(\pi_{(k)}^i) \leq 6d_t^{i_\star} \sqrt{n_t^{i_\star} + 1} + 5c\sqrt{(n_t^{i_\star} + 1) \ln \frac{K \ln t}{\delta}}.$$

The statement follows by setting $t = T$. $\qquad\qquad\square$

# E  REWARD FUNCTIONS DEFINITIONS

In this section, we present the definition of the human-engineered reward functions and the task reward functions used to evaluate the generated reward in Table 1. The task reward functions are the same as the ones used in Ma et al. (2024).

Table 1: Task reward functions definitions.

| ENVIRONMENT | TASK REWARD |
|---|---|
| CARTPOLE | $\sum \mathbb{1}\{\texttt{agent is alive}\}$ |
| BALLBALANCE | $\sum \mathbb{1}\{\texttt{agent is alive}\}$ |
| ANT | $\texttt{current\_distance - previous\_distance}$ |
| HUMNAOID | $\texttt{current\_distance - previous\_distance}$ |
| ALLEGROHAND | $\sum \mathbb{1}\{\texttt{rotation\_distance < 0.1}\}$ |
| SHADOWHAND | $\sum \mathbb{1}\{\texttt{rotation\_distance < 0.1}\}$ |

The human-designed reward functions from (Makoviychuk et al., 2021) are

- CARTPOLE

$$r = \left(1.0 - \texttt{pole\_angle}^2 - 0.01 \cdot |\texttt{cart\_vel}| - 0.005 \cdot |\texttt{pole\_vel}|\right).$$

  The reward is additionally multiplied by $-2.0$ if $|\texttt{cart\_pos}| > \texttt{reset\_dist}$ and multiplied by $-2.0$ once again if $\texttt{pole\_angle} > \frac{\pi}{2}$.

- BALLBALANCE

$$r = \texttt{pos\_reward} \times \texttt{speed\_reward} = \frac{1}{1 + \texttt{ball\_dist}} \times \frac{1}{1 + \texttt{ball\_speed}},$$

  where

$$\texttt{ball\_dist} = \sqrt{\texttt{ball\_pos\_x}^2 + \texttt{ball\_pos\_y}^2 + (\texttt{ball\_pos\_z} - 0.7)^2},$$

  where $0.7$ is the desired height above the ground, and

$$\texttt{ball\_speed} = \|\texttt{ball\_velocity}\|_2.$$

- ANT and HUMNAOID

$$
\begin{aligned}
r = {}& r_{\text{progress}} + r_{\text{alive}} \times \mathbb{1}\{\texttt{torso\_height} \geq \texttt{termination\_height}\} + r_{\text{upright}} \\
& + r_{\text{heading}} + r_{\text{effort}} + r_{\text{act}} + r_{\text{dof}} \\
& + r_{\text{death}} \times \mathbb{1}\{\texttt{torso\_height} \leq \texttt{termination\_height}\},
\end{aligned}
$$

  where

$$
\begin{aligned}
r_{\text{progress}} &= \texttt{current\_potential - previous\_potential} \\
r_{\text{upright}} &= \langle \texttt{torso\_up\_vector}, \texttt{up\_vector} \rangle > 0.93 \\
r_{\text{heading}} &= \texttt{heading\_vector} \times \begin{cases} 1.0, & \text{if } \texttt{norm\_angle\_to\_target} \geq 0.8 \\ \frac{\texttt{norm\_angle\_to\_target}}{0.8}, & \text{otherwise} \end{cases} \\
r_{\text{act}} &= -\sum \|\texttt{actions}\|^2 \\
r_{\text{effort}} &= \sum_{i=1}^{N} \texttt{actions}_i \times \texttt{normalized\_motor\_strength}_i \times \texttt{dof\_velocity}_i \\
\texttt{potential} &= -\frac{\|p_{\text{target}} - p_{\text{torso}}\|_2}{dt}
\end{aligned}
$$

- ALLEGROHAND and SHADOWHAND

$$r = -10r_{\texttt{dist}} + r_{\texttt{rot}} - 2 \times 10^{-4} r_{\texttt{act}}$$

where

$$r_{\texttt{dist}} = \|p_{\texttt{obj}} - p_{\texttt{target}}\|_2$$
$$r_{\texttt{rot}} = \frac{1}{|\texttt{rot\_dist}| + 0.1}$$
$$r_{\texttt{act}} = \sum \|\texttt{actions}\|^2$$
$$\texttt{rot\_dist} = 2 \times \arcsin\left(\max\left(1, \|q_{\texttt{obj}}, \overline{q_{\texttt{target}}}\|_2\right)\right)$$

where $q$ is the quaternion and $\overline{q}$ is its conjugate.

## E.1 REWARD FUNCTIONS SELECTED BY ORSO

We report the best reward function selected by ORSO below. The reward functions are reported as is, with only the formatting of comments and spacing changed to fit within the box.

---

**Reward Function for Allegro Hand**

```python
def compute_gpt_reward(
    object_rot: torch.Tensor,
    goal_rot: torch.Tensor,
    shadow_hand_dof_pos: torch.Tensor,
    shadow_hand_dof_vel: torch.Tensor,
    actions: torch.Tensor
) -> Tuple[torch.Tensor, Dict[str, torch.Tensor]]:

    # Configurable parameters
    dist_reward_scale = float(2.0)
    action_penalty_scale = float(0.05)
    success_tolerance = float(0.05)
    reach_goal_bonus = float(20.0)

    # Compute distance to goal rotation using Quaternion distance
    q_diff = object_rot - goal_rot
    dist_to_goal = torch.norm(q_diff, dim=-1)

    # Rotation distance reward (scaled)
    rot_reward = torch.exp(-dist_reward_scale * dist_to_goal)

    # Action penalty (scaled)
    action_penalty = torch.sum(actions**2, dim=-1)
    action_penalty_scaled = action_penalty_scale * action_penalty

    # Check if the goal has been reached within the tolerance
    success_mask = dist_to_goal < success_tolerance
    goal_bonus = torch.where(
        success_mask,
        torch.tensor(reach_goal_bonus, device=dist_to_goal.device),
        torch.tensor(0.0, device=dist_to_goal.device)
    )

    # Total reward
    reward = rot_reward - action_penalty_scaled + goal_bonus

    # Dictionary of individual reward components
    reward_components = {
        "rot_reward": rot_reward,
        "action_penalty": action_penalty_scaled,
        "goal_bonus": goal_bonus
    }

    return reward, reward_components
```

**Reward Function for Ant**

```python
def compute_gpt_reward(
    root_states: torch.Tensor,
    actions: torch.Tensor,
    dt: float
) -> Tuple[torch.Tensor, Dict[str, torch.Tensor]]:
    # Device
    device = root_states.device

    # Extract necessary information from the root states
    velocity = root_states[:, 7:10]  # [vx, vy, vz]
    torso_position = root_states[:, 0:3]  # [px, py, pz]

    # Forward velocity along the x-axis
    forward_velocity = velocity[:, 0]

    # Reward component: scaled forward velocity
    # Retain existing scaling factor
    forward_reward = forward_velocity * 2.0

    # Penalty for large actions
    # (to avoid unnecessary or jerky movements)
    action_penalty = torch.sum(actions**2, dim=-1)
    # Increased scaling factor for more impact
    action_penalty_scaled = action_penalty * 1.0

    # Desired height range (e.g., 0.45 to 0.55)
    target_height = torch.tensor(0.5, device=device)
    height_diff = torch.abs(torso_position[:, 2] - target_height)
    # Adjusted temperature parameter to increase contribution
    balance_temperature = 0.1
    # Retain existing scaling
    balance_reward = torch.exp(-height_diff/balance_temperature) * 5.0

    # Additional penalty for deviation from target angle
    # (to encourage running straight)
    target_angle = torch.tensor(0.0, device=device)
    # Assuming index 5 is yaw angle
    angle_diff = torch.abs(root_states[:, 5] - target_angle)
    angle_penalty = -torch.exp(-angle_diff / balance_temperature)

    # Survival bonus to encourage longer episode lengths
    # Reduced overall magnitude
    survival_bonus = torch.ones_like(forward_velocity) * 0.5

    # Total reward calculation
    reward = forward_reward + balance_reward +
        angle_penalty - action_penalty_scaled +
        survival_bonus

    # Dictionary of individual reward components for debugging
    reward_components = {
        'forward_reward': forward_reward,
        'action_penalty_scaled': -action_penalty_scaled,
        'balance_reward': balance_reward,
        'angle_penalty': angle_penalty,
        'survival_bonus': survival_bonus
    }

    return reward, reward_components
```

**Reward Function for Ball Balance**

```python
def compute_gpt_reward(
    ball_positions: torch.Tensor,
    ball_linvels: torch.Tensor
) -> Tuple[torch.Tensor, Dict[str, torch.Tensor]]:
    """
    Compute the reward for keeping the ball on the table top
    without falling.

    Args:
    - ball_positions: torch.Tensor of shape (N, 3) giving the
        positions of the balls.
    - ball_linvels: torch.Tensor of shape (N, 3) giving the
        linear velocities of the balls.

    Returns:
    - reward: the total reward as a torch.Tensor of shape (N,)
    - reward_components: dictionary with individual reward components.
    """

    # Assume ball_positions[:, 2] is the height z of the ball.
    target_height = torch.tensor(0.5, device=ball_positions.device)

    # Reward for staying close to the target height
    height_diff = torch.abs(ball_positions[:, 2] - target_height)
    # Decreased temperature for larger impact
    height_temp = torch.tensor(5.0, device=ball_positions.device)
    height_reward = torch.exp(-height_diff * height_temp)

    # Reward for having low linear velocity
    ball_linvels_norm = torch.linalg.norm(ball_linvels, dim=1)
    # Increased scale for more significant impact
    vel_scale = torch.tensor(10.0, device=ball_positions.device)
    vel_reward = torch.exp(-ball_linvels_norm * vel_scale)

    # Penalty for being far from the center (in xy-plane)
    center_xy = torch.tensor([0, 0], device=ball_positions.device)
    xy_diff = torch.linalg.norm(
        ball_positions[:, :2] - center_xy,
        dim=1
    )
    # Some threshold distance
    xy_threshold = torch.tensor(0.5, device=ball_positions.device)
    xy_penalty = torch.where(
        xy_diff > xy_threshold,
        -torch.exp(xy_diff - xy_threshold),
        torch.tensor(0.0, device=ball_positions.device)
    )

    # Combine the rewards
    total_reward = height_reward + vel_reward + xy_penalty

    # Compile individual components into a dictionary
    reward_components = {
        "height_reward": height_reward,
        "vel_reward": vel_reward,
        "xy_penalty": xy_penalty
    }

    return total_reward, reward_components
```

**Reward Function for Cartpole**

```python
def compute_gpt_reward(
    dof_pos: torch.Tensor,
    dof_vel: torch.Tensor,
) -> Tuple[torch.Tensor, Dict[str, torch.Tensor]]:

    # Extract pole angle and angular velocity
    pole_angle = dof_pos[:, 1]
    pole_ang_vel = dof_vel[:, 1]

    # Reward components
    # Reward for keeping the pole upright
    upright_bonus_t = 10.0
    upright_bonus = torch.exp(-upright_bonus_t*(pole_angle**2))

    # Penalty for pole's angular velocity (to encourage stability)
    ang_vel_penalty_t = 0.1
    ang_vel_penalty = torch.exp(-ang_vel_penalty_t*(pole_ang_vel**2))

    # Sum the rewards and penalties
    reward = upright_bonus + ang_vel_penalty

    # Create a dictionary of individual reward components for
    # debugging or further analysis
    reward_components = {
        'upright_bonus': upright_bonus,
        'ang_vel_penalty': ang_vel_penalty,
    }

    return reward, reward_components
```

**Reward Function for Humanoid**

```python
def compute_gpt_reward(
    root_states: torch.Tensor,
    targets: torch.Tensor,
    dt: float
) -> Tuple[torch.Tensor, Dict[str, torch.Tensor]]:
    # Extract relevant components
    velocity = root_states[:, 7:10]

    # Vector pointing to the target
    torso_position = root_states[:, 0:3]
    to_target = targets - torso_position
    to_target[:, 2] = 0

    # Normalize to_target to get direction
    direction_to_target = torch.nn.functional.normalize(
        to_target,
        p=2.0,
        dim=-1
    )

    # Project velocity onto direction to target to get velocity
    # component in the right direction
    velocity_towards_target = torch.sum(
        velocity * direction_to_target,
        dim=-1,
        keepdim=True
    )

    # Reward for moving towards the target quickly
    speed_reward = velocity_towards_target.squeeze()

    # Apply an exponential transformation to encourage higher speeds
    temp_speed = 0.1
    speed_reward_transformed = torch.exp(speed_reward/temp_speed)-1.0

    # Combine rewards (single component in this case)
    total_reward = speed_reward_transformed

    # Reward components in a dictionary form
    rewards = {"speed_reward": speed_reward_transformed}

    return total_reward, rewards
```

Reward Function for Shadow Hand

```python
def compute_gpt_reward(
    object_rot: torch.Tensor,
    goal_rot: torch.Tensor,
    actions: torch.Tensor,
    success_tolerance: float,
    reach_goal_bonus: float,
    rot_reward_scale: float,
    action_penalty_scale: float
) -> Tuple[torch.Tensor, Dict[str, torch.Tensor]]:

    # Rotation Distance Reward with adjusted scaling
    rot_dist = torch.norm(object_rot - goal_rot, dim=-1)
    # New temperature parameter for rotational reward
    rot_reward_temp = 3.0
    rot_reward = torch.exp(-rot_dist*rot_reward_scale/rot_reward_temp)

    # Goal Achievement Bonus
    goal_reached = rot_dist < success_tolerance
    goal_bonus = reach_goal_bonus * goal_reached.float()

    # Action Penalty with increased scale
    # Increasing the action penalty scale
    increased_aps = 2.0 * action_penalty_scale
    action_penalty = torch.sum(actions**2, dim=-1) * increased_aps

    # Intermediate Reward for making progress towards rotating to goal
    interm_steps_temp = 0.5
    intermediate_steps_reward = torch.exp(-rot_dist/interm_steps_temp)

    # Penalty for large deviations from goal orientation
    deviation_scale = 0.2
    deviation_penalty = rot_dist * deviation_scale

    # Calculate total reward
    total_reward = rot_reward + goal_bonus +
        intermediate_steps_reward - action_penalty -
        deviation_penalty

    # Create a dictionary of individual rewards for monitoring
    reward_dict = {
        "rot_reward": rot_reward,
        "goal_bonus": goal_bonus,
        "intermediate_steps_reward": intermediate_steps_reward,
        "action_penalty": action_penalty,
        "deviation_penalty": deviation_penalty
    }

    return total_reward, reward_dict
```

## F   IMPLEMENTATION DETAILS

**Rejection Sampling**   While the LLM produces seemingly good code, this does not guarantee that the sampled code is bug-free and runnable. In ORSO, we employ a simple rejection sampling technique to construct sets of only valid reward functions with high probability, such that reward functions that cannot be compiled or produce $\pm\infty$ or `NaN` values are discarded.

Given criteria $\phi$ to be satisfied, our rejection sampling scheme repeats the steps in Algorithm 3 until we have sampled the desired number, $K$, of valid reward functions.

---

**Algorithm 3** Rejection Sampling in ORSO

---

1:  Sample a candidate reward function $f \sim G$
2:  **if** $\phi(f)$ is satisfied **then**
3:      Add $f$ to the set of candidate reward functions
4:  **else**
5:      Reject reward function $f$
6:  **end if**

---

In our practical implementation, checking if criteria $\phi$ are satisfied consists of instantiating an environment with the generated reward function, running a random policy on it, and checking the values produced by the reward function. If the environment cannot be instantiated or if the values returned by the reward function are $\pm\infty$ or `NaN`, the reward function is rejected. It is worth making two important observations. First, this is much computationally cheaper than instantiating the environment for training because one does not need to initialize large neural networks and can use fewer parallel environments than the number necessary for training. Moreover, we note that the rejection sampling mechanism only guarantees a higher probability of a valid reward function code as the policy used to evaluate the function is random and the optimization process used during the training of an RL algorithm could still induce undesirable values.

**Iterative Improvement of the Reward Function Set**   In the initial phase of ORSO, the algorithm generates a set of candidate reward functions $\mathcal{R}^K$ for the online reward selection and policy optimization step. While this approach is effective if $\mathcal{R}^K$ contains an effective reward function, any selection process will fail to achieve a high task reward if the set does not contain a good reward function. To address this limitation, we introduce a mechanism for improving the reward function set through iterative resampling and in-context evolution. This is similar to Ma et al. (2024), however, we introduce some important changes to prevent the in-context evolution from overfitting to initially suboptimal reward functions.

Resampling is triggered when at least one reward function has been used to train a policy for the number of iterations specified in the environment configuration or if all the reward functions in the set incurred too large a regret compared to the previous best policy if the algorithm has undergone at least one resampling step.

There are several strategies for resampling reward functions, each with its trade-offs. The simplest approach is to sample new reward functions from scratch, using the same generator $G$ that was used in the initial phase. However, this method may not provide significant improvement, as it essentially restarts the search process without leveraging the information gained from the previous iterations of training.

A more sophisticated approach is to greedily in-context evolve the reward function from the best-performing candidate so far as is done in Ma et al. (2024). This involves making incremental adjustments to the reward function that has shown the most promise, potentially moving it closer to an optimal reward function. However, while this greedy strategy can lead to improvements, it also has the risk of overfitting to an initially suboptimal reward function if, for example, the initial set does not contain effective reward functions.

To mitigate the risk of overfitting, we introduce a simple strategy that allows the algorithm to be more exploratory. Specifically, we combine greedy evolution with random sampling: half of the reward functions are evolved in context from the best-performing candidate, while the other half is sampled from scratch. This approach allows the algorithm to explore new regions of the reward

function space while still exploiting the knowledge gained from previous iterations. We provide the full pseudo-code for ORSO with rejection sampling and iterative improvement in Algorithm 4.

---

**Algorithm 4** ORSO with Rejection Sampling and Iterative Improvement

---

**Require:** MDP $\mathcal{M} = (\mathcal{S}, \mathcal{A}, P, r, \gamma, \rho_0)$, algorithm $\mathfrak{A}$, generator $G$, budget $T$, threshold `n_iters`
1: Sample $K$ valid reward functions $\mathcal{R}^K = \{f^1, \ldots, f^K\} \sim G$ using Algorithm 3
2: Initialize $K$ policies $\{\pi^1, \ldots, \pi^K\}$
3: Initialize selection counts $N^K = \{0, \ldots, 0\}$
4: Set $t \leftarrow 1$
5: **while** $t \leq T$ **do**
6:      Select a model $i_t \in [K]$ according to a selection strategy
7:      Update $\pi^{i_t} \leftarrow \mathfrak{A}_{f^{i_t}}(\mathcal{M}, N, \pi^{i_t})$
8:      Evaluate $\mathcal{J}(\pi^{i_t}) \leftarrow \texttt{Eval}(\pi^{i_t})$
9:      Update selection counts: $N^{i_t} \leftarrow N^{i_t} + 1$
10:      Update variables (e.g., reward estimates and confidence intervals)
11:      **if** $N^{i_t} \geq$ `n_iters` or regret w.r.t. previous best is too high **then**
12:          Resample $\mathcal{R}^K \sim G$ (half in-context evolution, half from scratch)
13:          Sample a new set of reward functions $\mathcal{R}^K = \{f^1, \ldots, f^K\}$ using rejection sampling
14:          Reset policies $\{\pi^1, \ldots, \pi^K\}$
15:          Reset selection counts $N^K = \{0, \ldots, 0\}$
16:      **end if**
17:      $t \leftarrow t + 1$
18: **end while**
19: **return** $\pi_T^\star, f_T^\star = \arg\max_{i \in [K]} \mathcal{J}(\pi^i)$

---

## G    SELECTION ALGORITHMS AND HYPERPARAMETERS

In this section, we present the pseudocode for all reward selection algorithms used in our experiments with their associated hyperparameters in Table 2.

---

**Algorithm 5** $\varepsilon$-Greedy

---

**Require:** Number of arms $K$, total time $T$, exploration probability $\varepsilon$
1: Initialize counts $n_0^i = 0$ and total values $\widehat{u}_0^i = 0$ for all $i \in [K]$
2: **for** $t = 1, \ldots, T$ **do**
3:      Select arm

$$i_t = \begin{cases} \arg\max_i (\widehat{u}_t^i / n_t^i), & \text{with probability } 1 - \varepsilon \\ i \sim \text{Uniform}([K]), & \text{with probability } \varepsilon \end{cases}$$

4:      Play arm $i_t$ and observe reward $r_t$
5:      Set $n_t^i = n_{t-1}^i$, and $\widehat{u}_t^i = \widehat{u}_{t-1}^i$ for all $i \in [K] \setminus \{i_t\}$
6:      Update statistics for current learner $n_t^{i_t} = n_{t-1}^{i_t} + 1$ and $\widehat{u}_t^{i_t} = \widehat{u}_{t-1}^{i_t} + r_t$
7: **end for**

---

**Algorithm 6** Explore-then-Commit

---

**Require:** Number of arms $K$, total time $T$, exploration phase length $T_0$
1: Initialize counts $n_0^i = 0$ and total values $\widehat{u}_0^i = 0$ for all $i \in [K]$
2: // Explore
3: **for** $t = 1, \ldots, T_0$ **do**
4:      Select arm $i_t = (t \mod K) + 1$
5:      Play arm $i_t$ and observe reward $r_t$
6:      Set $n_t^i = n_{t-1}^i$, and $\widehat{u}_t^i = \widehat{u}_{t-1}^i$ for all $i \in [K] \setminus \{i_t\}$
7:      Update statistics for current learner $n_t^{i_t} = n_{t-1}^{i_t} + 1$ and $\widehat{u}_t^{i_t} = \widehat{u}_{t-1}^{i_t} + r_t$
8: **end for**
9: // Commit
10: $i_\star = \arg\max_i (u_t^i / n_t^i)$
11: **for** $t = T_0 + 1$ to $T$ **do**
12:      Play arm $i_\star$ and observe reward $r_t$
13: **end for**

---

**Algorithm 7** UCB (Upper Confidence Bound)

---

**Require:** Number of arms $K$, total time $T$, confidence multiplier $c$
1: Initialize counts $n_0^i = 0$ and total values $\widehat{u}_0^i = 0$ for all $i \in [K]$
2: **for** $t = 1, \ldots, K$ **do**
3:      Select arm $i_t = t$
4:      Play arm $i_t$ and observe reward $r_t$
5:      Update statistics for current learner $n_t^{i_t} = n_{t-1}^{i_t} + 1$ and $\widehat{u}_t^{i_t} = \widehat{u}_{t-1}^{i_t} + r_t$
6: **end for**
7: **for** $t = K + 1, \ldots, T$ **do**
8:      Select arm

$$i_t = \arg\max_i \left( \frac{\widehat{u}_t^i}{n_t^i} + c\sqrt{2\frac{\ln t}{n_t^i}} \right)$$

9:      Play arm $i_t$ and observe reward $r_t$
10:      Set $n_t^i = n_{t-1}^i$, and $\widehat{u}_t^i = \widehat{u}_{t-1}^i$ for all $i \in [K] \setminus \{i_t\}$
11:      Update statistics for current learner $n_t^{i_t} = n_{t-1}^{i_t} + 1$ and $\widehat{u}_t^{i_t} = \widehat{u}_{t-1}^{i_t} + r_t$
12: **end for**

---

---

**Algorithm 8** Exp3 (Exponential-weight algorithm for Exploration and Exploitation)

---

**Require:** Number of arms $K$, total time $T$, learning rate $\eta$

1: Initialize weights $w_0^i = 1$ and probabilities $p_0^i = 1/K$ for all $i \in [K]$
2: **for** $t = 1, \ldots, T$ **do**
3:      Select arm $i_t$ according to distribution $P_t = [p_t^1, \ldots, p_t^K]$
4:      Play arm $i_t$ and observe reward $r_t$
5:      Estimate reward $\widehat{r}_t = r_t / p_t^{i_t}$
6:      Update weight $w_t^{i_t} = w_{t-1}^{i_t} \exp(\eta \widehat{r}_t / K)$
7:      Update probabilities

$$p_t^i = (1 - \eta) \frac{w_t^i}{\sum_{j=1}^{K} w_t^j} + \frac{\eta}{K} \qquad \text{for all } i \in [K]$$

8: **end for**

---

Table 2: Hyperparameters for MAB Algorithms

| ALGORITHM | PARAMETER | VALUE |
|---|---|---|
| EPSILON-GREEDY | $\varepsilon$ | 0.1 |
| EXPLORE-THEN-COMMIT | $T_0$ | $5 \cdot K$ |
| UCB | $c$ | 1.0 |
| EXP3 | $\eta$ | 0.1 |

# H ADDITIONAL EXPERIMENTAL RESULTS

In this section, we report additional experimental evaluations. In particular, we show how different configurations of budget constraints $B$ and sizes $K$ of the reward function set perform with different reward selection algorithms in different environments in Figures 7 to 10.

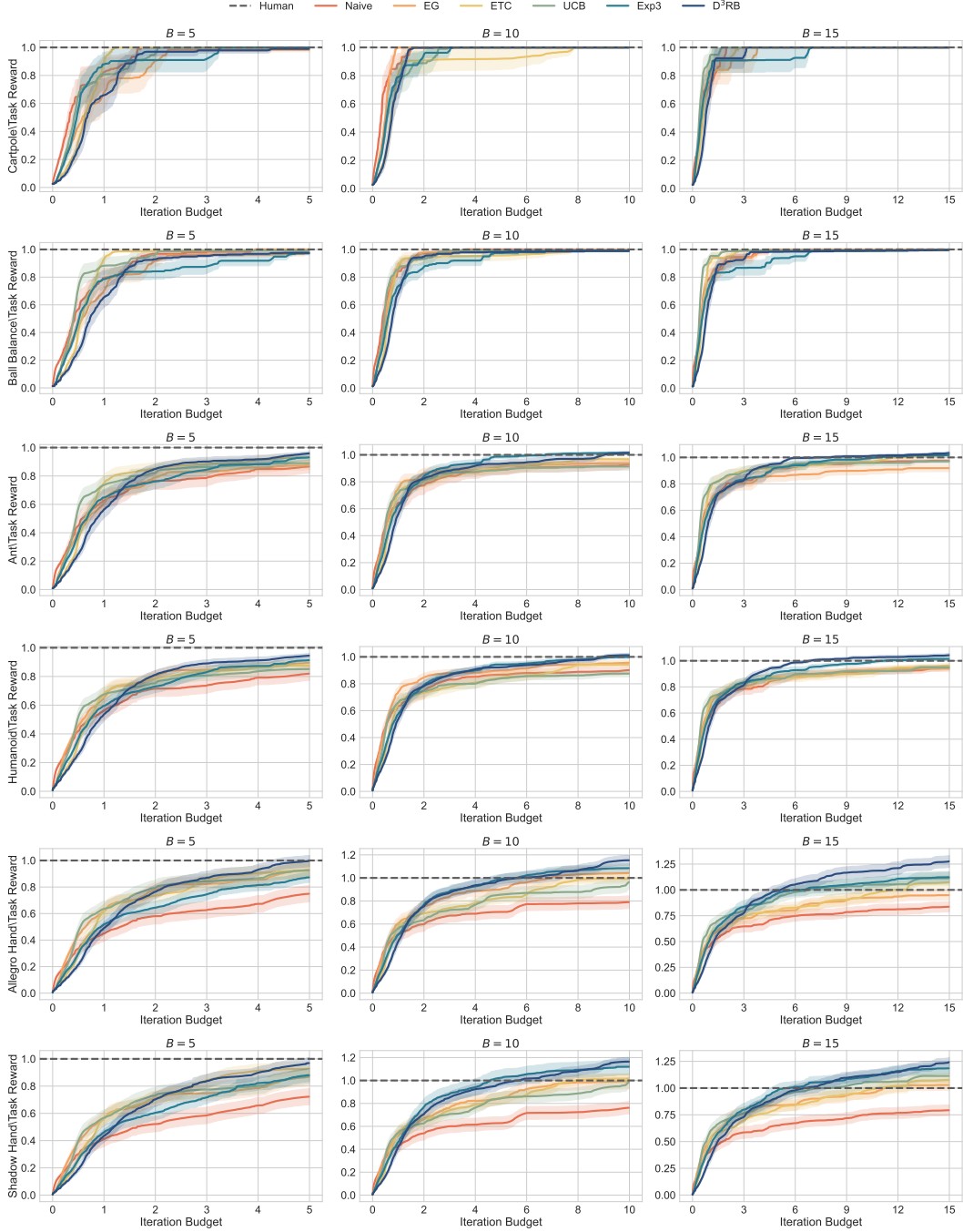

Figure 7: Number of iterations necessary to reach human-engineered reward function performance with different computation budgets and tasks. The shaded areas represent standard errors.

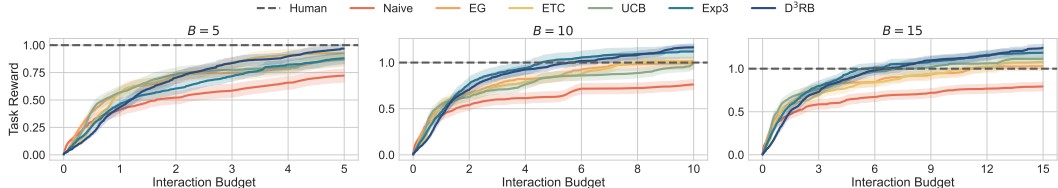

Figure 8: Number of iterations necessary to reach human-engineered reward function performance with different computation budgets. The shaded areas represent standard errors.

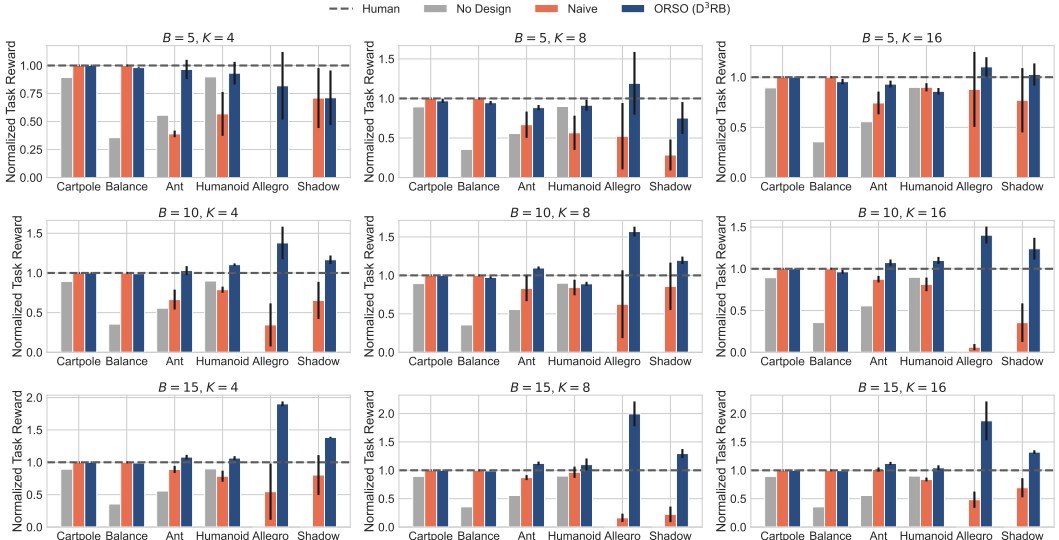

Figure 9: Average performance with standard errors for ORSO with different interaction budget constraints and reward function set size. The dashed horizontal line represents the policies trained with the human-engineered reward function.

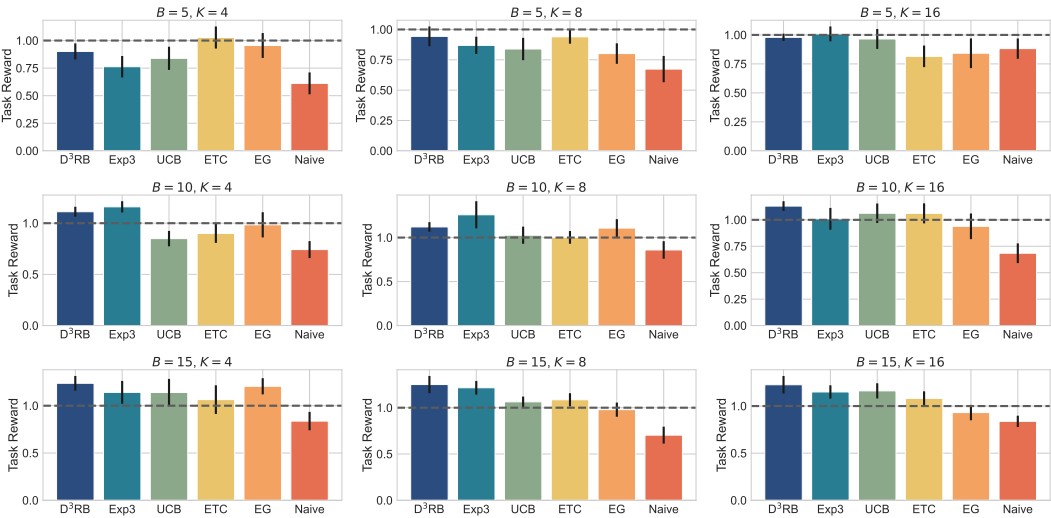

Figure 10: Comparison of different reward selection algorithms for ORSO with different budget constraints and reward function set size.

We also plot in Figure 11 the time necessary to achieve the same performance as policies trained with human-designed reward functions as a function of the number of parallel GPUs available for all budget constraints and all tasks considered.

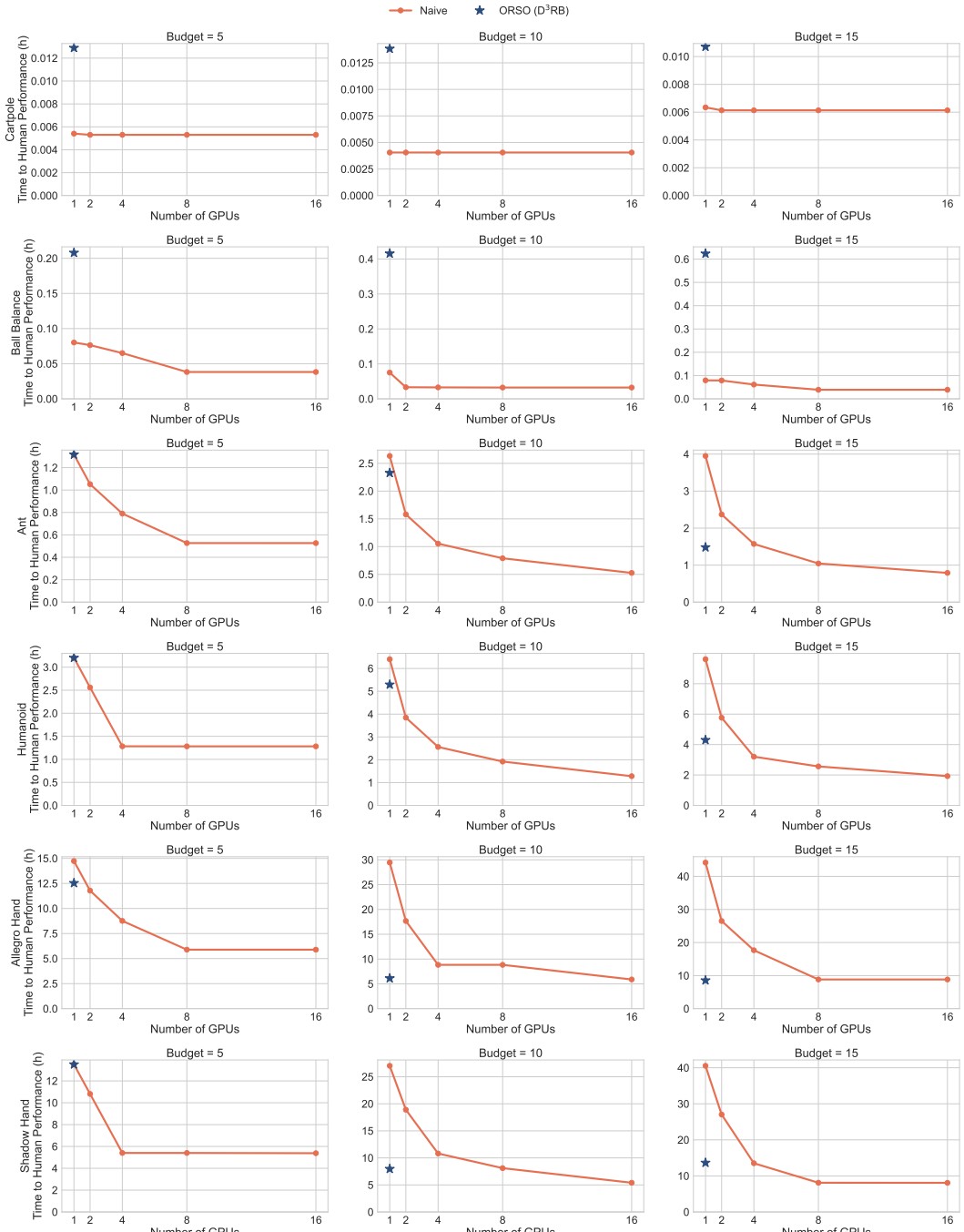

Figure 11: Time necessary to achieve the same performance as policies trained with human-designed reward functions as a function of the number of parallel GPUs.

## H.1 CHOSEN REWARD FUNCTIONS FOR LARGE REWARD SET

In order to validate that ORSO with D³RB indeed chooses the optimal reward function, we train a policy for each of the $K = 96$ reward functions for the ANT task in Figure 6. In Table 3, we report the mean task reward with 95% confidence intervals over five seeds. Rewards are ordered from best to worst, with those within one confidence interval of the best reward underlined. Bolded values indicate the reward functions selected by ORSO across the seeds we ran.

Table 3: Mean task reward for each Reward ID with 95% confidence intervals (CI).

| REWARD ID | MEAN ($\pm$ CI) | REWARD ID | MEAN ($\pm$ CI) |
|---|---|---|---|
| 34 | **10.24 $\pm$ 0.36** | 56 | 5.05 $\pm$ 0.62 |
| 18 | 10.01 $\pm$ 0.63 | 39 | 4.91 $\pm$ 0.64 |
| 71 | 9.98 $\pm$ 0.37 | 74 | 4.88 $\pm$ 0.49 |
| 79 | 9.88 $\pm$ 0.73 | 30 | 4.83 $\pm$ 0.78 |
| 21 | **9.77 $\pm$ 0.22** | 78 | 4.83 $\pm$ 0.35 |
| 94 | 9.70 $\pm$ 0.38 | 6 | 4.76 $\pm$ 0.91 |
| 66 | **9.67 $\pm$ 0.27** | 2 | 4.69 $\pm$ 0.92 |
| 81 | **9.55 $\pm$ 0.80** | 28 | 4.66 $\pm$ 1.21 |
| 70 | **9.51 $\pm$ 0.63** | 8 | 4.65 $\pm$ 0.44 |
| 37 | 9.46 $\pm$ 0.55 | 16 | 4.57 $\pm$ 1.08 |
| 33 | 9.34 $\pm$ 0.83 | 29 | 4.53 $\pm$ 0.81 |
| 95 | 9.27 $\pm$ 0.33 | 65 | 4.44 $\pm$ 0.62 |
| 47 | 9.24 $\pm$ 0.68 | 50 | 4.23 $\pm$ 1.94 |
| 63 | 9.21 $\pm$ 0.76 | 58 | 3.89 $\pm$ 0.43 |
| 54 | 9.20 $\pm$ 0.80 | 53 | 3.86 $\pm$ 0.44 |
| 80 | 9.16 $\pm$ 0.25 | 32 | 3.79 $\pm$ 0.73 |
| 62 | 8.88 $\pm$ 0.37 | 22 | 3.74 $\pm$ 0.54 |
| 38 | 8.81 $\pm$ 0.45 | 3 | 3.48 $\pm$ 1.66 |
| 49 | 8.81 $\pm$ 0.77 | 69 | 3.22 $\pm$ 0.42 |
| 35 | 8.69 $\pm$ 1.07 | 4 | 3.18 $\pm$ 0.42 |
| 5 | 8.61 $\pm$ 0.56 | 88 | 3.18 $\pm$ 0.43 |
| 52 | 8.35 $\pm$ 1.43 | 64 | 3.12 $\pm$ 0.16 |
| 67 | 8.32 $\pm$ 0.85 | 9 | 3.11 $\pm$ 0.39 |
| 46 | 8.30 $\pm$ 0.89 | 17 | 3.10 $\pm$ 0.15 |
| 68 | 8.20 $\pm$ 1.22 | 93 | 3.02 $\pm$ 0.21 |
| 75 | 8.09 $\pm$ 0.40 | 14 | 2.99 $\pm$ 0.52 |
| 84 | 8.05 $\pm$ 1.25 | 45 | 2.89 $\pm$ 0.29 |
| 85 | 7.77 $\pm$ 0.97 | 83 | 2.72 $\pm$ 0.82 |
| 72 | 7.64 $\pm$ 1.27 | 27 | 2.50 $\pm$ 0.72 |
| 55 | 7.43 $\pm$ 1.46 | 10 | 2.15 $\pm$ 0.43 |
| 20 | 7.26 $\pm$ 0.18 | 57 | 1.69 $\pm$ 0.80 |
| 23 | 7.26 $\pm$ 1.12 | 7 | 1.67 $\pm$ 1.01 |
| 86 | 7.15 $\pm$ 0.42 | 82 | 1.03 $\pm$ 0.35 |
| 36 | 7.06 $\pm$ 0.68 | 42 | 0.63 $\pm$ 0.80 |
| 91 | 6.93 $\pm$ 1.45 | 41 | 0.37 $\pm$ 0.30 |
| 1 | 6.50 $\pm$ 1.17 | 43 | 0.33 $\pm$ 0.16 |
| 31 | 6.36 $\pm$ 0.80 | 76 | 0.22 $\pm$ 0.08 |
| 61 | 6.06 $\pm$ 0.93 | 89 | 0.22 $\pm$ 0.07 |
| 19 | 5.78 $\pm$ 1.43 | 24 | 0.21 $\pm$ 0.03 |
| 25 | 5.67 $\pm$ 1.41 | 11 | 0.21 $\pm$ 0.08 |
| 48 | 5.65 $\pm$ 1.54 | 12 | 0.19 $\pm$ 0.14 |
| 59 | 5.59 $\pm$ 1.02 | 15 | 0.19 $\pm$ 0.14 |
| 26 | 5.50 $\pm$ 0.89 | 92 | 0.14 $\pm$ 0.07 |
| 60 | 5.47 $\pm$ 1.17 | 77 | 0.13 $\pm$ 0.04 |
| 44 | 5.47 $\pm$ 1.36 | 13 | 0.05 $\pm$ 0.00 |
| 40 | 5.34 $\pm$ 1.73 | 51 | 0.05 $\pm$ 0.00 |
| 73 | 5.33 $\pm$ 1.77 | 87 | 0.05 $\pm$ 0.02 |
| 0 | 5.30 $\pm$ 1.38 | 90 | 0.00 $\pm$ 0.00 |

## H.2 VISUALIZING ORSO

To better visualize how ORSO selects the best reward function, discards suboptimal ones efficiently, and thanks to this, explores more reward functions, we provide further visualizations in this section.

Figures 12 and 13 show a full training of ORSO ($D^3RB$) and the naive selection stratecy (EUREKA) with a budget $B = 15$ and $K = 16$ on the ALLEGROHAND task, respectively. In both figures, the top plot shows the task reward during training. The colors indicate the reward functions currently in use. The middle plot more clearly shows the reward function being currently used. The vertical axis contains the reward function indices. In both plots, the dashed vertical lines indicate that a resampling has been triggered. Lastly, the bottom plot shows the unnormalized cumulative regret during training.

Comparing the two figures, we can see that ORSO initially explores all reward functions near-uniformly, but quickly finds a policy that surpasses the policy from the human-engineered reward function, leading to a decrease in regret. On the other hand, EUREKA uniformly trains on each reward function leading the algorithm to explore fewer reward functions. Moreover, we see that the lack of rejection sampling can result in initial reward function sets that contain many invalid reward functions – indicated by a $\times$ in the figures.

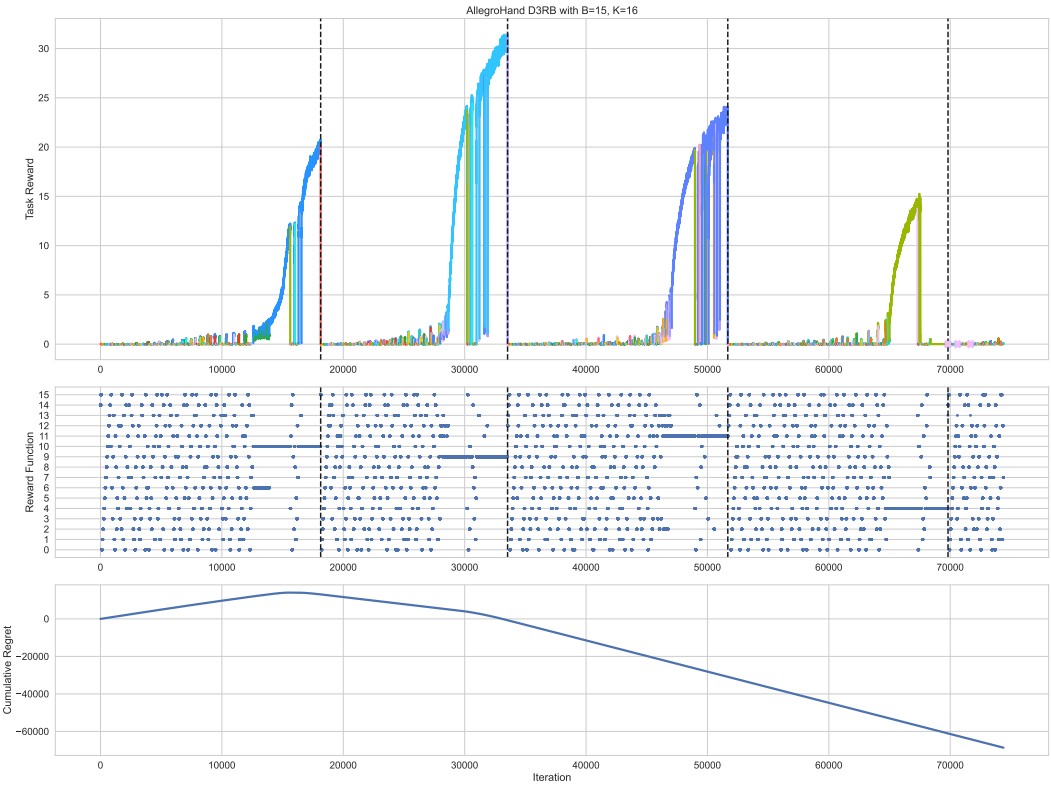

Figure 12: ORSO ($D^3RB$) on ALLEGROHAND with $B = 15$ and $K = 16$.

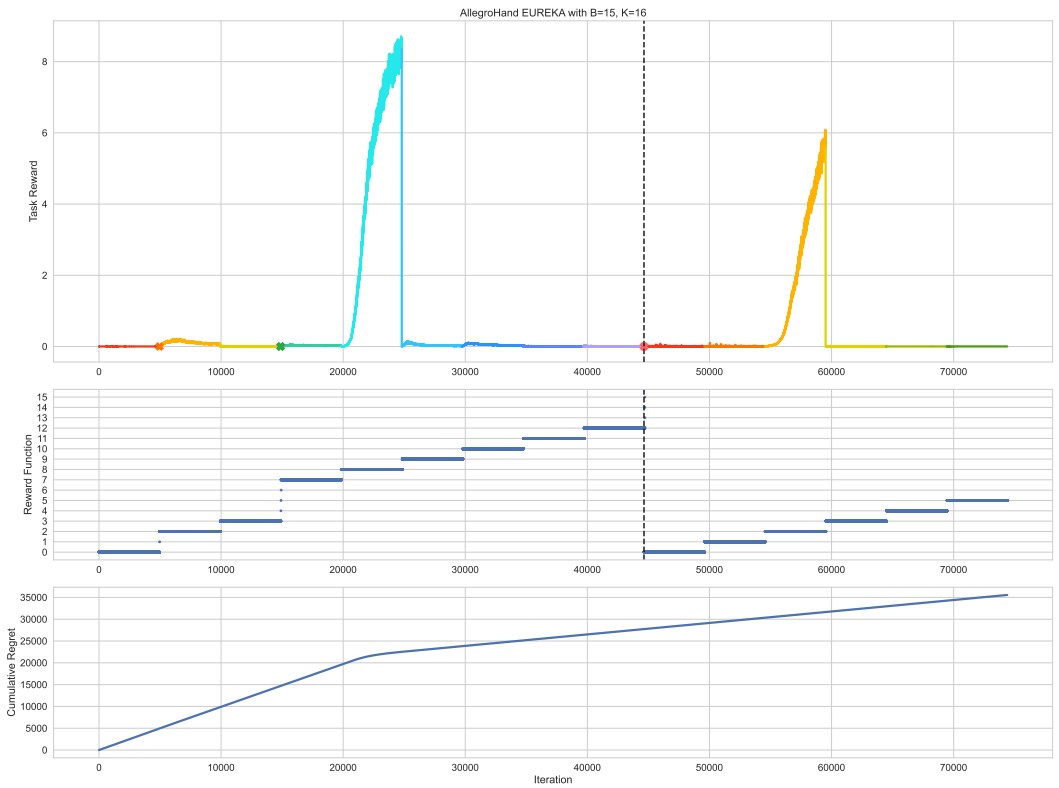

Figure 13: EUREKA on ALLEGROHAND with $B = 15$ and $K = 16$.

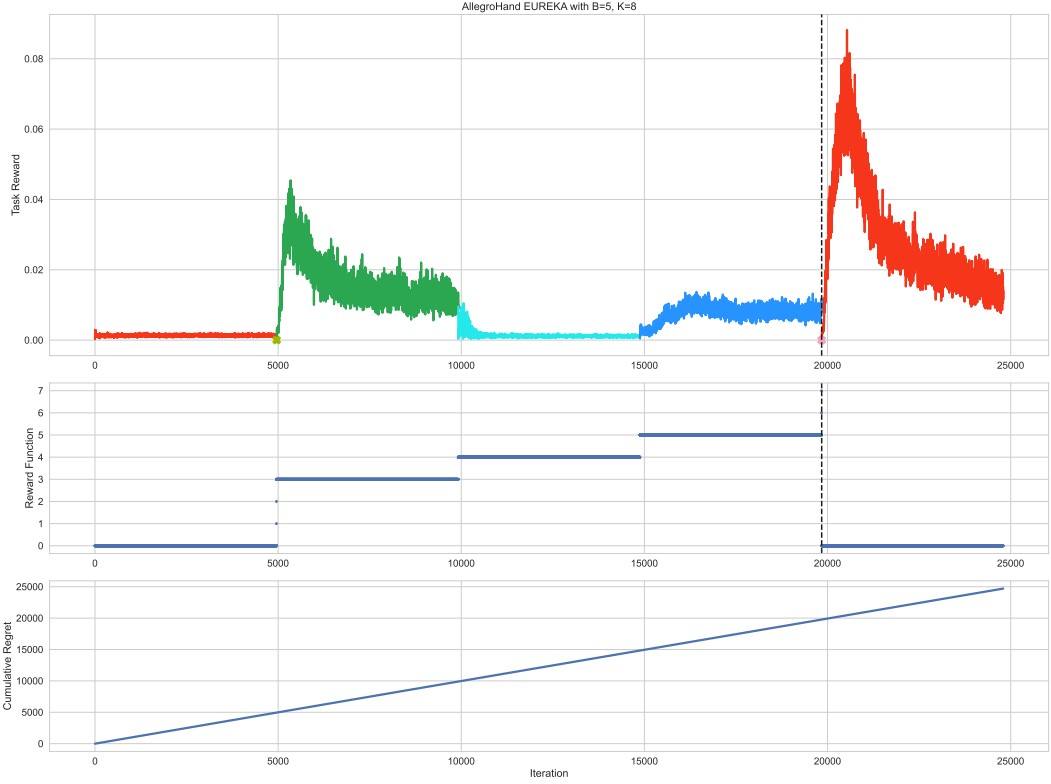

Figure 14: EUREKA on ALLEGROHAND with $B = 5$ and $K = 8$.

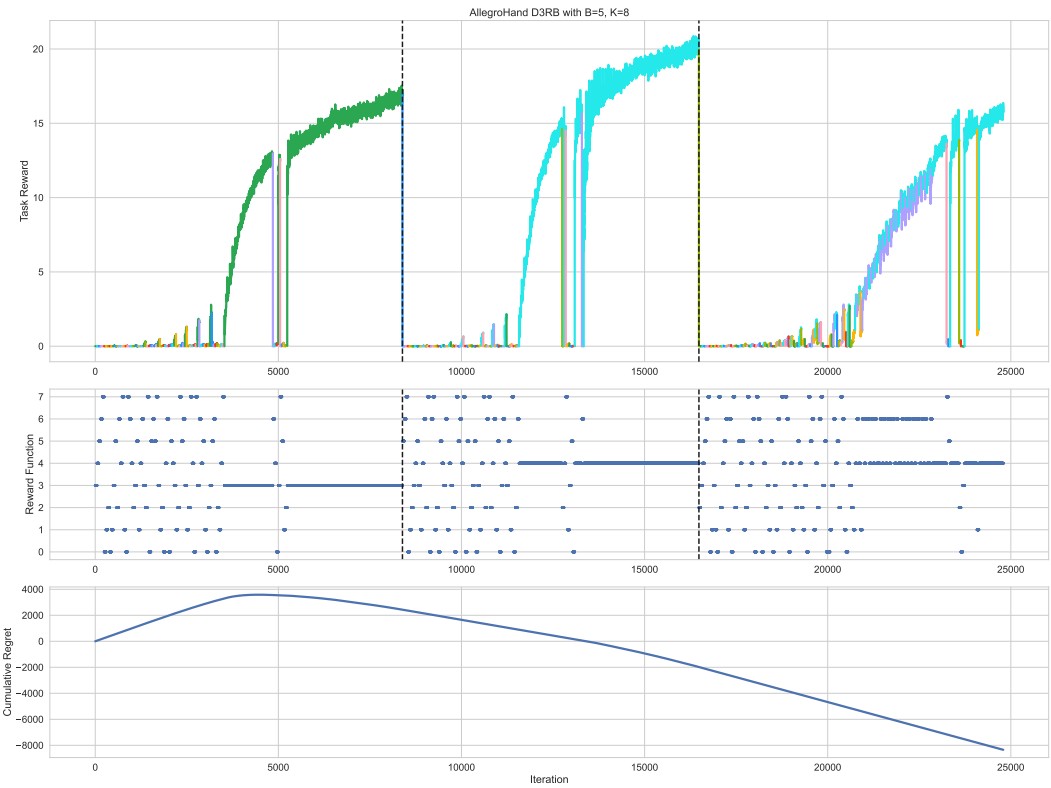

Figure 15: ORSO (D$^3$RB) on ALLEGROHAND with $B = 5$ and $K = 8$.

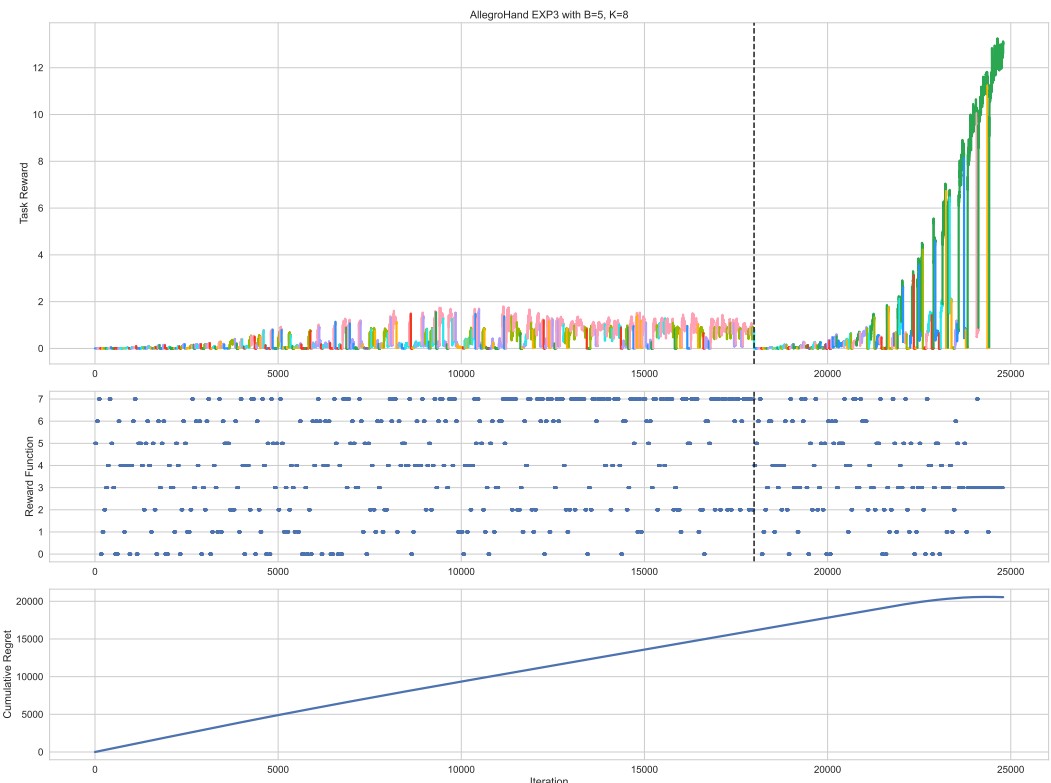

Figure 16: ORSO (Exp3) on ALLEGROHAND with $B = 5$ and $K = 8$.

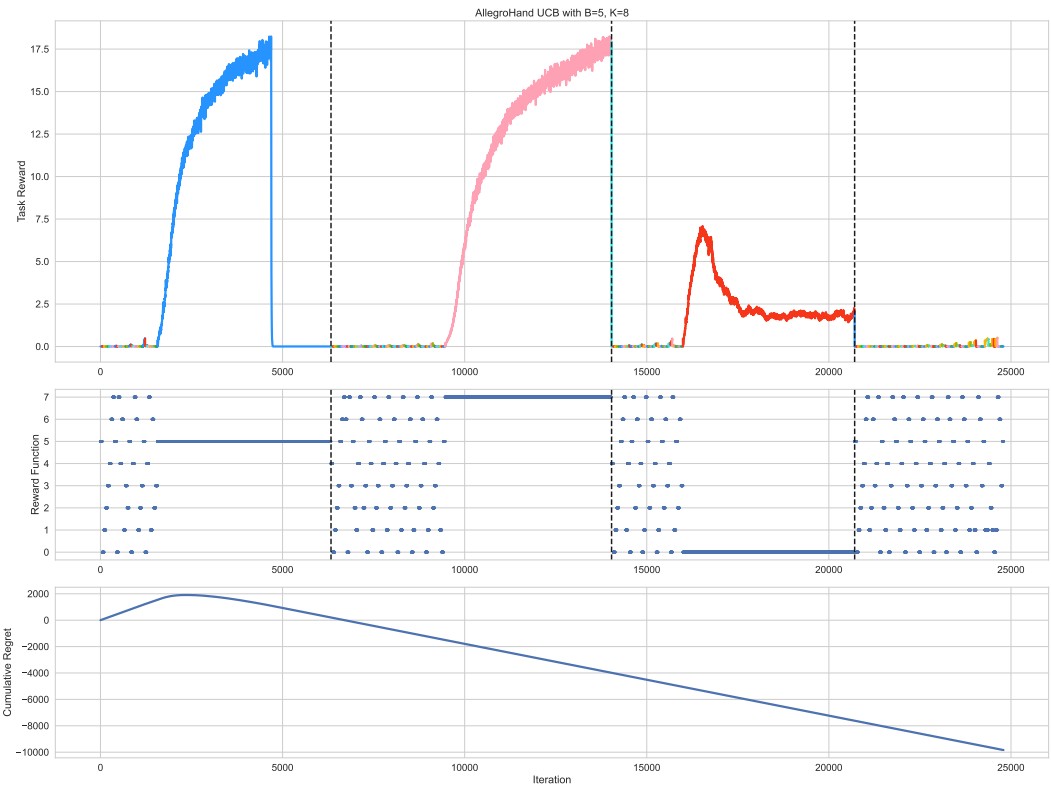

Figure 17: ORSO (UCB) on ALLEGROHAND with $B = 5$ and $K = 8$.

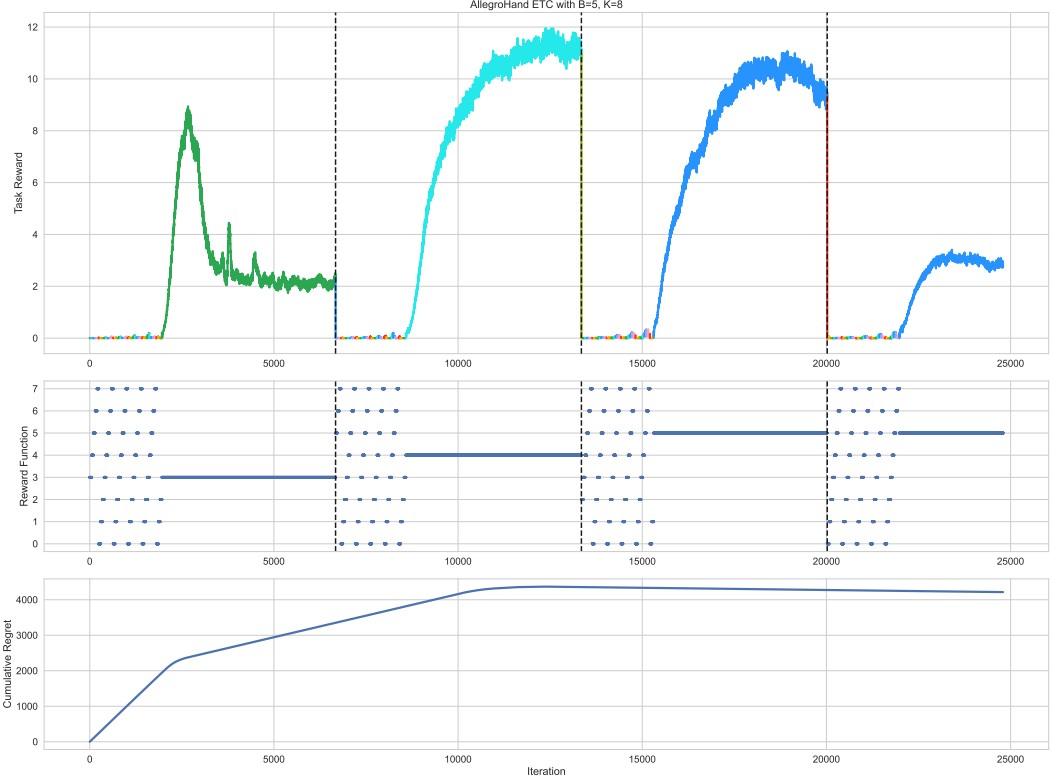

Figure 18: ORSO (ETC) on ALLEGROHAND with $B = 5$ and $K = 8$.

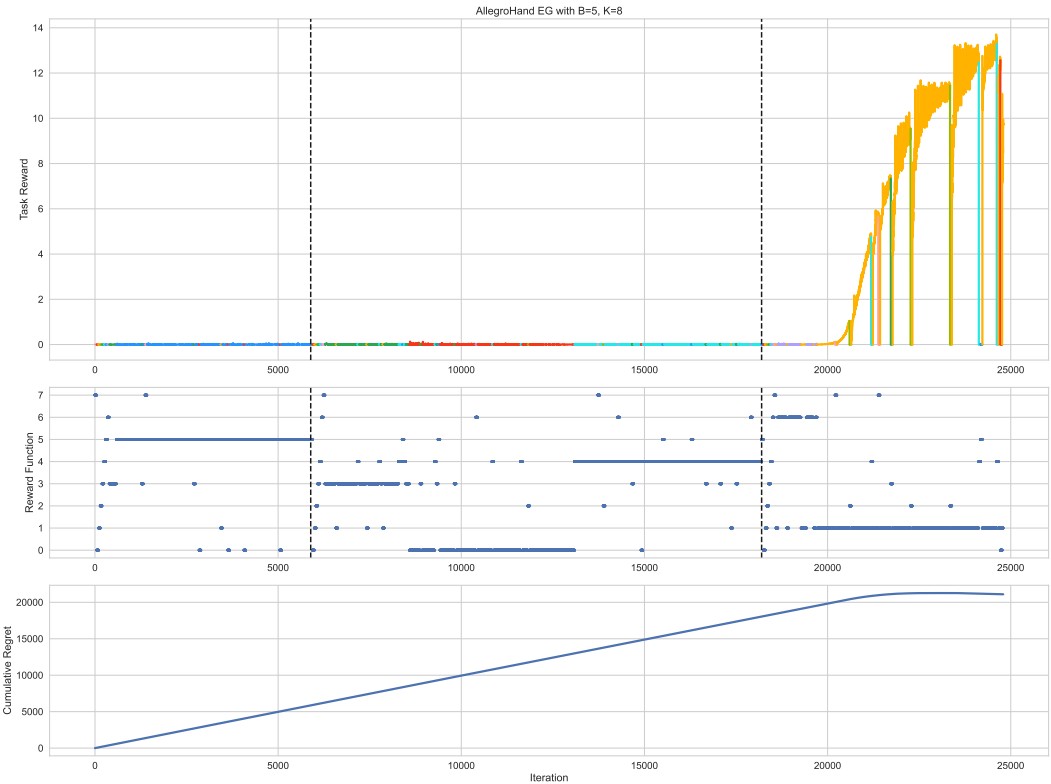

Figure 19: ORSO (EG) on ALLEGROHAND with $B = 5$ and $K = 8$.

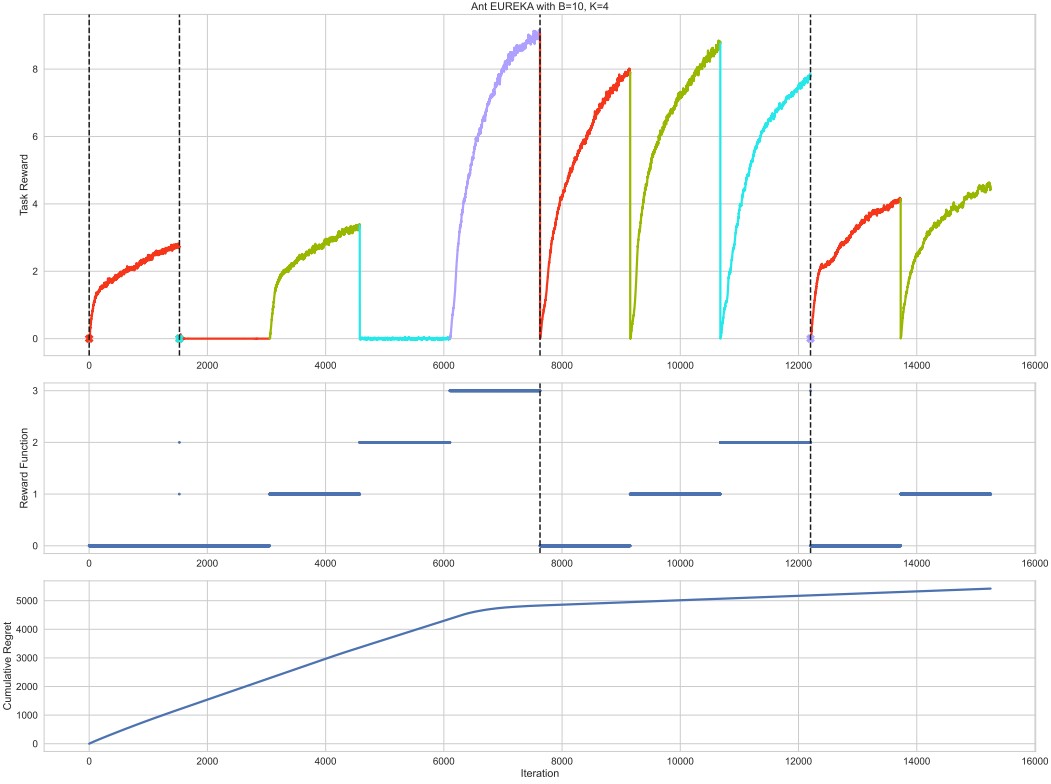

Figure 20: EUREKA on ANT with $B = 10$ and $K = 4$.

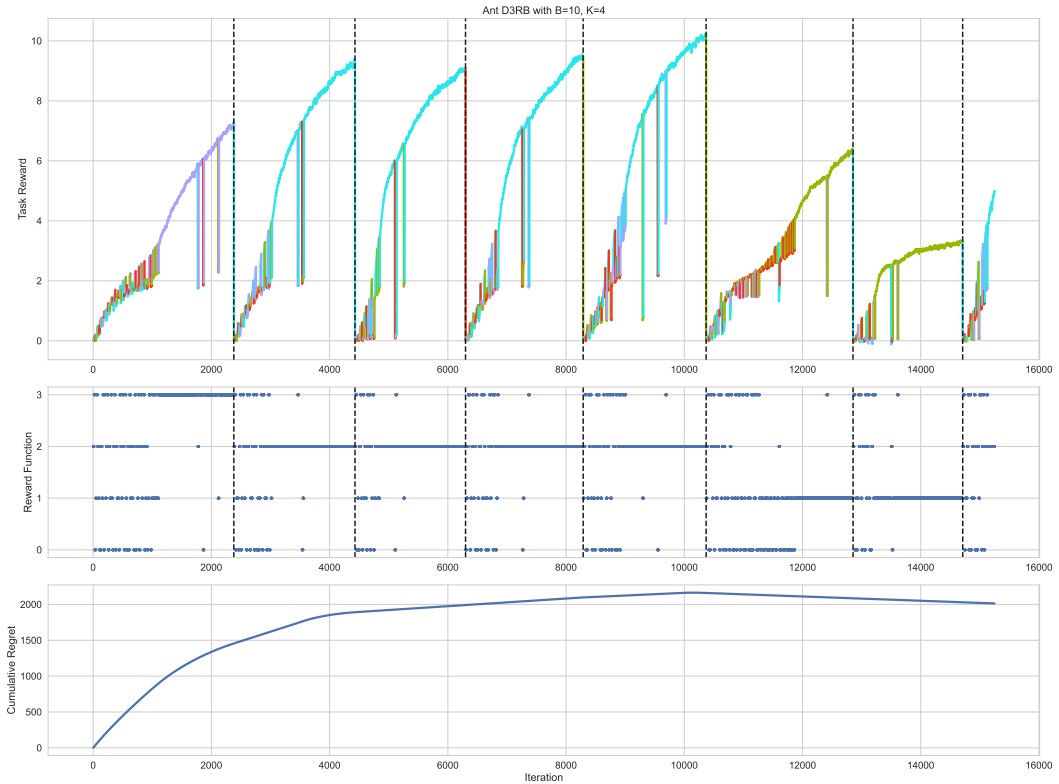

Figure 21: ORSO (D$^3$RB) on ANT with $B = 10$ and $K = 4$.

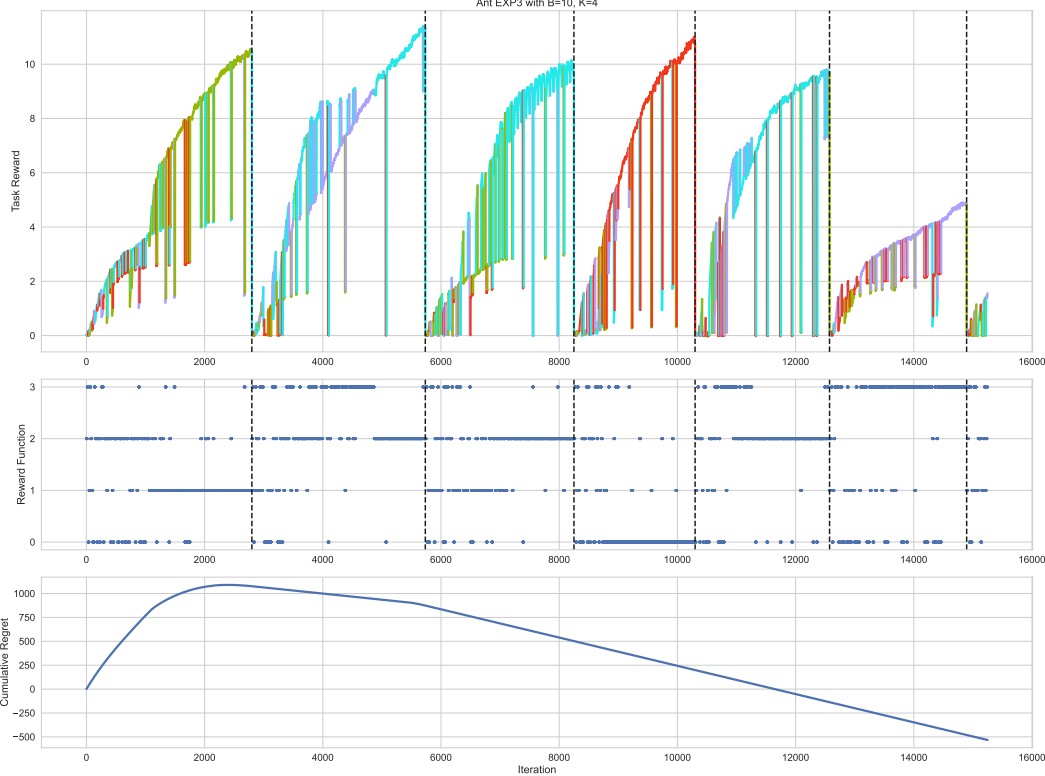

Figure 22: ORSO (Exp3) on ANT with $B = 10$ and $K = 4$.

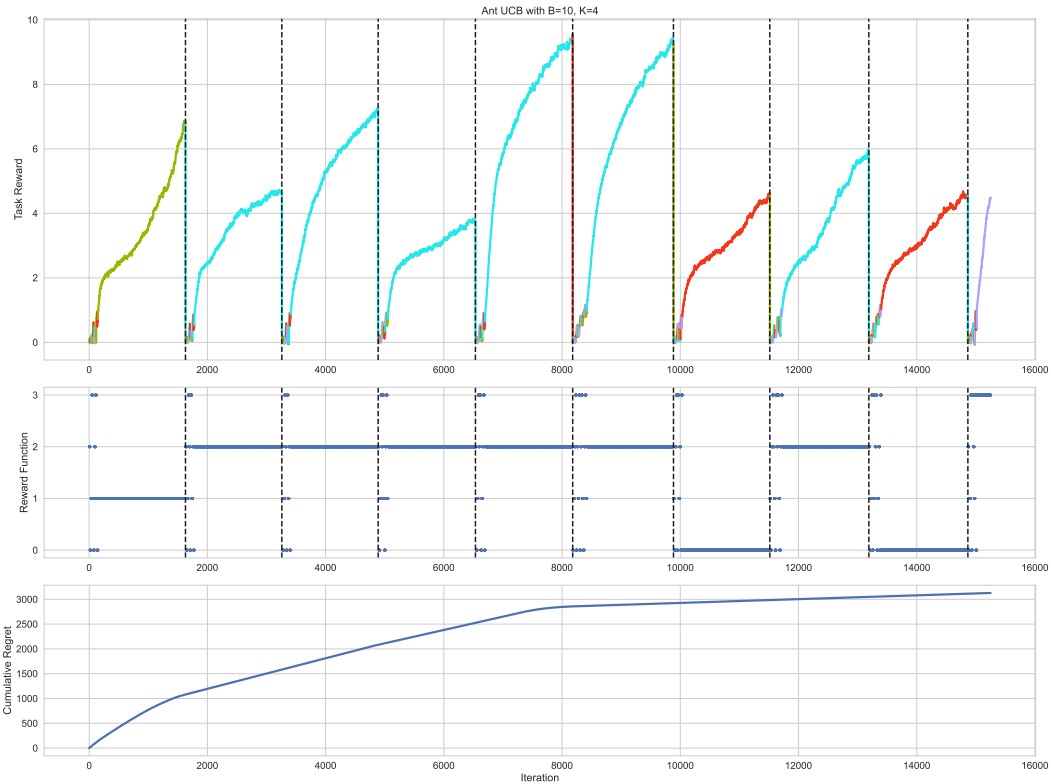

Figure 23: ORSO (UCB) on ANT with $B = 10$ and $K = 4$.

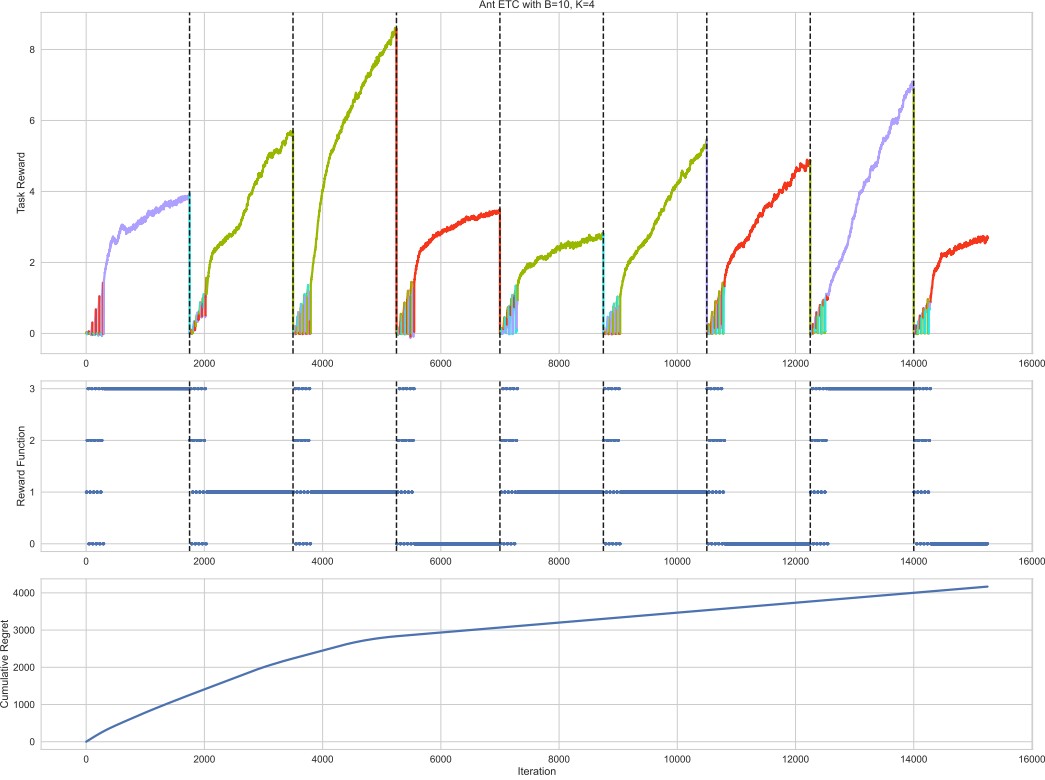

Figure 24: ORSO (ETC) on ANT with $B = 10$ and $K = 4$.

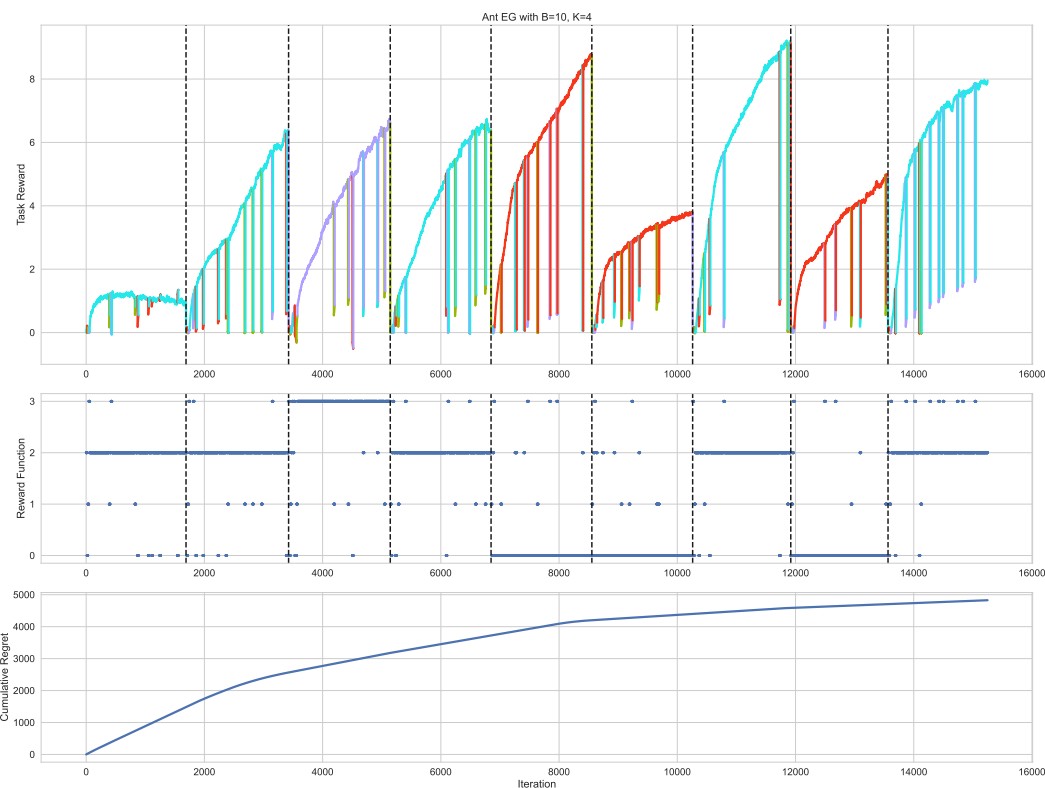

Figure 25: ORSO (EG) on ANT with $B = 10$ and $K = 4$.

