# OpenReview forum: "ORSO: Accelerating Reward Design via Online Reward Selection and Policy Optimization"
_ICLR.cc/2025/Conference — ICLR 2025 Poster_

### Official Review · Reviewer_YU7z · 2024-10-30

**Soundness:** 3
**Presentation:** 2
**Contribution:** 2
**Rating:** 3
**Confidence:** 3

**Summary:**

This manuscript proposes the Online Reward Selection and Policy Optimization (ORSO) to frame shaping reward selection as an online model selection problem. It automatically identifies promising shaping reward functions, balancing exploration and exploitation with provable regret guarantees. The ORSO method significantly improves sample efficiency and reduces computational time compared to traditional methods that fully evaluate each shaping reward function.

**Strengths:**

The idea is a little simple but effective.

**Weaknesses:**

Some experiments and theories can be added.

**Questions:**

1. The proposed method generates a set of candidate reward functions for the online selection phase. Does having K reward function candidates mean that the number of candidates is fixed? If all these candidates are not suitable or not the best, what is the solution?

2. The experiments compared the performance of policies trained using No Design, Human, Naive Selection, and ORSO to show the superiority of ORSO. However, the impact of selecting different reward functions for ORSO algorithm on experimental results has not been analyzed. If possible, please provide relevant experiments to demonstrate the experimental differences caused by selecting different reward functions.

3. Figure 4 shows the normalized cumulative regret for different selection algorithms. The manuscript mentioned that the ORSO’s regret can become negative, indicating that it finds reward functions that outperform the human baseline. The minimum value is zero in Figure 4, I didn’t observe the negative values.

4. There is a similar paper ORSO: Accelerating Reward Design via Online Reward Selection and Policy Optimization published in ARLET 2024. What is the difference between these two works? Has the ARLET paper been cited?

5. The number of references is small, and more recent articles on reward shaping can be added.

---

> ### Author Response · Authors · 2024-11-21
> **Addressing Weaknesses and Questions on Theoretical and Experimental Contributions (Part 1)**
>
> We thank the reviewer for the feedback. We address the points below.
>
> **Weaknesses**
>
> > Some experiments and theories can be added.
> >
>
> We believe the current manuscript provides a comprehensive theoretical framework and sufficient experimental validation. Theoretical contributions include the provable regret guarantees for reward selection, and our experiments span diverse tasks, ablations on different selection strategies, reward functions set size and budget. To help us better address your concern, could you please identify specific gaps in either our theoretical analysis or experimental validation that you feel need addressing? We would be happy to take the suggestions into consideration to improve the contribution of our work.
>
> **Questions**
> > The experiments compared the performance of policies trained using No Design, Human, Naive Selection, and ORSO to show the superiority of ORSO. However, the impact of selecting different reward functions for ORSO algorithm on experimental results has not been analyzed. If possible, please provide relevant experiments to demonstrate the experimental differences caused by selecting different reward functions.
> >
>
> Thank you for the suggestion to analyze different reward functions in detail. However, running full training on each reward function would be computationally prohibitive, which is precisely the challenge our method aims to address.
>
> To demonstrate the effectiveness of our approach, we provide results for the Ant task. Specifically, we trained a policy for each of the 96 reward functions used in Figure 5 (Figure 6 in the updated PDF). The table reports the mean task reward with 95% confidence intervals over five seeds. Rewards are ordered from best to worst, with those within one confidence interval of the best reward italicized (up to and including reward ID 54). Bolded values indicate the reward functions selected by ORSO + D3RB across the seeds we ran.
>
> Our results show that ORSO consistently selects reward functions within one confidence interval of the best reward. The occasional selection of different reward functions can be attributed to the inherent stochasticity in reinforcement learning training.
>
> | Reward ID | Value (+/- CI) |
> | --- | --- |
> | 34 | ***10.24 +/- 0.36*** |
> | 18 | *10.01 +/- 0.63* |
> | 71 | *9.98 +/- 0.37* |
> | 79 | *9.88 +/- 0.73* |
> | 21 | ***9.77 +/- 0.22*** |
> | 94 | *9.70 +/- 0.38* |
> | 66 | ***9.67 +/- 0.27*** |
> | 81 | ***9.55 +/- 0.80*** |
> | 70 | ***9.51 +/- 0.63*** |
> | 37 | *9.46 +/- 0.55* |
> | 33 | *9.34 +/- 0.83* |
> | 95 | *9.27 +/- 0.33* |
> | 47 | *9.24 +/- 0.68* |
> | 63 | *9.21 +/- 0.76* |
> | 54 | *9.20 +/- 0.80* |
> | 80 | 9.16 +/- 0.25 |
> | 62 | 8.88 +/- 0.37 |
> | 38 | 8.81 +/- 0.45 |
> | 49 | 8.81 +/- 0.77 |
> | 35 | 8.69 +/- 1.07 |
> | 5 | 8.61 +/- 0.56 |
> | 52 | 8.35 +/- 1.43 |
> | 67 | 8.32 +/- 0.85 |
> | 46 | 8.30 +/- 0.89 |
> | 68 | 8.20 +/- 1.22 |
> | 75 | 8.09 +/- 0.40 |
> | 84 | 8.05 +/- 1.25 |
> | 85 | 7.77 +/- 0.97 |
> | 72 | 7.64 +/- 1.27 |
> | 55 | 7.43 +/- 1.46 |
> | 20 | 7.26 +/- 0.18 |
> | 23 | 7.26 +/- 1.12 |
> | 86 | 7.15 +/- 0.42 |
> | 36 | 7.06 +/- 0.68 |
> | 91 | 6.93 +/- 1.45 |
> | 1 | 6.50 +/- 1.17 |
> | 31 | 6.36 +/- 0.80 |
> | 61 | 6.06 +/- 0.93 |
> | 19 | 5.78 +/- 1.43 |
> | 25 | 5.67 +/- 1.41 |
> | 48 | 5.65 +/- 1.54 |
> | 59 | 5.59 +/- 1.02 |
> | 26 | 5.50 +/- 0.89 |
> | 60 | 5.47 +/- 1.17 |
> | 44 | 5.47 +/- 1.36 |
> | 40 | 5.34 +/- 1.73 |
> | 73 | 5.33 +/- 1.77 |
> | 0 | 5.30 +/- 1.38 |
> | 56 | 5.05 +/- 0.62 |
> | 39 | 4.91 +/- 0.64 |
> | 74 | 4.88 +/- 0.49 |
> | 30 | 4.83 +/- 0.78 |
> | 78 | 4.83 +/- 0.35 |
> | 6 | 4.76 +/- 0.91 |
> | 2 | 4.69 +/- 0.92 |
> | 28 | 4.66 +/- 1.21 |
> | 8 | 4.65 +/- 0.44 |
> | 16 | 4.57 +/- 1.08 |
> | 29 | 4.53 +/- 0.81 |
> | 65 | 4.44 +/- 0.62 |
> | 50 | 4.23 +/- 1.94 |
> | 58 | 3.89 +/- 0.43 |
> | 53 | 3.86 +/- 0.44 |
> | 32 | 3.79 +/- 0.73 |
> | 22 | 3.74 +/- 0.54 |
> | 3 | 3.48 +/- 1.66 |
> | 69 | 3.22 +/- 0.42 |
> | 4 | 3.18 +/- 0.42 |
> | 88 | 3.18 +/- 0.43 |
> | 64 | 3.12 +/- 0.16 |
> | 9 | 3.11 +/- 0.39 |
> | 17 | 3.10 +/- 0.15 |
> | 93 | 3.02 +/- 0.21 |
> | 14 | 2.99 +/- 0.52 |
> | 45 | 2.89 +/- 0.29 |
> | 83 | 2.72 +/- 0.82 |
> | 27 | 2.50 +/- 0.72 |
> | 10 | 2.15 +/- 0.43 |
> | 57 | 1.69 +/- 0.80 |
> | 7 | 1.67 +/- 1.01 |
> | 82 | 1.03 +/- 0.35 |
> | 42 | 0.63 +/- 0.80 |
> | 41 | 0.37 +/- 0.30 |
> | 43 | 0.33 +/- 0.16 |
> | 76 | 0.22 +/- 0.08 |
> | 89 | 0.22 +/- 0.07 |
> | 24 | 0.21 +/- 0.03 |
> | 11 | 0.21 +/- 0.08 |
> | 12 | 0.19 +/- 0.14 |
> | 15 | 0.19 +/- 0.14 |
> | 92 | 0.14 +/- 0.07 |
> | 77 | 0.13 +/- 0.04 |
> | 13 | 0.05 +/- 0.00 |
> | 51 | 0.05 +/- 0.00 |
> | 87 | 0.05 +/- 0.02 |
> | 90 | 0.00 +/- 0.00 |

---

> ### Author Response · Authors · 2024-11-21
> **Addressing Weaknesses and Questions on Theoretical and Experimental Contributions (Part 2)**
>
> **Questions**
>
> - Q1
>
>     > The proposed method generates a set of candidate reward functions for the online selection phase. Does having K reward function candidates mean that the number of candidates is fixed? If all these candidates are not suitable or not the best, what is the solution?
>     >
>
>     This is a great question. Our method includes an iterative improvement step that allows for refinement of the reward functions (proposal of new candidates). Furthermore, our experimental results demonstrate that even with a fixed set of reward functions, ORSO with D3RB performs well, given the reward function set is large enough (our Ant experiments in Section 5.3 show that with K=48 or K=96 can still outperform human-design performance). The large candidate set makes it highly probable to include effective reward functions.
>
> - Q3
>
>     > Figure 4 shows the normalized cumulative regret for different selection algorithms. The manuscript mentioned that the ORSO’s regret can become negative, indicating that it finds reward functions that outperform the human baseline. The minimum value is zero in Figure 4, I didn’t observe the negative values.
>     >
>
>     Thank you for this observation. The **instantaneous regret** can indeed become negative when ORSO identifies reward functions that outperform the human baseline. This causes the **cumulative regret** to decrease, but it does not necessarily drop below zero in the normalized cumulative regret plots.
>
> - Q5
>
>     > The number of references is small, and more recent articles on reward shaping can be added.
>     >
>
>     Thank you for this suggestion. While the manuscript includes foundational and relevant works, we will expand the related work section to include additional recent papers on reward shaping. If there are specific references the reviewer believes are important, we welcome the input.
>
>
> We hope that our clarifications and the proposed additional experiments sufficiently address the reviewer’s questions and concerns. Given the theoretical results, comprehensive experimental evaluation, and practical relevance of ORSO, we believe our work makes a meaningful contribution to the field of reward design in reinforcement learning. We kindly ask the reviewer to reconsider their evaluation of the manuscript. Thank you for your thoughtful consideration.

---

> > ### Comment · Reviewer_YU7z · 2024-11-23
> >
> > Thanks for the authors' reply. I think it would be complicated to train a lot of reward candidates (e.g. K=96), and switching reward functions during training would be unstable. I suggest considering regression to get new rewards instead of selection.

---

> ### Author Response · Authors · 2024-11-24
>
> We thank the reviewer for the quick response. We addressed the instability in choosing among many reward functions in the response to Reviewer DEh7's comments (https://openreview.net/forum?id=0uRc3CfJIQ&noteId=GdpudWUVqQ). Moreover, we show in our experiments with K=48 and K=96 that this instability does not occur. The additional table provided in the review above serves to show that ORSO indeed selects reward functions within one CI of the optimal one.
>
> Regarding the suggestion to consider regression for obtaining new rewards instead of selection, we appreciate the idea but would like to better understand the intended approach.
>
> We would also ask that if the previous comments clarified the reviewer’s questions, they would consider increasing the score accordingly. Thank you again for your time and feedback.

---

> ### Comment · Reviewer_YU7z · 2024-11-27
>
> Thanks for the authors' response. Many of my concerns have been addressed.
>
> Regarding the "regression" approach, I note from Appendix E.1 that most reward functions created by LLMs are parsing the observation space to extract various pieces of information, designing reward components, and then summing them with weights. If all candidate reward functions are of similar form, rather than having the LLM generate K reward functions and simultaneously maintaining K policies for selection, I wonder if it would be more flexible to let the LLM/human only specify the components and then learn the corresponding weights (a zero weight indicates the absence of a reward component). This is just an idea of mine and not a request for the authors to implement during the rebuttal period.
>
> After carefully reading the revised paper and other reviewers' comments, I decided to maintain my score for the following reasons:
>
> 1. The LLM-generated reward functions have certain limitations and lack controllability. The LLM-generated reward functions are constrained by the LLM's understanding of the environment or the prior knowledge provided by humans. As shown in the examples from Appendix E.1, the LLM appears to have a clear understanding of what each element in the observation space represents. However, in many real-world environments, such as those with image-based observations, such detailed understanding is difficult to obtain. This limitation restricts the generalizability of ORSO.
>
> 2. In my view, the contribution of the paper is somewhat incremental. The method for generating candidate reward functions closely follows existing work (e.g., EUREKA). Similarly, the selection strategies employed are also existing algorithms (e.g., ETC, EG, UCB, EXP3, and D3RB). Simply combining these two parts and demonstrating improved performance over EUREKA without selection strategies does not constitute a sufficiently novel contribution.
>
> 3. The abstract claims that "While shaping rewards have been introduced to provide additional guidance, selecting effective shaping functions remains challenging and computationally expensive." However, this overlooks many methods that do not require explicit shaping function selection. Numerous algorithms exist that automate the generation of reliable reward models. Although the experiments in the paper explore the importance of reward selection and compare various selection strategies, the primary objective of the paper is *reward design*. To fully demonstrate its effectiveness, comparisons with SOTA (within the last five years) automated reward shaping methods would be necessary.
>
> Given these considerations, I have decided to maintain my score. I would like to thank the authors again for their response.

---

> > ### Author Response · Authors · 2024-11-27
> >
> > > The abstract claims that "While shaping rewards have been introduced to provide additional guidance, selecting effective shaping functions remains challenging and computationally expensive." However, this overlooks many methods that do not require explicit shaping function selection. Numerous algorithms exist that automate the generation of reliable reward models. Although the experiments in the paper explore the importance of reward selection and compare various selection strategies, the primary objective of the paper is *reward design*. To fully demonstrate its effectiveness, comparisons with SOTA automated reward shaping methods would be necessary.
> > >
> >
> > We note that we have compared with LIRPG in a previous comment [here](https://openreview.net/forum?id=0uRc3CfJIQ&noteId=U9RaoKtiXV), which is a widely adopted method. Our results show that ORSO with an LLM generator can significantly outperform this baseline. We redirect the reviewer to the linked response to avoid additional repetitions on the thread of responses.
> >
> > We hope our comments clarify the questions and concerns raised by the reviewer. We would be more than happy to further engage in discussion and clarify any remaining questions. We truly believe that the efficiency gains of ORSO can be useful for the broader research community, in particular for those who do not have large computational resources.

---

> > > ### Author Response · Authors · 2024-12-03
> > >
> > > Dear reviewer,
> > >
> > > as the discussion period is coming to an end, we would be grateful for any further feedback on our earlier response. Please let us know if you have any questions or need clarification. Otherwise, we kindly ask you to consider adjusting your score.
> > >
> > > We truly value your insights and participation in the review process.
> > >
> > > Kind regards,
> > >
> > > The authors

---

> ### Author Response · Authors · 2024-11-27
>
> We thank the reviewer for their response. We would like to address the concerns raised by the reviewer.
>
> > Regarding the "regression" approach, I note from Appendix E.1 that most reward functions created by LLMs are parsing the observation space to extract various pieces of information, designing reward components, and then summing them with weights. If all candidate reward functions are of similar form, rather than having the LLM generate K reward functions and simultaneously maintaining K policies for selection, I wonder if it would be more flexible to let the LLM/human only specify the components and then learn the corresponding weights (a zero weight indicates the absence of a reward component). This is just an idea of mine and not a request for the authors to implement during the rebuttal period.
> >
>
> We thank the reviewer for clarifying the regression approach. This is indeed an interesting approach for settings in which the possible reward components are known and the candidate rewards are all in the same form and would be exciting to explore as a future direction. However, we would like to note that one might not know all possible reward components or listing all possible components might not be feasible as there might be numerous possible transformations applied to, for example, the distance component. We further elaborate on this point [here](https://openreview.net/forum?id=0uRc3CfJIQ&noteId=Z0mSV85ycC).
>
> > The LLM-generated reward functions have certain limitations and lack controllability. The LLM-generated reward functions are constrained by the LLM's understanding of the environment or the prior knowledge provided by humans. As shown in the examples from Appendix E.1, the LLM appears to have a clear understanding of what each element in the observation space represents. However, in many real-world environments, such as those with image-based observations, such detailed understanding is difficult to obtain. This limitation restricts the generalizability of ORSO.
> >
>
> We agree that LLM-based generation is not the answer to reward generation in general. However, we would like to point out that many real-world applications [1, 2, 3] are non-image-state-based. Moreover, as the reviewer pointed out, generation is not the main contribution of our paper, which is the only part of the algorithm that requires access to the environment code. The ORSO framework can be applied to any form of reward function. The choice of using LLMs to generate reward functions is motivated by the success of such methods in EUREKA and other similar works.
>
> > In my view, the contribution of the paper is somewhat incremental. The method for generating candidate reward functions closely follows existing work (e.g., EUREKA). Similarly, the selection strategies employed are also existing algorithms (e.g., ETC, EG, UCB, EXP3, and D3RB). Simply combining these two parts and demonstrating improved performance over EUREKA without selection strategies does not constitute a sufficiently novel contribution.
> >
>
> While ORSO is similar to EUREKA (Ma et al., 2024), our work extends beyond a simple modification. In Figure 3 of the updated PDF, we show that ORSO has significant (up to **16x fewer GPUs needed** to achieve the same performance in the same time) gains in terms of necessary compute compared to EUREKA. All our experiments can be indeed run on a single commercial GPU (e.g., a 3090 Ti or even a 2080 Ti). This will open the possibility for researchers with smaller computational budgets to quickly iterate their experiments.
>
> Moreover, the connection between model selection techniques and reward design in RL has not been explore before and our experimental results show that we gain in efficiency, while maintaining effectiveness. We provide a novel analysis of D3RB, which is in stark contrast with the regret guarantees of the original paper. Namely, our guarantee depend on the true regret coefficients rather than the monotonic ones.
>
> **References**
>
> [1] Margolis, Gabriel B., et al. "Rapid locomotion via reinforcement learning." *The International Journal of Robotics Research* 43.4 (2024): 572-587.
>
> [2] Margolis, Gabriel B., and Pulkit Agrawal. "Walk these ways: Tuning robot control for generalization with multiplicity of behavior." *Conference on Robot Learning*. PMLR, 2023.
>
> [3] Lee, Joonho, et al. "Learning quadrupedal locomotion over challenging terrain." *Science robotics* 5.47 (2020): eabc5986.

---

### Official Review · Reviewer_DEh7 · 2024-10-30

**Soundness:** 2
**Presentation:** 3
**Contribution:** 2
**Rating:** 5
**Confidence:** 4

**Summary:**

The paper proposed an Online Reward Selection and Policy Optimization (ORSO) algorithm for reinforcement learning. ORSO pre-generates some candidate reward functions by linearly combining some reward components or by LLM, while learning, ORSO dynamically evaluates which candidate reward function can lead to better policy optimization, then selects the optimal candidate to guide the learning process.

**Strengths:**

The approach is easy to implement and effective at selecting reward functions, it also shows fast convergence in terms of both computational time and sample efficiency.

**Weaknesses:**

1. As this method follows a "pre-define and select one" paradigm, the final optimal performance that ORSO can achieve heavily depends on how good are the pre-generated candidate reward functions.
2. The author states that this is a reward shaping approach, but the paper doesn't compare it with any reward shaping or reward selection baselines. If the authors could compare ORSO with some representative reward shaping algorithms (such as [1][2][3][4]), it would better showcase its advantages.

[1] Devidze, Rati, Parameswaran Kamalaruban, and Adish Singla. "Exploration-guided reward shaping for reinforcement learning under sparse rewards." Advances in Neural Information Processing Systems 35 (2022): 5829-5842.

[2] Zheng, Zeyu, Junhyuk Oh, and Satinder Singh. "On learning intrinsic rewards for policy gradient methods." Advances in Neural Information Processing Systems 31 (2018).

[3] Memarian, Farzan, et al. "Self-supervised online reward shaping in sparse-reward environments." 2021 IEEE/RSJ International Conference on Intelligent Robots and Systems (IROS). IEEE, 2021.

[4] Burda, Y., Edwards, H., Storkey, A., and Klimov, O. (2018). Exploration by random network distillation. In International Conference on Learning Representations.

**Questions:**

1. Regarding Weakness 1, about candidate reward function generation, As described in  section 5.1.2, the candidate reward functions are directly generated through LLM, Could you clarify:

(a) What are the specific forms of these reward functions? Are they related to the observations, features, and/or pre-defined reward components? Can these generated rewards capture all the necessary aspects to define effective rewards?

(b) If the optimal reward function is not included in the generated candidates, how does ORSO ensure that the final optimized policy is indeed optimal?

2. the policy optimizes based on a given candidate reward function, would this make it difficult to ensure that the policy is optimizing the task's original objective (the environmental reward function defined by the MDP)?

3. Frequent switching of reward functions may lead to significant shifts in the policy's learning objectives. For instance, in a maze task, if reward function #1 focuses on avoiding obstacles, while reward function #2 focuses on resource collection, switching between these two may lead to inconsistent learning targets. Would this cause instability in the learning process?

4. I'm unclear about the evaluation metric in the experiments, specifically, in Figure 2 (left), it shows performance as a percentage of the human-designed reward function. In Section 5.1.1, the paper states, "No design is with the task reward function r for each MDP". I assume this refers to the original environmental reward function, which should be the primary objective the agent aims to optimize. However, in Figure 2 (left), the "No design" baseline is around half of the human-designed reward (I assume this figure reports cumulative rewards under each baseline's own reward function). This seems unfair and could introduce bias for deviating from the MDP’s original task objective.

For example, suppose the MDP provides rewards of $0, 0, 1$ for states $s_1, s_2, s_3$ (only 3 states). A human-designed reward function might assign $0, 1, 1$ for $s_1, s_2, s_3$. Consequently, the cumulative reward under the human-designed reward function would be higher, and it also proposes new targets (both $s_2$ and $s_3$ are equally important). From my understanding, the performance should be evaluated consistently on the original MDP reward (the true objective), meaning that the "No design" case should actually serve as an upper bound.

---

> ### Author Response · Authors · 2024-11-21
> **Reward Shaping Comparison, Reward Function Generation, and Evaluation Clarifications (Part 1)**
>
> We thank the reviewer for the feedback and the suggestions to improve the paper. We address the comments below.
>
> - W2
>
>     > The author states that this is a reward shaping approach, but the paper doesn't compare it with any reward shaping or reward selection baselines. If the authors could compare ORSO with some representative reward shaping algorithms (such as [1][2][3][4]), it would better showcase its advantages.
>     >
>
>     These papers collectively discuss various approaches to reward shaping and intrinsic motivation in reinforcement learning, with a focus on improving learning efficiency in sparse-reward environments through methods like exploration-guided rewards (ExploRS), learning intrinsic rewards, self-supervised reward shaping, and random network distillation (RND).
>
>     We note that methods like [1] are complementary to ORSO, meaning that one can use ORSO to propose shaped reward functions and then apply other shaping methods on top to further improve the performance of such reward functions.
>
>     We chose to compare ORSO-designed reward functions with LIRPG because it is one of the most widely adopted methods for reward design in reinforcement learning. We provide experimental results with LIRPG on the Ant task. LIRPG jointly trains a policy and learns an intrinsic shaping reward, such that the intrinsic reward leads to higher extrinsic reward.
>
>     In each experiment, we use the task reward function, the human-designed reward function, and the reward function selected by ORSO as the extrinsic reward for LIRPG, respectively. We run each experiment for 5 random seeds and report the mean base environmental reward achieved by training with each method, along with 95% confidence intervals.
>
>     | Method | Without LIRPG | With LIRPG |
>     | --- | --- | --- |
>     | No Design | 4.67 +/- 0.84 | **5.73 +/- 1.08** |
>     | Human | **9.84 +/- 0.30** | 10.02 +/- 0.30 (*) |
>     | ORSO | **11.09 +/- 0.68** | 11.51 +/- 0.45 (*) |
>     - **LIRPG cannot design better rewards than ORSO**: When LIRPG is applied to the task reward, it results in lower performance compared to using the human-designed or ORSO-selected rewards.
>     - **LIRPG as a complementary method**: We emphasize that LIRPG can complement ORSO. By applying LIRPG to reward functions selected by ORSO (which have already undergone shaping), LIRPG may help learn an additional function that aids the agent in optimizing ORSO-designed rewards.
>
>     The bolded entries have undergone one stage of reward shaping, while the entries in the table above marked with (*) have undergone two stages of reward design. First a performant reward function was obtained from a human designed or ORSO (both outperforming LIRPG on the task reward). Then, given the good quality of the reward, we show that we can apply LIRPG on such reward functions to some marginal improvement. We only provide such results for completeness. We note that the evaluation of each policy is done with respect to the task reward function (No Design).
>
>     **LIRPG does not help if the extrinsic reward function is too sparse.** We also test LIRPG on sparse-reward manipulation tasks, such as the Allegro Hand. However, LIRPG does not provide any improvement over the environmental reward as the reward function is “too sparse.” This agrees with the experimental results in Figure 7 of [1], where the authors show that increasing the sparsity of the feedback (every 10, 20, or 40 steps) can decrease the performance of LIRPG.
>
> - 1 (a)
>
>     > What are the specific forms of these reward functions? Are they related to the observations, features, and/or pre-defined reward components? Can these generated rewards capture all the necessary aspects to define effective rewards?
>     >
>
>     The reward functions are generated as Python code. The input of the reward functions are environment observations and agent actions. We report some reward functions in the Appendix of the updated PDF file.
>
> - 1 (b)
>
>     > If the optimal reward function is not included in the generated candidates, how does ORSO ensure that the final optimized policy is indeed optimal?
>     >
>
>     The regret guarantee we provide is with respect to the optimal reward function in the set of candidate functions. While it is true that if the set does not contain an optimal one, ORSO would not achieve high task reward, we find that in practice this rarely happens. Practically, in order to find a performant reward functions, we need sampling and iterative improvement. Thanks to ORSO’s efficiency, we can explore a large set of functions within a limited budget, which leads to a higher probability of sampling the optimal reward function from the generator.

---

> ### Author Response · Authors · 2024-11-21
> **Reward Shaping Comparison, Reward Function Generation, and Evaluation Clarifications (Part 2)**
>
> 1. Q1
>
>     > The policy optimizes based on a given candidate reward function, would this make it difficult to ensure that the policy is optimizing the task's original objective (the environmental reward function defined by the MDP)?
>     >
>
>     This is correct. If the candidate reward function does not optimize the for original environmental reward, the learned policy would not be optimal with respect to the original task. Reward candidates that, when optimized for, lead to low original environmental reward are discarded by our selection algorithm and those that lead to higher environmental reward will be chosen more often. Reward design involves aligning auxiliary reward functions with the task reward, especially when the task reward is sparse or difficult to optimize. This is typical in robotics [2, 3], cybersecurity [4] and more. Approaches like potential reward shaping [5] can provably preserve optimality, however, they have been shown to not work very well in practice [6], however, as shown in the additional LIRPG experiments above, such methods can help improve base reward functions marginally.
>
> 2. Q2
>
>     > Frequent switching of reward functions may lead to significant shifts in the policy's learning objectives. For instance, in a maze task, if reward function #1 focuses on avoiding obstacles, while reward function #2 focuses on resource collection, switching between these two may lead to inconsistent learning targets. Would this cause instability in the learning process?
>     >
>
>     In the implementation, this behavior does not occur because a separate policy is instantiated for each reward function, as detailed in Algorithm 1, line 2. Consequently, each policy is updated only as frequently as its corresponding reward function is selected.
>
> 3. Q3
>
>     > I'm unclear about the evaluation metric in the experiments, specifically, in Figure 2 (left), it shows performance as a percentage of the human-designed reward function. In Section 5.1.1, the paper states, "No design is with the task reward function r for each MDP". I assume this refers to the original environmental reward function, which should be the primary objective the agent aims to optimize. However, in Figure 2 (left), the "No design" baseline is around half of the human-designed reward (I assume this figure reports cumulative rewards under each baseline's own reward function). This seems unfair and could introduce bias for deviating from the MDP’s original task objective.
>     >
>
>     All evaluations in our experiments are with respect to the MDP’s task reward $r$. Because the human-designed reward functions were specifically designed so that training on those would improve task reward, we normalize result such that the policy of human designed reward functions is at 1.0. The “No Design” line being lower than the “Human” line means that training with the original task reward achieves lower performance than training with the human-designed reward function as measured by the original environmental reward function. This is exactly the objective of reward design: we aim to find reward functions that, when optimized, will lead to better performance with respect to the original task reward (because the designed reward is more amenable to optimization). In Figure 2 (left), the "No design" baseline — using the task reward function alone — achieves approximately half the performance of the human-designed reward. This demonstrates the difficulty of optimizing directly for the task reward and highlights the benefit of effective reward design in facilitating better optimization.
>
> - Regarding the example MDP, we would like to reiterate that the evaluation is done with respect to the the original task reward. Therefore, if one were to evaluate using the human-designed reward function (0, 1, 1), an agent that equally visits the states 2 and 3 would be optimal but would not be optimal if we evaluate using the task reward (0, 0, 1). Therefore, in our evaluation, the human-designed reward function would not be considered a “good” one. On the other hand, a reward function of the form (0, 1, 2) would be “good” as it provides more guidance compared to (0, 0, 1) and the optimal strategy is still to visit state 3 as often as possible. This is more obvious if we consider an MDP with $n$ states $s_1, \ldots, s_n$ with task reward function (0, 0, …, 0, 1), i.e., zero everywhere, except for the n-th state. This reward function is clearly sparse and hard to optimize. A good human-designed reward function could look like (1/n, 2/n, …, (n-1)/n, n), which still leads the agent towards the n-th state but provides more guidance during training.

---

> ### Author Response · Authors · 2024-11-21
> **Reward Shaping Comparison, Reward Function Generation, and Evaluation Clarifications (Part 3)**
>
> We hope these clarifications address the concerns raised. If clarifications and additional experiments resolve your concerns, we would be grateful if you could consider raising your score to reflect the improvements.
>
> **References**
>
> [1] Zheng, Zeyu, Junhyuk Oh, and Satinder Singh. "On learning intrinsic rewards for policy gradient methods." Advances in Neural Information Processing Systems 31 (2018).
>
> [2] Cheng, Xuxin, et al. "Expressive whole-body control for humanoid robots." *arXiv preprint arXiv:2402.16796* (2024).
>
> [3] Margolis, Gabriel B., and Pulkit Agrawal. "Walk these ways: Tuning robot control for generalization with multiplicity of behavior." *Conference on Robot Learning*. PMLR, 2023.
>
> [4] Bates, Elizabeth, Vasilios Mavroudis, and Chris Hicks. "Reward Shaping for Happier Autonomous Cyber Security Agents." *Proceedings of the 16th ACM Workshop on Artificial Intelligence and Security*. 2023.
>
> [5] Ng, A. Y., Harada, D., & Russell, S. (1999). *Policy invariance under reward transformations: Theory and application to reward shaping.* In ICML.
>
> [6] Cheng, Ching-An, Andrey Kolobov, and Adith Swaminathan. "Heuristic-guided reinforcement learning." *Advances in Neural Information Processing Systems* 34 (2021): 13550-13563.

---

> ### Comment · Reviewer_DEh7 · 2024-11-25
>
> Thanks for the authors' response.
>
> I have some questions want to discuss:
>
> 1. regarding Q1(a), the specific forms of these reward functions, thanks for updating the detailed reward functions in the appendix. I have two main concerns:
> (a) seems the reward function needs much task-specific knowledge, like:
> ```
> velocity = root_states[:, 7:10] # [vx, vy, vz]
> torso_position = root_states[:, 0:3] # [px, py, pz]
> ```
> how does LLM know this kind of information (like which dimensions refer to which metric) or it is given in the prompts? If we still need to provide this prior knowledge to LLM, not sure why not directly use a human-designed reward function. For example, if we already know the `[7:10]` represents the velocity, and suppose my final target is to let the ant be as fast as possible, then why not just write out this reward function?
>
> Moreover, I see the final reward is the weight-sum of different components:
> ```
> reward = forward_reward + balance_reward + angle_penalty - action_penalty_scaled + survival_bonus
> ```
> can just learn some weights of different components, instead of "generating a lot of candidate reward functions and choose one from them"?
>
> (b) How about some more complex environments, like Atari? given the pixel observations, does it still work to generate python reward functions?
>
> 2. Regarding the author's claim:
>
> > This is more obvious if we consider an MDP with n states $s_1,\dots,s_n$ with task reward function (0, 0, …, 0, 1), i.e., zero everywhere, except for the n-th state. This reward function is clearly sparse and hard to optimize. A good human-designed reward function could look like (1/n, 2/n, …, (n-1)/n, n), which still leads the agent towards the n-th state but provides more guidance during training.
>
> But this still leads to non-consistent optimal policies, let's take a three-state example: $s_1,s_2,s_3$, and two reward schemes: $R_1: s_1 =0, s_2=0, s_3=1$, and $R_2: s_1 =1/3, s_2=2/3, s_3=3/3=1$ as the authors' claim. Then we consider the following trajectory: $t_1: s_1 \rightarrow s_2 \rightarrow s_3$, which we know clearly is the optimal trajectory under the environmental rewards. Then compute the returns under two reward schemes ($\gamma=0.9$):
>
> $$
> return_1(t_1) = 0 + 0.9 * 0 + 0.9^2 * 1 = 0.81
> $$
>
> $$
> return_2(t_1) = 1/3 + 0.9 * 2/3 + 0.9^2 * 1 \approx 1.71
> $$
>
> while we further consider the trajectory: $t_2: s_1 \rightarrow s_2 \rightarrow s_2 \rightarrow s_3$, then:
>
> $$
> return_1(t_2) = 0 + 0.9 * 0 + 0.9^2 0 + 0.9^3 * 1 = 0.729
> $$
>
> $$
> return_2(t_2) = 1/3 + 0.9 * 2/3 + 0.9^2 * 2/3 + 0.9^3 * 1 \approx 2.17
> $$
>
> We can see: $return_2(t_2) > return_2(t_1)$ but $return_1(t_2) < return_1(t_1)$, which means under the human-designed reward scheme, the optimal policy changes (actually, staying at $s_2$ longer can get higher returns). This is where my main concerns from, in the LLM-generated reward functions, could it be possible the reward functions make the agent stay in some local optimum and then lead to sub-optimal or non-consistent policies?
>
> 3. Regarding the novelty and contribution, from the papers' statements at around Line 309, and to my knowledge, the OSRA mainly modified the EUREKA (Ma et al., 2024) from selecting and testing one-by-one to initializing all candidates and testing in parallel, which is not a big improvement, but naturally trading space for time.
>
> 4. Lastly, I encourage the authors to compare ORSO with at least one more related baseline, as for now, the study of comparing ORSO with different sources of reward functions is good enough, but no other auto-reward-function-generation algorithms are compared, they also show good performance on sparse-reward envs and reward design.

---

> > ### Author Response · Authors · 2024-11-27
> >
> > # Part 1/2
> >
> > We thank the reviewer for the additional comments and thinking points. We address the concerns below.
> >
> > > (a) seems the reward function needs much task-specific knowledge. […] how does LLM know this kind of information (like which dimensions refer to which metric) or it is given in the prompts?
> > >
> >
> > We appreciate the reviewer's comment. Regarding the necessary task-specific knowledge, we do not manually provide this information in the prompt, but we do provide the observation space. While the LLM finds that some coordinates correspond to velocity and position, there might be a lot of candidates where this is not the case.
> >
> > As noted, while our approach leverages LLMs to generate candidate reward functions, we do not claim this as the primary contribution. Instead, our focus is on an efficient selection process that can handle diverse reward function candidates.
> >
> > > can just learn some weights of different components, instead of "generating a lot of candidate reward functions and choose one from them"?
> > >
> >
> > While weight learning is a valid approach, knowing what reward function components to include often non-trivial. Take the Ant environment example:
> >
> > ```jsx
> > height_diff = torch.abs(torso_position[:, 2] - target_height)
> > ```
> >
> > It is not clear whether one should use absolute difference, squared difference, or another metric. This still has to be decided and designed.
> >
> > > (b) How about some more complex environments, like Atari? given the pixel observations, does it still work to generate python reward functions?
> > >
> >
> > We acknowledge the current limitation with pixel-based environments. Our method is most effective in state-based environments where reward function generation is more straightforward.
> >
> > However, we emphasize that our core contribution is the ORSO selection process. Regardless of how reward functions are generated—whether through LLMs, domain experts, or other methods—ORSO provides an efficient mechanism for identifying the most promising candidates.
> >
> > > This is where my main concerns from, in the LLM-generated reward functions, could it be possible the reward functions make the agent stay in some local optimum and then lead to sub-optimal or non-consistent policies?
> > >
> >
> > This is a valid concern. Suboptimal reward functions can indeed emerge and there is no guarantee that LLM generated reward functions are optimal with respect to the task reward.
> >
> > However, interestingly, a "suboptimal" reward function (i.e., its global optimum is not the same as the task reward functions) might still outperform a theoretically optimal one (the task reward function) if the latter is challenging to optimize. Many existing works [1, 2, 3, 4, 5, 6, 7] use reward shaping without optimality guarantee with respect to the base reward function, but achieve impressive performance in real-world applications.
> >
> > Our approach provides a systematic way to explore and validate different reward formulations and we empirically show that with enough sampling and the iterative improvement step, one can obtain reward functions that when train with will outperform policies trained directly with the task reward function (with performance measure with the task reward function).
> >
> > Regarding the 3-state MDP example, we would like to note that the setting we considered is the infinite horizon MDP. Therefore, a better comparison would be between $t_ 1 : s_1 \to s_2 \to s_3 \to s_3 \to \dots$ and $t_2 : s_1 \to s_2 \to s_2 \to s_3 \to \dots$, where we assume that $s_3$ is an absorbing state. In this case, one would have consistent results between the human-designed and the task reward functions. Either way, the evaluation is performed with the base reward function as specified in the previous response, so this problem would not arise as a reward function that highly rewards the agent when trained with, but leads to low task reward would be discarded.
> >
> > **References**
> >
> > [1] Liu, Minghuan, et al. "Visual whole-body control for legged loco-manipulation." *arXiv preprint arXiv:2403.16967* (2024).
> >
> > [2] Margolis, Gabriel B., et al. "Rapid locomotion via reinforcement learning." *The International Journal of Robotics Research* 43.4 (2024): 572-587.
> >
> > [3] Margolis, Gabriel B., and Pulkit Agrawal. "Walk these ways: Tuning robot control for generalization with multiplicity of behavior." *Conference on Robot Learning*. PMLR, 2023.
> >
> > [4] Lee, Joonho, et al. "Learning quadrupedal locomotion over challenging terrain." *Science robotics* 5.47 (2020): eabc5986.
> >
> > [5] Ma, Yecheng Jason, et al. "Eureka: Human-level reward design via coding large language models." *arXiv preprint arXiv:2310.12931* (2023).
> >
> > [6] Ha, Huy, et al. "Umi on legs: Making manipulation policies mobile with manipulation-centric whole-body controllers." *arXiv preprint arXiv:2407.10353* (2024).
> >
> > [7] Kaufmann, Elia, et al. "Champion-level drone racing using deep reinforcement learning." *Nature* 620.7976 (2023): 982-987.

---

> > > ### Author Response · Authors · 2024-11-27
> > >
> > > # Part 2/2
> > >
> > > > Regarding the novelty and contribution, from the papers' statements at around Line 309, and my basic understanding, the OSRA mainly modified the EUREKA (Ma et al., 2024) from selecting and testing one-by-one to initializing all candidates and testing in parallel, which is not a big improvement, but naturally trading space for time.
> > > >
> > >
> > > While ORSO is similar to EUREKA (Ma et al., 2024), our work extends beyond a simple modification. In Figure 3 of the updated PDF, we show that ORSO has significant (up to **16x fewer GPUs needed** to achieve the same performance in the same time) gains in terms of necessary compute compared to EUREKA. The connection between model selection techniques and reward design in RL has not been explore before and our experimental results show that we gain in efficiency, while maintaining effectiveness.
> > >
> > > > Lastly, I encourage the authors to compare ORSO with at least one more related baseline, as for now, the study of comparing ORSO with different sources of reward functions is good enough, but no other auto-reward-function-generation algorithms are compared, they also show good performance on sparse-reward envs and reward design.
> > > >
> > >
> > > We appreciate the suggestion and are committed to enhancing the experimental section. We are open to and happy to incorporate additional baseline comparisons in the final version. We note that we have compared with LIRPG (https://openreview.net/forum?id=0uRc3CfJIQ&noteId=U9RaoKtiXV), which is a widely adopted method. Our results show that ORSO with an LLM generator can significantly outperform this baseline.
> > >
> > > We want to emphasize that our primary goal is not to propose a new reward generation algorithm, but to introduce a more efficient and effective method of reward function selection.
> > >
> > > We hope our comments address the reviewer’s comments. We are happy to engage in further discussions and clarify any remaining questions.

---

> > > > ### Comment · Reviewer_DEh7 · 2024-11-27
> > > >
> > > > Thanks to the authors for providing detailed responses to my follow-up questions and offering further clarification. I believe the key point emphasized by the authors is that ORSO focuses on reward selection rather than reward generation. However, clearly defining the scope of problems that can be addressed (e.g., robotics with available environmental descriptions/codes) based on the source of reward generation remains important.
> > > >
> > > > Additionally, in terms of reward selection, ORSO still relies on existing model selection strategies (Line 4 of Algorithm 1), and its performance is strongly dependent on the specific model selection algorithm employed. Given the limited contribution of simply applying various existing model selection algorithms to the domain of reward model selection, I will maintain my current score (below the acceptance threshold).

---

> > > > > ### Author Response · Authors · 2024-11-27
> > > > >
> > > > > We thank the reviewer for the additional comments and feedback. We however respectfully disagree with some of the statements.
> > > > >
> > > > > > Thanks to the authors for providing detailed responses to my follow-up questions and offering further clarification. I believe the key point emphasized by the authors is that ORSO focuses on reward selection rather than reward generation. However, clearly defining the scope of problems that can be addressed (e.g., robotics with available environmental descriptions/codes) based on the source of reward generation remains important.
> > > > > >
> > > > >
> > > > > We note that the only component that requires access to the environment code is the generation process with LLMs. As the reviewer highlighted (”I believe the key point emphasized by the authors is that ORSO focuses on reward selection rather than reward generation.”), ORSO mainly focuses on effective and efficient selection. This part does not require access to the code. Indeed, selection can be applied to any form of reward function as is also stated in Section 3.1, “Reward Generation” of the paper.
> > > > >
> > > > > > Additionally, in terms of reward selection, ORSO still relies on existing model selection strategies (Line 4 of Algorithm 1), and its performance is strongly dependent on the specific model selection algorithm employed. Given the limited contribution of simply applying various existing model selection algorithms to the domain of reward model selection, I will maintain my current score (below the acceptance threshold).
> > > > > >
> > > > >
> > > > > We disagree that the contribution of ORSO is limited. As stated in our previous response,  we show that ORSO has significant (up to **16x fewer GPUs needed** to achieve the same performance in the same time) gains in terms of necessary compute compared to EUREKA. Because all our experiments can be run on a single commercial GPU (e.g., a 3090 Ti or even a 2080 Ti), this will open the possibility for researchers with smaller computational budgets to quickly iterate their experiments, which we believe to be an important contribution.
> > > > >
> > > > > Moreover, the connection between model selection techniques and reward design in RL has not been explore before and our experimental results show that we gain in efficiency, while maintaining effectiveness. We provide a novel analysis of D3RB, which is in stark contrast with the regret guarantees of the original paper. Namely, our guarantee depend on the true regret coefficients rather than the monotonic ones.

---

> > > > > > ### Author Response · Authors · 2024-12-03
> > > > > >
> > > > > > Dear reviewer,
> > > > > >
> > > > > > as the discussion period is coming to an end, we would be grateful for any further feedback on our earlier response. Please let us know if you have any questions or need clarification. Otherwise, we kindly ask you to consider adjusting your score.
> > > > > >
> > > > > > We truly value your insights and participation in the review process.
> > > > > >
> > > > > > Kind regards,
> > > > > >
> > > > > > The authors

---

### Official Review · Reviewer_C5vj · 2024-11-02

**Soundness:** 3
**Presentation:** 3
**Contribution:** 3
**Rating:** 8
**Confidence:** 4

**Summary:**

This paper introduces ORSO, a method that aims to increase the efficiency of the reward shaping selection process by casting it as an automated online model selection problem.
ORSO is a general algorithmic framework that implements the following steps: (i) a reward function generation method is used to provide a set of candidate shaping reward functions, (ii) a selection algorithm is used to select a shaping reward function, (iii) an RL algorithm is used to train the policy associated with the selected shaping reward function for a set amount of iterations, (iv) the trained policy is then evaluated against the task reward function and the utility is used to update the parameters of the selection algorithm. This process is repeated until a predefined computational budget is exhausted.
While the components within the ORSO framework are modular and exchangeable, this work uses (i) an LLM based generator as the reward function generation method, (ii) PPO as the RL algorithm, (iii) D3RB, as the reward function selection algorithm (ablations are additionally conducted with Exp3, D3RB,UCB, ETC, and EG).
Experiments are conducted across tasks of varying difficulty, 3 budgets, and 3 reward functions sets.
Results indicate that the ORSO performance in terms of task return scales with budget, and is comparable - and can surpass - that of human defined shaping reward functions; additionally ORSO is twice as fast as a prior shaping reward function selection method (EUREKA).

**Strengths:**

Claims: (1)  ORSO reduces both time and computational costs by more than half compared to earlier methods, making reward design accessible to a wider range of researchers. (2) By formalizing the reward design problem and providing a theoretical analysis of ORSO’s regret when using the D3RB algorithm, we also contribute to the theoretical understanding of reward design in RL.

- The work proposes an original formulation of the shaping reward function selection process, viewing it as an online model selection problem.
- The regret based analysis provides a clear and intuitive way to monitor ORSO's performance and efficiency gains.
- Thanks to the problem formulation, the method is kept elegant in its simplicity and largely amounts to the application of existing approaches into a unified framework.
- Results on a variety of tasks and domains against available baselines support the efficiency and performance claims.

Clarity:
- The writing is clear, the paper is well structured, and appropriate context is provided to the reader.

Significance:
- This work tackles the issue of reward design for RL. This has been and continues to be one of the most significant challenges keeping RL from widespread successful deployment in the real world.

**Weaknesses:**

Contribution:
- Ultimately, ORSO searches over a set of shaping reward functions. While the framework is simple and elegant, to my understanding, it ultimately relies on and is limited by the performance of the "shaping reward function generator".
- ORSO is only benchmarked against methods for which a performant human-engineered reward function can be defined. Impact would be higher if method could generalize beyond these settings.
- ORSO in its current form does not seem to offer the flexibility to deal with changing / annealing of shaping reward functions throughout training, a common technique in reward shaping literature.

**Questions:**

- Unique contributions could be made clearer and explicitly called out in the introduction.

- Lines 54-59: I understand you are highlighting unique challenges compared to standard multi-arm bandit settings, yet ORSO uses the same selection algorithms typically used to solve such multi-arm bandit problems. The paper could be clearer in defining exactly which components of ORSO are key to address the unique challenges presented.

- I recommend expanding on the various resampling strategies, if more than one was tried out, and their impact on performance as this seems to be a key ingredient to the method's success.

- I would recommend adding the synthetically generated best-performing shaping reward functions for each task to appendix E. Are the reward functions sensible to the human reader? This has implications on how well these shaping reward functions could be further refined by human experimenters, and possibly give insights on their logical soundness.

- Also, was any constraint, structure, human knowledge beyond the imposed when prompting the generation of such rewards or could the prompt be arguably generated programmatically (if so, I recommend just stating it - the code base is not available during the review process to verify)?  While not directly related to the ORSO contribution, this is arguably important to showcase as ORSO heavily relies on the existence of an automated way of generating reward functions without human priors.

- Please look at the weaknesses section and help clarify if any are can be addressed.

---

> ### Author Response · Authors · 2024-11-21
> **Clarifications on ORSO (Part 1)**
>
> We thank the reviewer for the comments and the insightful feedback provided. We address the points below.
>
> **Weaknesses**
>
> - W1
>
>     > Ultimately, ORSO searches over a set of shaping reward functions. While the framework is simple and elegant, to my understanding, it ultimately relies on and is limited by the performance of the "shaping reward function generator".
>     >
>
>     While it is true that ORSO’s performance relies on the set of reward functions given by the generator, our practical implementation includes an iterative improvement resampling step that allows the algorithms to improve the set of candidate reward functions. An alternative generator could be devised. For instance, if the required reward components are known, an alternative generator could sample different weights for the reward components from a uniform distribution over plausible ranges. For example, let the reward function \(r(s, a)\) be a weighted sum of known components \(r_1(s, a), r_2(s, a), \ldots, r_k(s, a)\), with weights \(\boldsymbol{w} = [w_1, w_2, \ldots, w_k]\). ORSO could then operate on this sampled set (our experiments show that ORSO works well even with large reward sets). Then the results could be used to update a posterior distribution over the parameter weights. Repeating this process would allow the framework to explore any reward function space defined by its components.
>
> - W2
>
>     > ORSO is only benchmarked against methods for which a performant human-engineered reward function can be defined. Impact would be higher if method could generalize beyond these settings.
>     >
>
>     We would love to test ORSO on tasks where no human-designed reward function is available. If the reviewer has specific continuous control tasks and implementation in mind, we would be happy to explore them. However, this may fall outside the discussion timeframe, though we could incorporate these suggestions in future work.
>
> - W3
>
>     > ORSO in its current form does not seem to offer the flexibility to deal with changing / annealing of shaping reward functions throughout training, a common technique in reward shaping literature.
>     >
>
>     While we have not implemented shaping reward annealing in this version, this feature could be easily added. Since the selection algorithms track the frequency of each reward function being chosen, shaping rewards could be dynamically re-weighted, e.g., inversely proportional to the selection count. We encourage the reviewer to check our codebase following the publication of ORSO, as we plan to incorporate such variants to enhance the framework’s utility for the community.

---

> ### Author Response · Authors · 2024-11-21
> **Clarifications on ORSO (Part 2)**
>
> **Questions**
>
> - Q1
>
>     > Lines 54-59: I understand you are highlighting unique challenges compared to standard multi-arm bandit settings, yet ORSO uses the same selection algorithms typically used to solve such multi-arm bandit problems. The paper could be clearer in defining exactly which components of ORSO are key to address the unique challenges presented.
>     >
>
>     The main difficulty of using classical MAB algorithms lies in the non-stationarity and statefulness of each learner (candidate reward function and corresponding policy) in ORSO. Algorithms such as UCB provide convergence guarantees if the utility of each arm is stationary. However, it is clear that as one trains using a reward function the task reward achieved by the corresponding policy will change. We integrate algorithms for online model selection (D3RB), which are designed to deal with stateful learners, into our ORSO implementation and show that such methods indeed overcome some of the problems encountered by MAB algorithms (e.g., they tend to commit to seemingly optimal reward functions early on and not exploring reward other reward functions enough). Therefore, the key component for successful selection is D3RB, which can deal with stateful learners.
>
> - Q2
>
>     > I recommend expanding on the various resampling strategies, if more than one was tried out, and their impact on performance as this seems to be a key ingredient to the method's success.
>     >
>
>     We have reported the resampling strategy in the Appendix, where we also discuss the pros and cons of different resampling strategies. We do not have have statistically significant evidence that the resampling strategy in our experiments is optimal, nor do we claim so. However, we observed in some early experiments that the chosen strategy (we resample one half of the new set of reward functions by conditioning the reward generation on the code of the previous best reward function and the other half is generated from scratch as if it was the first time the reward function set was generated) was a good tradeoff between greedily improving the best reward function for all new rewards and always resampling from scratch.
>
> - Q3
>
>     > I would recommend adding the synthetically generated best-performing shaping reward functions for each task to appendix E. Are the reward functions sensible to the human reader? This has implications on how well these shaping reward functions could be further refined by human experimenters, and possibly give insights on their logical soundness.
>     >
>
>     This is a great suggestion. We have reported the code for the best reward functions in the Appendix in the updated PDF.
>
> - Q4
>
>     > Also, was any constraint, structure, human knowledge beyond the imposed when prompting the generation of such rewards or could the prompt be arguably generated programmatically (if so, I recommend just stating it - the code base is not available during the review process to verify)? While not directly related to the ORSO contribution, this is arguably important to showcase as ORSO heavily relies on the existence of an automated way of generating reward functions without human priors.
>     >
>
>     In our experiments, we do not add any particular human knowledge in addition to simply exposing the observation space and some useful environment variables available in the environment definition. We will make sure to clarify this in the manuscript. We would also like to add that if human prior is available (e.g., we might know which reward components are important for quadruped locomotion — xy velocity tracking, yaw velocity tracking, action penalty, body height, etc. — but do not know how to weigh them), we can use a Bayesian way to update the distribution of each coefficient as detailed in the response to the first Weaknesses bullet point. This would indeed be an interesting future experiment

---

> ### Author Response · Authors · 2024-11-26
>
> Dear Reviewer,
>
> We sincerely appreciate your thoughtful and positive feedback on our paper. Your insights were incredibly helpful in refining our work. We hope that our responses to your comments have addressed your concerns and clarified the contributions of our study.
>
> If there are any remaining questions or areas where you feel further clarification is needed, we would be happy to provide additional details or engage in further discussion. We hope that our responses demonstrate the merits of the paper, and we kindly ask if you would consider revisiting your evaluation in light of these updates.
>
> Thank you again for your time and effort in reviewing our submission. We greatly value your perspective and look forward to your final decision.
>
> Best regards,
>
> The Authors

---

> > ### Comment · Reviewer_C5vj · 2024-11-27
> >
> > Dear authors,
> > Thanks for thoroughly addressing my feedback. I am satisfied with the responses and with the proposed modifications. I will update my score accordingly.

---

> > > ### Author Response · Authors · 2024-11-27
> > >
> > > We thank the reviewer for their thoughtful feedback and for revisiting their evaluation of our work. We greatly appreciate the updated score and your confidence in our submission. Your comments and suggestions have been invaluable in improving the clarity and quality of our paper.

---

### Official Review · Reviewer_knxR · 2024-11-03

**Soundness:** 3
**Presentation:** 4
**Contribution:** 3
**Rating:** 8
**Confidence:** 4

**Summary:**

This paper studies automated reward shaping by posing it as an online reward selection problem. Instead of multiple training runs to observe the impact of different shaping functions, this work aims to identify the best one within a fixed time budget. To this aim, the authors develop ORSO, a regret-minimizing approach that utilizes multi-armed bandits where a candidate shaping function is an arm. More specifically, ORSO uses the D3RB algorithm to select an arm. Upon selection of an arm, ORSO trains a policy corresponding to the said arm for a fixed number of iterations and then evaluates the policy with respect to the task rewards. The paper provides regret guarantees, assuming that a learner monotonically dominates all learners and its average performance increases monotonically.

The paper evaluates a practical implementation of ORSO in continuous control tasks with varying complexity whose rewards are either sparse or unshaped. The experimental results show that ORSO is faster than an LLM-based reward design method, can surpass human-designed rewards, and performs better as the budget increases. The authors also provide an ablation study for different arm selection strategies and different numbers of candidate shaping functions.

**Strengths:**

- The paper is very well-written. The motivation is clearly explained, and the problem and assumptions are well described. Moreover, given the assumptions, the proposed approach and its theoretical guarantees are clear.

- The experimental set-up makes sense, and the evaluated baselines allow us to see how reward design is critical, humans can be sub-optimal at it, and naive attempts are prone to fail.

- The experimental results clearly showcase its advantages in comparison to the baselines. In addition, the ablation study evaluates the impact of different components of ORSO and provides detailed insights.

**Weaknesses:**

- I urge the authors to move the related work, at least the most relevant parts, to the main document.

- Assumption 4.2 seems limiting. A discussion of why the assumptions are viable or how they are needed would strengthen the paper's arguments. It would be even better to explain their role in causing the contrast with the regret guarantees in Pacchiano et al. (2023).

- As the quality of candidate shaping functions plays an important role, an ablation study to understand the impact of wrong/redundant candidates would help the reader understand the limitations of ORSO.

**Questions:**

- In what cases would the monotonicity assumption be violated? Do the environments in the experimental set-up violate or obey the assumption? How would ORSO handle such violations?

- Future work mentions exciting directions. Since the naive approach is failing, how likely is a VLM-based reward design method to fail?

---

> ### Author Response · Authors · 2024-11-21
> **Clarifications and Revisions Based on Feedback**
>
> We thank the reviewer for their thorough review. Your detailed understanding of our work is greatly appreciated, as reflected in your summary and questions. We address your points below:
>
> 1. Related Work
>
>     We agree with moving the related work section to the main paper. This will help better contextualize our contributions. The revised PDF contains some related work in the main paper and a more thorough treatment of relevant literature in the appendix.
>
> 2. Assumption 4.2 and Monotonicity
>     - In our experimental settings, we observed that monotonicity holds in most cases
>     - The few violations we observed (visible in Figures 17 and 18) occurred with alternative selection strategies, not with D3RB
>     - These cases likely represent situations where the selection strategy committed to a suboptimal reward function, while the optimal reward function still exhibited monotonic behavior
>     - The convergence result presented in our manuscript (dependence on true regret coefficient) is different from the one presented in the original D3RB paper (dependence on the monotonic regret coefficient). The intuition behind the guarantees provided in our paper is that the true regret coefficient dependence implies that even if the optimal reward function has a “slow start”, it can still be selected and trained on and will achieve similar regret to running only the optimal.
> 3. Impact of Wrong/Redundant Reward Functions
>     - While a direct analysis of wrong/redundant reward functions would be ideal, it would be computationally prohibitive given our large search space
>     - However, our analysis of varying K (number of reward functions) serves as a useful proxy:
>         - The results in the appendix show ORSO's robustness to K given sufficient budget
>         - On the other hand
>             - With small K, naive selection can evolve quickly but risks converging to and evolving suboptimal rewards
>             - With large K, naive selection may explore more options and allow evolution to find optimal rewards, however, this will lead the naive selection algorithm to spend significant amount of time and compute on suboptimal reward functions initially
> 4. VLM-based Reward Design
>
>     In some initial experiments, we tested the possibility of using VLMs to evaluate the behavior of trained agents and remove the need to the manually specified task reward function. We however observed that, at the time of the experiments, VLMs struggled to evaluate behaviors correctly and would hallucinate most of the time. We believe that as VLMs improve, this can be an exciting direction to explore.
>
>
> We thank the reviewer again for their careful reading and constructive feedback that has helped us identify areas where we can strengthen the paper's presentation and discussion. We incorporate these clarifications in the revised version.

---

> > ### Comment · Reviewer_knxR · 2024-11-25
> >
> > Thank you for your response. The clarifications on the assumption 4.2 and whether monotonicity holds in experimented environments are helpful.
> >
> > Extending the discussions in the main document about these clarifications and the impact of wrong/redundant reward functions would improve the presentation.
> >
> > I will keep my score as is towards acceptance.

---

> > > ### Author Response · Authors · 2024-11-25
> > >
> > > We thank the reviewer for their thorough review and valuable feedback. We are glad the clarifications regarding Assumption 4.2 and the monotonicity in the experimented environments were helpful.
> > >
> > > We agree that extending the discussion on these points, as well as addressing the impact of incorrect or redundant reward functions, would enhance the presentation of the paper. We will incorporate these improvements in the final version.
> > >
> > > Thank you again for your constructive comments and for supporting the paper’s acceptance.

---

### Official Review · Reviewer_t4Xg · 2024-11-04

**Soundness:** 3
**Presentation:** 3
**Contribution:** 3
**Rating:** 6
**Confidence:** 4

**Summary:**

This paper introduces Online Reward Selection and Policy Optimization (ORSO), an approach that defines reward selection as an online model selection problem. The approach uses exploration strategies to identify shaping reward functions.

**Strengths:**

Soundness
======
The approach is generally sound.

Significance & Related work
=========
The paper presents an in-depth related work section, and well defined preliminaries (note redundancy of section 2)  that lead to demonstrations of several results.

Experimentation
=========
The paper presents an in-depth ablation analysis of the performance of various selection algorithms.

Presentation
=========
The paper is well written.

**Weaknesses:**

Soundness
======
It is unclear in the experiments in Fig 2 what ‘human-level performance’ or ‘human-designed reward function’ is and how it is defined/computed. Note that the proof for D1 needs to be rewritten for clarity to show base case and inductive hypothesis, should proof by induction still be the chosen approach.

Experimentation
=========
The paper presents an in-depth ablation analysis of the performance of various selection algorithms, however, the impact of poorly chosen task rewards needs to be analysed.

Presentation
=========
Presentation is good, as above, Section 2 is too short and redundant.

**Questions:**

* What is human designed reward and how it is computed?

---

> ### Author Response · Authors · 2024-11-21
> **Clarifications on Rewards, Proofs, and Presentation**
>
> We thank the reviewer for their thoughtful feedback and for highlighting the strengths of our work, including the soundness of our approach, the in-depth related work section, and the detailed ablation analysis. Below, we address the specific questions and concerns raised.
>
> - **Human-Designed Reward Function**
>
>     The term *human-designed reward function* refers to reward functions manually created by domain experts who implemented the environments used in our experiments. These experts designed the rewards to reflect task objectives based on their domain knowledge. Details on how each reward function (task reward and human-designed reward) is defined can be found in Appendix E (Reward Functions Definitions). We note that the plots always plot the task reward. When **Human** is indicated, it means “performance of a policy trained with the human-designed reward function on the task reward.”
>
> - **Clarification on Proof for Lemma D1**
>
>     We appreciate the reviewer pointing out that the proof for D1 could benefit from additional clarity. We have updated the proof of Lemma D1 with the following structure in the updated PDF.
>
>     **Base Case ($t=1$)**
>
>     At $t=1$, for all algorithms $i \in [K]$:
>
>     - $\widehat{d}^i_1 = d_{\min}$ (by initialization)
>     - $n_1^i = 1$ if $i$ is the first algorithm chosen, 0 otherwise
>     - Therefore $n_1^i \leq n_1^{i_\star} + 1$ holds
>
>     **Inductive Step**
>
>     Inductive Hypothesis: Assume that for some $t \geq 1$:
>
>     - $\widehat{d}^{i_\star}_{t-1} = d_{\min}$
>     - $n_{t-1}^i \leq n_{t-1}^{i_\star} + 1$ for all $i \in [K]$
>
>     We need to show these properties hold for $t$. Let $i_t = i_\star$. When $\mathcal{E}$ holds, the LHS of D$^3$RB's misspecification test satisfies [proof as is].
>
>     Combining these inequalities shows the misspecification test will not trigger, thus:
>
>     1. $\widehat{d}^{i_\star}_t$ remains at $d_{\min}$
>     2. For all $i \in [K]$, $n_t^i \leq n_t^{i_\star} + 1$ continues to hold
>
>     This finalizes the proof.
>
> - **Impact of Poorly Chosen Task Rewards**
>
>     Regarding the concern about analyzing the impact of poorly chosen task rewards: this is fundamentally a task specification problem [1]. In any reinforcement learning framework, a misspecified task reward — whether used with ORSO or another method — will inevitably lead to optimization towards unintended objectives. While this highlights a broader challenge in reward design, it is not unique to ORSO.
>
> - **Presentation**
>
>     We agree that Section 2 might be a bit redundant for RL experts. We provided this section to help non-experts with the necessary notation. We provide more a more thorough presentation of online model selection preliminaries in the appendix.
>
>
> We hope these changes, with the strengths the reviewer already highlighted will increase your confidence in our work. If these revisions resolve your concerns, we would be grateful if you could consider raising your score to reflect the improvements. We appreciate your time and constructive feedback, which has helped us refine our submission.
>
> **References**
>
> [1] Agrawal, Pulkit. "The task specification problem." *Conference on Robot Learning*. PMLR, 2022.

---

> ### Author Response · Authors · 2024-11-26
>
> Dear Reviewer,
>
> We sincerely appreciate your thoughtful and positive feedback on our paper. Your insights were incredibly helpful in refining our work. We hope that our responses to your comments have addressed your concerns and clarified the contributions of our study.
>
> If there are any remaining questions or areas where you feel further clarification is needed, we would be happy to provide additional details or engage in further discussion. We hope that our responses demonstrate the merits of the paper, and we kindly ask if you would consider revisiting your evaluation in light of these updates.
>
> Thank you again for your time and effort in reviewing our submission. We greatly value your perspective and look forward to your final decision.
>
> Best regards,
>
> The Authors

---

> > ### Comment · Reviewer_t4Xg · 2024-11-27
> > **Thank you for your response**
> >
> > I thank the authors for their comments and revision. I will retain my score.

---

> > > ### Author Response · Authors · 2024-11-27
> > >
> > > We thank the reviewer once again for their valuable feedback and constructive suggestions, and we are happy to hear their score indicates support for accepting our paper.

---

### Official Review · Reviewer_sCCd · 2024-11-14

**Soundness:** 2
**Presentation:** 3
**Contribution:** 2
**Rating:** 5
**Confidence:** 4

**Summary:**

The paper employs the data-driven online model selection algorithm D^3RB (Pacchiano et al., 2023) to choose between candidate reward functions for reinforcement learning, where these candidates are generated through an LLM. It replaces the naive reward selection from Ma et al. (2023) with D^3RB, enabling more efficient online selection among candidate rewards. A simple example shows how D^3RB helps prevent budget exhaustion by avoiding over-allocation to a single option. The paper further evaluates the algorithm’s effectiveness on Isaac Gym (Makoviychuk et al., 2021), comparing baseline and human-designed rewards. Ablations also explore various bandit exploration strategies, including UCB, EXP3, ETC, EG, and Naive.

**References**

- Pacchiano, A., Dann, C., & Gentile, C. (2023). *Data-driven regret balancing for online model selection in bandits.* arXiv preprint arXiv:2306.02869.

- Ma, Y. J., Liang, W., Wang, G., Huang, D.A., Bastani, O., Jayaraman, D., Zhu, Y., Fan, L., & Anandkumar, A. (2023). *Eureka: Human-level reward design via coding large language models.* arXiv preprint arXiv:2310.12931.

- Makoviychuk, V., Wawrzyniak, L., Guo, Y., Lu, M., Storey, K., Macklin, M., Hoeller, D., Rudin, N., Allshire, A., & Handa, A., et al. (2021). *Isaac Gym: High performance GPU-based physics simulation for robot learning.* arXiv preprint arXiv:2108.10470.

**Strengths:**

The paper addresses a significant issue in reinforcement learning, as sparse rewards indeed pose challenges, and reward shaping can provide dense feedback, potentially accelerating learning. The writing is clear, with well-presented graphs and experiments, and a well-defined motivation. The paper also successfully applies a recent online model selection algorithm, demonstrating in the proposed benchmark how the algorithm can aid in selecting the appropriate reward functions with limited budget and can scale effectively with increased budget and candidate reward functions.

**Weaknesses:**

**Writing**: In Section 3.1, ORSO is presented as an effective and efficient method for reward function design. However, my understanding is that ORSO simply selects a reward function from a set of candidates, with the reward generation handled separately. If this is correct, it would be helpful if the paper clearly defined ORSO’s components, specifying whether reward generation is part of ORSO or assumed to be user-provided. Clarification here would improve understanding.

**Complexity**: The proposed method requires maintaining $K$ separate policies for each reward, which could create substantial memory overhead. Standard approaches in reward shaping often aim to achieve similar benefits while maintaining only a single policy, highlighting a potential drawback in scalability.

**Performance Guarantees**: Traditional reward shaping studies typically ensure that the shaped reward aligns with a base reward function. Here, there is no guarantee that the proposed objective aligns with the base reward, leaving the outcome quality solely dependent on the reward generation process. This could raise concerns about consistency in performance.

**Experiments**:

1. The paper does not compare with established reward shaping methods, such as those by Ng et al. (1999), Zou et al. (2019), Zheng et al. (2018), Sorg et al. (2010), and Gupta et al. (2023). Including these comparisons would strengthen the experimental evaluation.

2. Iterative resampling appears to be essential for obtaining high-quality reward functions, as it involves reinitializing policies and refining rewards over iterations. However, the paper lacks discussion on the resampling process's challenges, frequency, and impact on performance. Additionally, it is unclear if resampling is an integral part of ORSO or a separate process.

3. In the experiments, performance is evaluated against human-designed rewards, but the actual evaluation should ideally be based on the baseline reward. This raises questions about the metrics used for evaluation. In Figure 2, the upper bounds should correspond to **No Design**, as the objective should be to assess performance against the MDP’s base reward.

4. The reward generation process seems to require access to code-level details of the MDP, which may not be feasible in cases where the environment is not code-based. Discussion of this limitation would improve transparency regarding the method’s applicability.


**References**

- Ng, A. Y., Harada, D., & Russell, S. (1999). *Policy invariance under reward transformations: Theory and application to reward shaping.* In ICML.

- Zou, H., Ren, T., Yan, D., Su, H., & Zhu, J. (2019). *Reward shaping via meta-learning.* arXiv preprint arXiv:1901.09330.

- Zheng, Z., Oh, J., & Singh, S. (2018). *On learning intrinsic rewards for policy gradient methods.* Advances in Neural Information Processing Systems.

- Sorg, J., Lewis, R. L., & Singh, S. (2010). *Reward design via online gradient ascent.* Advances in Neural Information Processing Systems.

- Gupta, D., Chandak, Y., Jordan, S. M., Thomas, P. S., & da Silva, B. C. (2023). *Behavior Alignment via Reward Function Optimization.* arXiv preprint arXiv:2310.19007.

**Questions:**

The weaknesses section raised some key questions, which I’ll summarize here for clarity:

1. ORSO functions only as a selection algorithm, correct? Reward generation isn’t part of the algorithm itself?

2. Does the paper simply apply the D^3RB algorithm, or does it introduce theoretical improvements? This aspect is somewhat unclear.

3. As I understand it, there’s no guarantee that the optimal policy obtained with the selected reward aligns with the optimal policies for the base reward (i.e., **No Design**), correct?

4. Could you clarify the evaluation metrics? The ideal benchmark should be based on the base reward, as that’s ultimately the reward we aim to optimize.

5. The terminology around “effective budget” seems confusing. Based on the proposed algorithms, the effective budget for environment interactions should be $TN$ rather than $T$, since each iteration assumes running the algorithm for at least $N$ steps to yield a final policy. Could you clarify this? This also seems to affect Figure 1—does preferred reward selection occur after $N$ iterations or a single one? As I understand it, each reward would at-least have to be evaluated $N$ times, hence making a minimum budget of $KN$, right?

---

> ### Author Response · Authors · 2024-11-21
> **Clarifications and Additional Insights on ORSO's Reward Generation, Complexity, and Experimental Results (Part 1)**
>
> We thank the reviewer for the comments and suggestions. We clarify the weaknesses and questions below.
>
> **Writing**
>
> The reward generation can be seen as both part of the ORSO or not. If one has a set of generated reward functions, then we can use ORSO to simply jointly select a performant reward function and train a policy with it. On the other hand, if there is no reward function provided beforehand, one can also see the generation as part of ORSO. Algorithm 4 provides the full pseudo-code with generation and iterative improvement of the reward function, where a new set of reward function is generated when the most selected reward function has been used to train for a maximum number of iterations.
>
> **Complexity**
>
> It is true that we need to maintain K separate policies. In our setting, the policies being simple MLPs, it does not create much overhead. We note that the naive approach (given a set of reward functions, train a policy until convergence for each reward function and then pick the best policy) still needs to instantiate K policies.
>
> **Performance Guarantees**
>
> > How do you ensure consistency in performance when there is no guarantee that the proposed objective aligns with the base reward function?
> >
>
> Our performance guarantees are with respect to the set of reward functions used for selection. While potential reward shaping could be applied in ORSO to ensure optimal policy invariance guarantees, we did not incorporate this because empirical evidence suggests that potential shaping often does not improve performance substantially in practice (as shown in [1]). Specifically, applying potential shaping to human-designed rewards frequently leads to suboptimal outcomes. Instead, we focus on an online selection process that chooses the reward function to train on based on its performance on the base reward function, ensuring that the chosen reward function maximizes the base reward. Empirically, we observe that with a sufficiently diverse set of candidate rewards, it is likely that one will perform similarly to or better than a human-designed reward when evaluated using the original task reward.
>
> **Experiments**
>
> - E1
>
>     > The paper does not compare with established reward shaping methods, such as those by Ng et al. (1999), Zou et al. (2019), Zheng et al. (2018), Sorg et al. (2010), and Gupta et al. (2023). Including these comparisons would strengthen the experimental evaluation.
>     >
>
>     We note that methods like [2] are complementary to ORSO, meaning that one can use ORSO to propose shaped reward functions and then apply other shaping methods on top to further improve the performance of such reward functions.
>
>     We chose to compare ORSO-designed reward functions with LIRPG because it is one of the most widely adopted methods for reward design in reinforcement learning. We provide experimental results with LIRPG on the Ant task. LIRPG jointly trains a policy and learns an intrinsic shaping reward, such that the intrinsic reward leads to higher extrinsic reward.
>
>     In each experiment, we use the task reward function, the human-designed reward function, and the reward function selected by ORSO as the extrinsic reward for LIRPG, respectively. We run each experiment for 5 random seeds and report the mean base environmental reward achieved by training with each method, along with 95% confidence intervals.
>
>     | Method | Without LIRPG | With LIRPG |
>     | --- | --- | --- |
>     | No Design | 4.67 +/- 0.84 | **5.73 +/- 1.08** |
>     | Human | **9.84 +/- 0.30** | 10.02 +/- 0.30 (*) |
>     | ORSO | **11.09 +/- 0.68** | 11.51 +/- 0.45 (*) |
>     - **LIRPG cannot design better rewards than ORSO**: When LIRPG is applied to the task reward, it results in lower performance compared to using the human-designed or ORSO-selected rewards.
>     - **LIRPG as a complementary method**: We emphasize that LIRPG can complement ORSO. By applying LIRPG to reward functions selected by ORSO (which have already undergone shaping), LIRPG may help learn an additional function that aids the agent in optimizing ORSO-designed rewards.
>
>     The bolded entries have undergone one stage of reward shaping, while the entries in the table above marked with (*) have undergone two stages of reward design. First a performant reward function was obtained from a human designed or ORSO (both outperforming LIRPG on the task reward). Then, given the good quality of the reward, we show that we can apply LIRPG on such reward functions to some marginal improvement. We only provide such results for completeness. We note that the evaluation of each policy is done with respect to the task reward function (No Design).

---

> ### Author Response · Authors · 2024-11-21
> **Clarifications and Additional Insights on ORSO's Reward Generation, Complexity, and Experimental Results (Part 2)**
>
> (continued response)
>     **LIRPG does not help if the extrinsic reward function is too sparse.** We also test LIRPG on sparse-reward manipulation tasks, such as the Allegro Hand. However, LIRPG does not provide any improvement over the environmental reward as the reward function is “too sparse.” This agrees with the experimental results in Figure 7 of [1], where the authors show that increasing the sparsity of the feedback (every 10, 20, or 40 steps) can decrease the performance of LIRPG.
>
> - E2
>
>     > Iterative resampling appears to be essential for obtaining high-quality reward functions, as it involves reinitializing policies and refining rewards over iterations. However, the paper lacks discussion on the resampling process's challenges, frequency, and impact on performance. Additionally, it is unclear if resampling is an integral part of ORSO or a separate process.
>     >
>
>     The details of the iterative resampling process and its frequency are discussed in Appendix F.
>
> - E3
>
>     > In the experiments, performance is evaluated against human-designed rewards, but the actual evaluation should ideally be based on the baseline reward. This raises questions about the metrics used for evaluation. In Figure 2, the upper bounds should correspond to **No Design**, as the objective should be to assess performance against the MDP’s base reward.
>     >
>
>     We would like to clarify that the evaluation is always based on the original task reward. The human-designed reward functions are heavily engineered reward functions, such that policies trained on them will achieve high original task reward (indeed, human-designed reward functions are strict improvements over the original task rewards). This is the reason for the red line being higher than the gray line. the question Figure 2 is answering is “How long does it take to design reward functions that perform as well as human-designed reward functions when evaluated with the original task reward?” Directly optimizing for the “No Design” reward does not achieve optimal policies because these reward can be sparse of poorly shaped, leading to a particularly hard optimization problem.
>
> - E4
>
>     > The reward generation process seems to require access to code-level details of the MDP, which may not be feasible in cases where the environment is not code-based. Discussion of this limitation would improve transparency regarding the method’s applicability.
>     >
>
>     This is correct. The generator used in our work requires a simulator for the environment and its code, which is common in robot learning, the application analyzed in the experiment in this paper. As mentioned above, the generation can be seen as both part of ORSO or not. If one is working with a non-code based environment, multiple reward functions can still be instantiated and ORSO can be applied to such pre-defined set of reward functions.
>
>
> **Questions**
>
> - Q1
>
>     > ORSO functions only as a selection algorithm, correct? Reward generation isn’t part of the algorithm itself?
>     >
>
>     As state above, the two components can be decoupled. When using the iterative improvement step, the generation can be seen as part of ORSO. Instead, if one already has a set of candidate rewards, ORSO can be used on the fixed set directly.

---

> ### Author Response · Authors · 2024-11-21
> **Clarifications and Additional Insights on ORSO's Reward Generation, Complexity, and Experimental Results (Part 3)**
>
> - Q2
>
>     > Does the paper simply apply the D^3RB algorithm, or does it introduce theoretical improvements? This aspect is somewhat unclear.
>     >
>
>     We use the D3RB algorithm, for which we present guarantees under different assumptions. The convergence result presented in our manuscript (dependence on true regret coefficient) is different from the one presented in the original D3RB paper (dependence on the monotonic regret coefficient). The intuition behind the guarantees provided in our paper is that the true regret coefficient dependence implies that even if the optimal reward function has a “slow start”, it can still be selected and trained on and will achieve similar regret to running only the optimal.
>
> - Q3 + Q4
>
>     > As I understand it, there’s no guarantee that the optimal policy obtained with the selected reward aligns with the optimal policies for the base reward (i.e., **No Design**), correct?
>     >
>
>     > Could you clarify the evaluation metrics? The ideal benchmark should be based on the base reward, as that’s ultimately the reward we aim to optimize.
>     >
>
>     Yes, while there is no guarantee that the policies learned with the selected shaping reward perfectly align with the optimal policy for the task reward (base reward / No Design), our approach explicitly selects shaping rewards that maximize the task reward, and our experiments show significant performance gains over the No Design baseline. The “selection reward” in our framework measures the task reward achieved by each candidate reward function, ensuring that the selected shaping reward leads to improvements in the original task reward. As a clarification, the task reward is always used as the evaluation metric to assess the quality of policies and their corresponding candidate rewards.
>
> - Q5
>
>     > The terminology around “effective budget” seems confusing. Based on the proposed algorithms, the effective budget for environment interactions should be TN rather than T, since each iteration assumes running the algorithm for at least N steps to yield a final policy. Could you clarify this? This also seems to affect Figure 1—does preferred reward selection occur after N iterations or a single one? As I understand it, each reward would at-least have to be evaluated N times, hence making a minimum budget of KN, right?
>     >
>
>     We appreciate the reviewer’s comment and apologize for any confusion caused by the terminology.  In **Algorithm 1**, \(T\) refers to the number of selection steps, and the total number of iterations is indeed \(T \times N\), where $N$ is the number of training iterations a reward function is trained on before selecting another one. To clarify the notion of “budget” and the allocation of iterations, we provide a more structured explanation below:
>
>     - Budget: In our context, the budget refers to the total number of PPO iterations allowed for training
>     - Number of Selection Steps: $T$
>     - Number of Training Iterations per Selection Step: $N$
>     - Number of Iterations to Train Baselines: `n_iters`
>
>     In our experiments, we fix the total budget to be a fixed multiple of `n_iters` (the number of iterations used to train the baselines, i.e., task reward function and human-designed reward function) for each task and ablate the choice of the multiple. That is, we have `total_iterations_budget = n_iters x B = T x N`, where we ablate the choice of $B$ with $B \in \{5, 10, 15\}$. We choose `N = n_iters / 100`, so that $T = 100 \times B$.
>
>     We note that we can arbitrarily choose values like `1e6` iterations for the budget, but this alone does not provide insight into the relative cost compared to training with the baseline reward functions.
>
>     Regarding **Figure 1**, it is a schematic illustration intended to highlight the trade-off between allocating budget to suboptimal versus optimal reward functions when the budget is limited. The figure depicts an extreme case where \(N = \texttt{n\_iters}\) and \(T < 2 \times \texttt{n\_iters}\). However, this situation does not occur in our experiments. For example, in the **Ant** task, we use \(\texttt{n\_iters} = 1500\) and set \(N = 15\).
>
>
> We hope these clarifications address the concerns raised. If clarifications and additional experiments resolve your concerns, we would be grateful if you could consider raising your score to reflect the improvements.
>
> **References**
>
> [1] Cheng, Ching-An, Andrey Kolobov, and Adith Swaminathan. "Heuristic-guided reinforcement learning." *Advances in Neural Information Processing Systems* 34 (2021): 13550-13563.
>
> [2] Zheng, Zeyu, Junhyuk Oh, and Satinder Singh. "On learning intrinsic rewards for policy gradient methods." Advances in Neural Information Processing Systems 31 (2018).

---

> > ### Comment · Reviewer_sCCd · 2024-11-26
> >
> > Thank you to the authors for their detailed responses and clarifications. After careful consideration, I would like to maintain my current score.
> >
> > The primary reasons for this decision are as follows:
> > 1. The method does not guarantee the optimality of the policy with respect to the base reward (theoretically), which makes it difficult to see how the approach will be practically useful in real-world applications.
> > 2. While access to environment code is feasible in robotics domains, it is generally not true for other scenarios. If the paper specifically targets robotics domains, this focus should be made clear in the writing, as the current framing suggests a more general-purpose method for reward shaping.
> > 3. Methods such as LIRPG and other reward-shaping approaches typically do not require maintaining multiple copies of policies. While the proposed method benefits significantly from parallelizing the search space, it seems to have prohibitive requirements, including access to parallel simulators/environments.
> >
> > Regarding the point about reward generation being part of ORSO, I do not believe that the way it is currently proposed and written supports the claim that reward generation is an integral part of the method.
> >
> > Additionally, I resonate with the points raised by reviewer DEh7 and agree with their assessment. For these reasons, I would like to keep my score unchanged.
> >
> > Thank you again for your thoughtful engagement with the feedback.

---

> > > ### Author Response · Authors · 2024-11-27
> > >
> > > We thank the reviewer for the thoughtful response and for motivating their score. However, we would like to respectfully disagree with some of the points raised by the reviewer.
> > >
> > > > The method does not guarantee the optimality of the policy with respect to the base reward (theoretically), which makes it difficult to see how the approach will be practically useful in real-world applications.
> > > >
> > >
> > > Lack of theoretical optimality does not hinder practicality in real-world applications. While it is true that the method does not theoretically guarantee optimality with respect to the base reward, our empirical evaluation shows significant and statistically significant improvements in performance when training policies with the rewards selected by ORSO. Additionally, it worth noting that optimality of the reward function does not necessarily translate to empirical performance gain. Many existing works [1, 2, 3, 4, 5, 6, 7] use reward shaping without optimality guarantee with respect to the base reward function, but achieve impressive performance in real-world applications.
> > >
> > > > While access to environment code is feasible in robotics domains, it is generally not true for other scenarios. If the paper specifically targets robotics domains, this focus should be made clear in the writing, as the current framing suggests a more general-purpose method for reward shaping.
> > > >
> > >
> > > As the author pointed out, the generation is not the main contribution of our paper, which is the only part of the algorithm that requires access to the environment code. The ORSO framework can be applied to any form of reward function. The choice of using LLMs to generate reward functions is motivated by the success of such methods in EUREKA and the ability to interpret the reward functions (compared to methods such as LIRPG, where the intrinsic reward is a black box model)
> > >
> > > > Methods such as LIRPG and other reward-shaping approaches typically do not require maintaining multiple copies of policies. While the proposed method benefits significantly from parallelizing the search space, it seems to have prohibitive requirements, including access to parallel simulators/environments.
> > > >
> > >
> > > This is a great observation. ORSO indeed requires additional space requirements to maintain multiply policy networks. However, this is generally not a prohibitive computational requirement. In Figure 3 of the updated PDF, we show that ORSO has significant (up to **16x fewer GPUs needed** to achieve the same performance in the same time) gains in terms of necessary compute compared to EUREKA. All our experiments can be indeed run on a single commercial GPU (e.g,. a 3090 Ti or even a 2080 Ti).
> > >
> > > We also believe access to parallel simulators/environments is not an unreasonable requirement. Simulators such as Isaac Gym, gymnax and MuJoCo MJX are now widely adopted and available to the community. Moreover, GPU parallelization of environment is not necessary for ORSO.
> > >
> > > We hope the additional clarifications will highlight the impact of our proposed framework. We are happy to engage in further discussions with the reviewer and clarify any additional questions they may have.
> > >
> > > **References**
> > >
> > > [1] Liu, Minghuan, et al. "Visual whole-body control for legged loco-manipulation." *arXiv preprint arXiv:2403.16967* (2024).
> > >
> > > [2] Margolis, Gabriel B., et al. "Rapid locomotion via reinforcement learning." *The International Journal of Robotics Research* 43.4 (2024): 572-587.
> > >
> > > [3] Margolis, Gabriel B., and Pulkit Agrawal. "Walk these ways: Tuning robot control for generalization with multiplicity of behavior." *Conference on Robot Learning*. PMLR, 2023.
> > >
> > > [4] Lee, Joonho, et al. "Learning quadrupedal locomotion over challenging terrain." *Science robotics* 5.47 (2020): eabc5986.
> > >
> > > [5] Ma, Yecheng Jason, et al. "Eureka: Human-level reward design via coding large language models." *arXiv preprint arXiv:2310.12931* (2023).
> > >
> > > [6] Ha, Huy, et al. "Umi on legs: Making manipulation policies mobile with manipulation-centric whole-body controllers." *arXiv preprint arXiv:2407.10353* (2024).
> > >
> > > [7] Kaufmann, Elia, et al. "Champion-level drone racing using deep reinforcement learning." *Nature* 620.7976 (2023): 982-987.

---

> > > ### Author Response · Authors · 2024-11-27
> > > **Additional Comment on Theoretical Contribution**
> > >
> > > We would also like to add that we provide a novel analysis of D3RB, which is in stark contrast with the regret guarantees of the original paper. Namely, our guarantee depend on the true regret coefficients rather than the monotonic ones.
> > >
> > > Therefore, even though our there is no guarantee that the learned policy will achieve the same optimal behavior as the optimal policy for the task reward (which might not achievable because of the task reward not being amenable to optimization), we guarantee that ORSO with D3RB will eventually select and train with the optimal reward function within the set.

---

> > > > ### Author Response · Authors · 2024-12-03
> > > >
> > > > Dear reviewer,
> > > >
> > > > as the discussion period is coming to an end, we would be grateful for any further feedback on our earlier response. Please let us know if you have any questions or need clarification. Otherwise, we kindly ask you to consider adjusting your score.
> > > >
> > > > We truly value your insights and participation in the review process.
> > > >
> > > > Kind regards,
> > > >
> > > > The authors

---

### Meta-Review · Area_Chair_qHTm · 2024-12-21

**Metareview:**

This paper proposes a reward design solution method that uses automatically generated candidate reward functions (from an LLM) and optimizes a policy for each reward function. The method then dynamically selects which reward function is likely to be the best one under the original reward function. The algorithm iterates on training and evaluating each candidate reward function for a fixed amount of time. The algorithm is not applicable to every problem, as noted by one reviewer, and makes a strong assumption that good candidate reward functions can be generated. However, the paper does demonstrate that it can solve the problem it set out to. Thus, I recommend this paper for acceptance.

**Additional Comments On Reviewer Discussion:**

There were several discussions with the reviewers. One reviewer (sCCd) maintained their score and raised concerns of the method's applicability. However, I do not believe these concerns are cause for a rejection. In one case, the reviewer raised their score to an eight, leading to a sufficiently positive reception of the paper to consider acceptance. While the paper is not perfect, there do not appear to be major flaws warranting rejection.

---

### Decision · Program_Chairs · 2025-01-22

Accept (Poster)